# Overcoming the Curse of Dimensionality in Reinforcement Learning Through Approximate Factorization

**Chenbei Lu** [1]  **Laixi Shi** [2]  **Zaiwei Chen** [3]  **Chenye Wu** [4]  **Adam Wierman** [5]

## Abstract

Factored Markov Decision Processes (FMDPs) offer a promising framework for overcoming the curse of dimensionality in reinforcement learning (RL) by decomposing high-dimensional MDPs into smaller and independently evolving components. Despite their potential, existing studies on FMDPs face three key limitations: reliance on perfectly factorizable models, suboptimal sample complexity guarantees for model-based algorithms, and the absence of model-free algorithms. To address these challenges, we introduce *approximate factorization*, which extends FMDPs to handle imperfectly factored models. Moreover, we develop a model-based algorithm and a model-free algorithm (in the form of variance-reduced Q-learning), both achieving the first near-minimax sample complexity guarantees for FMDPs. A key novelty in the design of these two algorithms is the development of a graph-coloring-based optimal synchronous sampling strategy. Numerical simulations based on the wind farm storage control problem corroborate our theoretical findings.

## 1. Introduction

Reinforcement learning (RL) (Sutton & Barto, 2018) has become an increasingly popular framework for solving sequential decision-making problems in recent years (Kober et al., 2013; Haydari & Yılmaz, 2020; Chen et al., 2022; Charpentier et al., 2021). However, despite its promising potential, RL still faces significant challenges, in particular the *curse of dimensionality* (Sutton & Barto, 2018).

[1]Institute for Interdisciplinary Information Sciences, Tsinghua University [2]Department of Electrical and Computer Engineering, Johns Hopkins University [3]Edwardson School of Industrial Engineering, Purdue University [4]School of Science and Engineering, The Chinese University of Hong Kong, Shenzhen [5]Computing & Mathematical Sciences Department, Caltech. Correspondence to: Chenye Wu <wuchenye@cuhk.edu.cn>.

*Proceedings of the 42ⁿᵈ International Conference on Machine Learning*, Vancouver, Canada. PMLR 267, 2025. Copyright 2025 by the author(s).

Specifically, the size of the state and action spaces grows exponentially with the dimension of the RL problem, making it difficult—if not impossible—to efficiently represent, learn, and optimize policies in high-dimensional settings.

To mitigate the curse of dimensionality, a common practical approach is to employ function approximation to approximate the solution of an RL problem within a prespecified function class, such as neural networks. While effective in certain applications, these methods often lack strong theoretical guarantees, with most existing work focusing on linear function approximation (Tsitsiklis & Van Roy, 1996; Bhandari et al., 2018; Srikant & Ying, 2019; Chen et al., 2023) and some extensions to nonlinear methods, which typically rely on strong assumptions (Fan et al., 2020; Xu & Gu, 2020). Consequently, achieving provable sample efficiency for large-scale sequential decision-making remains a significant challenge in RL.

Fortunately, many real-world applications, when modeled as Markov Decision Processes (MDPs), exhibit structured transition probabilities and reward functions that can be leveraged to design algorithms with improved sample complexity. For instance, in robotic control, high-dimensional state spaces often consist of substates that evolve independently or depend on a low-dimensional subset of the previous state. Specifically, consider a warehouse with multiple mobile robots transporting goods, each robot typically plans and moves based on its own position and objective, largely independent of the internal states of other robots unless they are highly close. These problems can be modeled using the factored MDP (FMDP) framework (Osband & Van Roy, 2014), which decomposes the original MDP into smaller and independently evolving MDPs. In this framework, the sample complexity scales with the sum, rather than the product, of the individual state-action space sizes, thereby offering a potential solution to the curse of dimensionality.

While promising, the FMDP framework has several major limitations: (1) it assumes that the original MDP can be perfectly decomposed into smaller MDPs; (2) for model-based algorithms, a significant gap remains between the best-known sample complexity and the theoretical lower bound (Chen et al., 2020; Xu & Tewari, 2020); and (3) no provable model-free algorithms currently exist for FMDPs.

Addressing these challenges is essential for improving RL efficiency in large-scale problems.

### 1.1. Contributions

The main contributions of our work are threefold.

**Approximate Factorization of MDP and Efficient Sampling.** We propose an approximate factorization scheme that flexibly decomposes any MDP into low-dimensional components, while allowing for imperfect factorization. To facilitate algorithm design, we introduce a multi-component factorized synchronous sampling approach, formulated as a cost-optimal graph coloring problem. Unlike previous FMDP methods, which estimate transitions for a single component per sample, our approach enables simultaneous sampling of multiple components from a single sample, significantly reducing the sample complexity for both model-based and model-free algorithms.

**Model-Based Algorithm.** We propose a novel model-based RL algorithm that uses synchronous sampling to exploit low-dimensional structures from approximate factorization. The algorithm achieves provable problem-dependent sample complexity (cf. Theorem 5.1), outperforming existing minimax-optimal bounds for standard MDPs (Azar et al., 2012). For the special case of FMDPs, it improves the best-known sample complexity (Chen et al., 2020) by up to a factor equal to the number of components and matches the instance-dependent lower bound (Xu & Tewari, 2020).

**Model-Free Algorithm.** We introduce a model-free algorithm, *Variance-Reduced Q-Learning with Approximate Factorization* (VRQL-AF), which also builds on the synchronous sampling method. VRQL-AF achieves the same problem-dependent sample complexity guarantees (cf. Theorem 6.1) as our model-based algorithm, up to logarithmic factors, and outperforms existing minimax-optimal algorithms for standard MDPs (Wainwright, 2019b). In the special case of FMDPs, VRQL-AF is, to the best of our knowledge, the first provable model-free algorithm with near-optimal sample complexity guarantees. This improvement is enabled by a tailored factored empirical Bellman operator induced by synchronous sampling, combined with a variance-reduction technique that minimizes variance across factored components.

To support the above contributions, we include a detailed literature review in Appendix A and comprehensive experiments in Appendix H, covering both synthetic FMDPs and a real-world wind farm storage control application.

## 2. Model and Background

We consider an infinite-horizon discounted MDP $M = (\mathcal{S}, \mathcal{A}, P, r, \gamma)$, where $\mathcal{S}$ is the finite state space, $\mathcal{A}$ is the finite action space, $P$ is the *unknown* transition kernel, with $P(s' \mid s, a)$ denoting the transition probability from state $s$ to $s'$ given action $a$, $r : \mathcal{S} \times \mathcal{A} \to [0, 1]$ is the *unknown* reward function, and $\gamma \in (0, 1)$ is the discount factor. Given a policy $\pi : \mathcal{S} \to \Delta(\mathcal{A})$ (where $\Delta(\mathcal{A})$ denotes the set of probability distributions supported on $\mathcal{A}$), its Q-function $Q^\pi \in \mathbb{R}^{|\mathcal{S}||\mathcal{A}|}$ is defined as

$$Q^\pi(s,a) = \mathbb{E}_{\pi,P}\left[\sum_{t=0}^{\infty} \gamma^t r(s_t, a_t) \,\middle|\, s_0 = s, a_0 = a\right]$$

for all $(s, a)$, where $\mathbb{E}_{\pi,P}[\,\cdot\,]$ denotes the expectation over trajectories generated by the transition kernel $P$ and the policy $\pi$. With the Q-function defined above, the agent's goal is to find an optimal policy $\pi^*$ such that its associated Q-function is uniformly maximized over all state-action pairs. It has been shown that such an optimal policy always exists (Puterman, 2014).

At the heart of solving an MDP is the Bellman equation, which states that the optimal Q-function, denoted by $Q^*$, is the unique solution to the fixed-point equation

$$Q = \mathcal{H}(Q),$$

where $\mathcal{H} : \mathbb{R}^{|\mathcal{S}||\mathcal{A}|} \to \mathbb{R}^{|\mathcal{S}||\mathcal{A}|}$ is the Bellman optimality operator defined as

$$[\mathcal{H}(Q)](s,a) = \mathbb{E}_{s' \sim P(\cdot|s,a)}[r(s,a) + \gamma \max_{a' \in \mathcal{A}} Q(s', a')]$$

for all $(s, a)$. Once $Q^*$ is obtained, an optimal policy can be computed by choosing actions greedily based on $Q^*$.

Suppose that the model parameters of the MDP are known. Then, we can efficiently compute $Q^*$ through the value iteration method: $Q_{k+1} = \mathcal{H}(Q_k)$. Since the operator $\mathcal{H}(\cdot)$ is a contraction mapping (Sutton & Barto, 2018), the value iteration method converges geometrically to $Q^*$ (Banach, 1922). However, in RL, the agent does not know the model parameters of the underlying MDP, making direct value iteration infeasible. A common solution is the model-based approach, which estimates the model parameters through empirical sampling and then applies value iteration to the estimated model. Alternatively, model-free methods, such as Q-learning (Watkins & Dayan, 1992), bypass model estimation by directly solving the Bellman equation using stochastic approximation (Robbins & Monro, 1951).

Throughout this paper, we work under a widely used sampling mechanism setting – the generative model (Kearns et al., 2002; Kakade, 2003). This model allows us to query any state-action pair $(s, a)$ to obtain a random sample of the next state $s'$ and the immediate reward $r(s, a)$ according to the underlying transition kernel and reward function. While not explored in this work, extending our results to the Markov sampling setting—provided the behavior policy induces a uniformly ergodic Markov chain—is a promising direction for future research.

## 3. Approximate Factorization of MDPs

In this section, we introduce the concept of *approximate factorization*. Assume without loss of generality that each state $s$ can be represented as an $n$-dimensional vector. Accordingly, the overall state space $\mathcal{S}$ can be written as $\mathcal{S} = \prod_{i=1}^{n} \mathcal{S}_i$, where $\mathcal{S}_i \subseteq \mathbb{R}$ for all $i \in [n] := \{1, 2, \ldots, n\}$. Similarly, assume without loss of generality that each action is an $m$-dimensional vector and $\mathcal{A} = \prod_{j=1}^{m} \mathcal{A}_j$, where $\mathcal{A}_j \subseteq \mathbb{R}$ for all $j \in [m]$. Throughout this paper, each component $s[i] \in \mathcal{S}_i$ of a state $s \in \mathbb{R}^n$ is called a substate. A subaction is defined analogously. The following definitions of *scope*, *scope set* and *scope variable*, will be useful for introducing approximate factorization.

**Definition 3.1.** Consider a factored $d$-dimensional set $\mathcal{X} = \prod_{i=1}^{d} \mathcal{X}_i$, where $\mathcal{X}_i \subseteq \mathbb{R}$ for all $i \in [d]$. For any $Z \subseteq [d]$ (which we call a *scope*), the corresponding *scope set* $\mathcal{X}[Z]$ is defined as $\mathcal{X}[Z] = \prod_{i \in Z} \mathcal{X}_i$. Any element $x[Z] \in \mathcal{X}[Z]$ (which is a $|Z|$-dimensional vector) is called a *scope variable*. When $Z$ contains only one element, i.e., $Z = \{i\}$ for some $i \in [d]$, we denote $x[i]$ as $x[\{i\}]$.

An approximate factorization is characterized by a tuple $\omega = (\omega^P, \omega^R)$, which specifies how both the transition kernel and the reward function are factorized. In the following, we elaborate on the definitions of $\omega^P$ and $\omega^R$ in detail.

### 3.1. Approximate Factorization of the Transition Kernel

A factorization scheme $\omega^P$ of the transition kernel is characterized by a tuple

$$\omega^P = \big( K_\omega, \{Z_k^S \subseteq [n] \mid k \in [K_\omega]\},$$
$$\{Z_k^P \subseteq [n+m] \mid k \in [K_\omega]\}\big),$$

where

- $K_\omega \in [n]$ denotes the number of components into which we partition the transition kernel;

- $\{Z_k^S \mid k \in [K_\omega]\}$ is a collection of scopes over the $n$-dimensional state space $\mathcal{S}$. This collection forms a partition of $[n]$, i.e., $\bigcup_{k=1}^{K_\omega} Z_k^S = [n]$ and $Z_{k_1}^S \cap Z_{k_2}^S = \emptyset$ for any $k_1 \neq k_2$. For each scope $Z_k^S \subseteq [n]$, the associated subspace is denoted by $\mathcal{S}[Z_k^S]$;

- $\{Z_k^P \mid k \in [K_\omega]\}$ is a collection of scopes over the joint state-action space $\mathcal{X} := \mathcal{S} \times \mathcal{A}$. For example, in an MDP with a 3-dimensional state space and a 2-dimensional action space, if $Z_k^P = \{1, 2, 4\}$, then the associated scope set is $\mathcal{X}[Z_k^P] = \mathcal{S}_1 \times \mathcal{S}_2 \times \mathcal{A}_1$.

For simplicity of notation, we use $-Z_k^S := [n] \setminus Z_k^S$ to denote the complement of $Z_k^S$ with respect to $[n]$. The factorization scheme $\omega^P$ characterizes the component-wise dependency structure within the transition kernel. To motivate

our approximate factorization scheme, we first consider the special case where the transition kernel $P$ is *perfectly factorizable* with respect to $\omega^P$. That is, for all $x = (s, a) \in \mathcal{X}$, $s' \in \mathcal{S}$, and $k \in [K_\omega]$, we have

$$P(s' \mid x) = \prod_{k=1}^{K_\omega} P_k(s'[Z_k^S] \mid x[Z_k^P]), \qquad (1)$$

where $P_k(\cdot \mid x[Z_k^P])$ is a valid probability distribution for all $x[Z_k^P] \in \mathcal{X}[Z_k^P]$. Eq. (1) implies that the transitions of $s'[Z_k^S]$ for different $k$ are conditionally independent and each depends only on the corresponding substate-subaction pair in $\mathcal{X}[Z_k^P]$. An MDP that admits both perfect transition and reward decompositions (cf. Section 3.2) is called an FMDP (Osband & Van Roy, 2014). In this case, one only needs to solve $K_\omega$ smaller MDPs rather than the original high-dimensional one, thereby significantly reducing the complexity of finding an optimal policy.

As discussed in Section 1, while the FMDP framework offers a promising way to address the curse of dimensionality, the assumption of perfect factorization is relatively restrictive. To overcome this, we propose a more general scheme of approximate factorization, which allows deviations from the exact structure in Eq. (1). Specifically, to develop an approximate factorization scheme (i.e., to get an estimate of the "marginal" transition probabilities $P(s'[Z_k^S] \mid x[Z_k^P])$ in Eq. (1)), we define the set of feasible marginal transition probabilities in the following. For any $k \in [K_\omega]$, let

$$\mathcal{P}_k = \bigg\{ P_k \in \mathbb{R}^{|\mathcal{X}[Z_k^P]| \times |\mathcal{S}[Z_k^S]|} \Big| \exists c \in \Sigma^{|\mathcal{X}[-Z_k^P]|}, \forall s'[Z_k^S] \in \mathcal{S}[Z_k^S],$$
$$x[Z_k^P] \in \mathcal{X}[Z_k^P], \text{ s.t. } P_k(s'[Z_k^S] \mid x[Z_k^P]) =$$
$$\sum_{x'[-Z_k^P]} c[x'[-Z_k^P]] P(s'[Z_k^S] \mid x[Z_k^P], x'[-Z_k^P]) \bigg\},$$

where we use $\Sigma^d$ to denote the $d$-dimensional probability simplex. Then, we choose $P_k \in \mathcal{P}_k$ arbitrarily for all $k \in [K_\omega]$. As we will see in later sections, the development of our theoretical results does not rely on the specific choice of $P_k \in \mathcal{P}_k$. To understand the definition of $\mathcal{P}_k$, again consider the perfectly factorizable setting, in which case $P(s'[Z_k^S] \mid x[Z_k^P], x'[-Z_k^P]) = P(s'[Z_k^S] \mid x[Z_k^P], x''[-Z_k^P])$ for any $x'[-Z_k^P], x''[-Z_k^P] \in \mathcal{X}[-Z_k^P]$ because the transition of $s'[Z_k^S]$ does not depend on the substate-subaction pairs in $\mathcal{X}[-Z_k^P]$. As a result, $\mathcal{P}_k$ is a singleton set for all $k$, which is consistent with Eq. (1). Coming back to the case where the underlying MDP is not perfectly factorizable, choosing $P_k$ from $\mathcal{P}_k$ can be viewed as a way of approximating the "marginal" transition probabilities.

Given an approximation scheme $\omega^P$, we define the *approximation error* with respect to the transition kernel as

$$\Delta_\omega^P = \sup_{P_1 \in \mathcal{P}_1, \ldots, P_{K_\omega} \in \mathcal{P}_{K_\omega}} \max_{s' \in \mathcal{S}, x \in \mathcal{X}} \big| P(s' \mid x)$$

$$-\prod_{k=1}^{K_\omega} P_k(s'[Z_k^S] \mid x[Z_k^P])|. \tag{2}$$

To understand $\Delta_\omega^P$, see that for any $s' \in \mathcal{S}$ and $x \in \mathcal{X}$, we have the following:

$$P(s' \mid x) - \prod_{k=1}^{K_\omega} P_k(s'[Z_k^S] \mid x[Z_k^P]) = \Delta_1 + \Delta_2,$$

where

$$\Delta_1 = P(s' \mid x) - \prod_{k=1}^{K_\omega} P_k(s'[Z_k^S] \mid x),$$

$$\Delta_2 = \prod_{k=1}^{K_\omega} P_k(s'[Z_k^S] \mid x) - \prod_{k=1}^{K_\omega} P_k(s'[Z_k^S] \mid x[Z_k^P]).$$

The first term $\Delta_1$ represents the error due to ignoring the correlation between the transitions of $s'[Z_k^S]$ for different $k$, and vanishes if the transitions of substates from different scope sets are independent. The second term $\Delta_2$ arises due to ignoring the dependence of the transitions of $s'[Z_k^S]$ from $x[-Z_k^P]$. Note that when the MDP is perfectly factorizable, we have $\Delta_\omega^P = 0$ (cf. Eq. (1)).

### 3.2. Approximate Factorization of the Reward Function

For the reward function, the factorization is characterized by the tuple

$$\omega^R = \left(\ell_\omega, \{Z_i^R \in [n+m] \mid i \in [\ell_\omega]\}\right),$$

where

- $\ell_\omega$ is a positive integer representing the number of components into which we decompose the reward function;

- $\{Z_i^R \mid i \in [\ell_\omega]\}$ is a set of scopes with respect to the joint state-action space $\mathcal{X}$.

Under this factorization scheme, the global reward function $r(x)$ is approximated by the sum of "local" reward functions from $\{r_i : \mathcal{X}[Z_i^R] \to [0,1]\}_{i \in [\ell_\omega]}$, where each $r_i$ depends only on the scope variable $x[Z_i^R]$. Similarly, given an approximation scheme $\omega^R$ for the reward function, we define its approximation error as

$$\Delta_\omega^R = \max_{x \in \mathcal{X}} \left| r(x) - \sum_{i=1}^{\ell_\omega} r_i(x[Z_i^R]) \right|, \tag{3}$$

which is a measurement of the deviation from our approximation to the true reward function. The approximation error $\Delta_\omega^R$ arises due to the fact that the reward function $r(x)$ may depend on substate-subaction pairs outside the scope set $\mathcal{X}[Z_i^R]$. In the special case where the underlying MDP can be perfectly factorized into several small MDPs and the sum of their reward functions is equal to the original reward function, we have $\Delta_\omega^R = 0$.

For further illustration of our approximate factorization scheme, see Appendix B for an example.

## 4. Factorized Multi-Component Synchronous Sampling

To introduce our model-based and model-free algorithms, we first present an efficient sampling algorithm that exploits the factorization structure. Consider classical model-based RL, which typically involves two steps: (1) estimating the transition kernel and reward function through empirical sampling, and (2) applying value iteration or policy iteration to the estimated model. The curse of dimensionality arises primarily in the first step, where each state-action pair must be sampled multiple times, resulting in sample complexity proportional to $|\mathcal{S}||\mathcal{A}|$. By leveraging approximate factorization, we significantly reduce this burden. For clarity, we focus here on estimating the transition kernel; the same approach applies to the reward function.

The goal is to construct an approximation $\widehat{P}$ of the original transition kernel $P$. We do this by estimating a collection of low-dimensional transition kernels $\{\widehat{P}_k \mid k \in [K_\omega]\}$ and computing $\widehat{P}$ according to

$$\widehat{P}(s' \mid x) = \prod_{k=1}^{K_\omega} \widehat{P}_k(s'[Z_k^S] \mid x[Z_k^P]) \tag{4}$$

for all $s' \in \mathcal{S}$ and $x \in \mathcal{S} \times \mathcal{A}$. Instead of sampling exhaustively from the global state-action space, we sample full transitions and extract the relevant substate-subaction dimensions, as is standard in factored MDPs (Osband & Van Roy, 2014).

To estimate $\widehat{P}_k(s'[Z_k^S] \mid x[Z_k^P])$ for any $k \in [K_\omega]$, we define the restricted sampling set $X_k^P$ as:

$$X_k^P = \{x \in \mathcal{X} \mid x[-Z_k^P] = x^{\text{default}}[-Z_k^P]\}, \tag{5}$$

where $x^{\text{default}} \in \mathcal{X}$ is an arbitrary but fixed reference point. Sampling from $X_k^P$ allows us to estimate the $k$-th factorized component without needing to cover the full state-action space. For each $x \in X_k^P$, we sample the next state $N$ times, obtaining samples $\{s_{x,i}^k\}_{i \in [N]}$. The empirical transition kernel $\widehat{P}_k$ is then computed as

$$\widehat{P}_k(s'[Z_k^S] \mid x[Z_k^P]) = \frac{1}{N} \sum_{i=1}^{N} \mathbb{1}(s_{x,i}^k[Z_k^S] = s'[Z_k^S]) \tag{6}$$

for all $s'[Z_k^S] \in \mathcal{S}[Z_k^S]$ and $x \in X_k^P$. The overall estimate $\widehat{P}$ is then obtained via Eq. (4).

Using this approach, the total number of required samples is $\sum_{k=1}^{K_\omega} |\mathcal{X}[Z_k^P]|N$. Since the total substate-subaction space

size, $\sum_{k=1}^{K_\omega} |\mathcal{X}[Z_k^P]|$, is typically much smaller, often exponentially smaller, than the size of the full state-action space $|\mathcal{S}||\mathcal{A}|$, the overall sample complexity is significantly reduced compared to classical model-based RL.

This also explains why the best-known sample complexity depends on $\sum_{k=1}^{K_\omega} |\mathcal{X}[Z_k^P]|$ (Chen et al., 2020). To improve this to the lower bound of $\max_k |\mathcal{X}[Z_k^P]|$, we show that the key is to leverage structural properties of factored MDPs and to design synchronous sampling strategies that allow multiple factors to be sampled simultaneously.

### 4.1. Structure-Aware Sample Reuse Strategies

The sampling method described above can be further optimized by exploiting the structure of the factorization scheme to improve sample efficiency. Previously, the samples $\{s_{x,i}^k\}_{i\in[N]}$ were used solely to estimate $\widehat{P}_k(s'[Z_k^S] \mid x[Z_k^P])$ for component $k$. However, these samples may also be reused to estimate other low-dimensional transition kernels, depending on the relationships among their associated scopes. To formalize this idea, we introduce two key strategies that leverage the structural relationships between the scope sets of different components.

**Synchronous Sampling with Inclusive Scopes.** For any $k_1, k_2 \in [K_\omega]$ such that $Z_{k_1}^P \subseteq Z_{k_2}^P$, the samples used to estimate the transition kernel of component $k_2$ can be reused to estimate that of component $k_1$. Specifically, consider the sampling set

$$X_{k_2}^P = \{x \in \mathcal{X} \mid x[-Z_{k_2}^P] = x^{\text{default}}[-Z_{k_2}^P]\}$$

for component $k_2$, where $x^{\text{default}}[-Z_{k_2}^P]$ is a fixed, arbitrary element from $\mathcal{X}[-Z_{k_2}^P]$. By sampling from each $x \in X_{k_2}^P$, we obtain samples of the next state $\{s_{x,i}^{k_2}\}_{i\in[N]}$. Since $Z_{k_1}^P \subseteq Z_{k_2}^P$, these samples inherently contain information about the transitions of component $k_1$. Therefore, we can estimate the transition probabilities for component $k_1$ according to Eq. (6) with $k = k_1$ using the same samples, where the sampling set $X_{k_1}^P$ is defined based on $X_{k_2}^P$ as

$$X_{k_1}^P = \{x \in X_{k_2}^P \mid x[Z_{k_2}^P \setminus Z_{k_1}^P] = x^{\text{default}}[Z_{k_2}^P \setminus Z_{k_1}^P]\}.$$

This reuse of samples improves overall sample efficiency by avoiding redundant sampling for component $k_1$.

**Synchronous Sampling with Exclusive Scopes.** For any $k_1, k_2 \in [K_\omega]$ such that their associated scopes are disjoint, i.e., $Z_{k_1}^P \cap Z_{k_2}^P = \emptyset$, we can estimate the transitions for both components simultaneously using shared samples. Specifically, define the *joint sampling set* $X_{k_1,k_2}^P$ as

$$X_{k_1,k_2}^P = \{(x[Z_{k_1}^P]_{(i \bmod |\mathcal{X}[Z_{k_1}^P]|+1)}, x[Z_{k_2}^P]_{(i \bmod |\mathcal{X}[Z_{k_2}^P]|+1)},$$
$$x^{\text{default}}[-(Z_{k_1}^P \cup Z_{k_2}^P)]) \mid i \in [D_{\max}]\},$$

where $D_{\max} = \max(|\mathcal{X}[Z_{k_1}^P]|, |\mathcal{X}[Z_{k_2}^P]|)$, and $x[Z_{k_1}^P]_{(i)}$ denotes the $i$-th element in a fixed ordering of $\mathcal{X}[Z_{k_1}^P]$. The modulo operation ensures that we cycle through all possible values of each component's state-action space. By sampling from each $x \in X_{k_1,k_2}^P$ $N$ times, we obtain samples $\{s_{x,i}\}_{i=1}^N$, which are used to estimate the transition probabilities for both components according to Eq. (6). Appendix C provides an example of synchronous sampling with disjoint scopes. This strategy improves sample efficiency by reducing the total number of samples compared to sampling each component independently.

### 4.2. Cost-Optimal Synchronous Sampling Strategy

Building on the aforementioned two key strategies, we next present our sampling approach. Recall that we have $K_\omega$ components associated with scope sets $Z_1^P, Z_2^P, \ldots, Z_{K_\omega}^P$. For each $k \in [K_\omega]$, we define its corresponding sampling cost as the size of the scope set $\mathcal{X}[Z_k^P]$. Using the inclusive scope property, we first eliminate components whose scope sets are subsets of others. Let $\mathcal{K}_\omega^*$ denote the set of remaining components defined as

$$\mathcal{K}_\omega^* = \{k \in [K_\omega] \mid \forall k_1 \neq k \text{ such that } Z_k^P \not\subset Z_{k_1}^P\}.$$

Next, to use the exclusive scope property, we divide $\mathcal{K}_\omega^*$ into subsets $\{\mathcal{G}_i\}_{1\leq i\leq \kappa_p}$ (where $\kappa_p$ denotes the total number of subsets) such that within each subset $\mathcal{G}_i$, the components have disjoint scopes, i.e., $Z_{k_1}^P \cap Z_{k_2}^P = \emptyset$ for all $k_1, k_2 \in \mathcal{G}_i$. This enables us to apply the strategy of synchronous sampling with exclusive scopes within each subset. Specifically, for each subset $\mathcal{G}_k$ of $\mathcal{K}_\omega^*$, we construct the joint sampling sets as

$$X_{\mathcal{G}_k}^P = \{\{x[Z_j^P]_{(i \bmod |\mathcal{X}[Z_j^P]|+1)}\}_{j\in\mathcal{G}_k},$$
$$x^{\text{default}}[-(\cup_{j\in\mathcal{G}_k} Z_j^P)] \mid i \in [D_{\max,k}]\},$$

where $D_{\max,k} = \max_{j\in\mathcal{G}_k} |\mathcal{X}[Z_j^P]|$ ensures that all scope sets associated with this subset are sampled. Therefore, the cost of sampling a joint sampling set is determined by the largest scope sets in the group as

$$c_k(\mathcal{G}_k) = D_{\max,k} = \max_{j\in\mathcal{G}_k} |\mathcal{X}[Z_j^P]|.$$

Our objective is to minimize the total sampling cost across all groups defined as

$$\min_{\kappa_p} \min_{\mathcal{G}_1,\ldots,\mathcal{G}_{\kappa_p}} \sum_{k=1}^{\kappa_p} c_k(\mathcal{G}_k).$$

This problem is closely related to the *Graph Coloring Problem* (GCP) (Jensen & Toft, 2011; Karp, 2010). In Appendix G, we detailedly elaborate on the connection between our

cost-optimal sampling problem and the GCP. The solution yields the optimal number of groups $\kappa_p \in [K_\omega]$ and the division scheme $\{\mathcal{G}_k\}_{k\in[\kappa_p]}$. Notably, $\kappa_p$ corresponds to the *chromatic number* (Erdős & Hajnal, 1966) (the minimal number of colors required) in GCP, which is often much smaller than the number of nodes in the graph (Khot, 2001; Sopena, 1997; Goddard & Xu, 2012). This result translates to $\kappa_p \ll K_\omega$ in our context, leading to substantial sampling cost reductions. The detailed discussion on $\kappa_p$ can refer to Section 5.2.

### 4.3. MDP Model Estimation

Given $\{\mathcal{G}_k\}_{k\in[\kappa_p]}$ and the joint sampling sets $\{X^P_{\mathcal{G}_k}\}_{k\in[\kappa_p]}$, we are ready to sample and estimate the transition kernel. Specifically, for each state $x \in X^P_{\mathcal{G}_k}$, we perform $N$ sampling trials to obtain next-state samples based on the generative model. Specifically, we generate $N$ samples $\{s^k_{x,i}\}^N_{i=1}$ by sampling from $P(\cdot \mid x)$ as detailed in Algorithm 1. For each component $j \in \mathcal{G}_k$ and $s'[Z^S_j] \in \mathcal{S}[Z^S_j]$, we compute the empirical transition kernel according to

$$\widehat{P}_j(s'[Z^S_j] \mid x[Z^P_j]) = \frac{1}{N} \sum\nolimits_{i\in[N]} \mathbb{1}(s^k_{x,i}[Z^S_j] = s'[Z^S_j]).$$

---

**Algorithm 1** Cost-Optimal Factorized Synchronous Sampling Algorithm

1: **Input:** Approximate factorization $\omega = (\omega^P, \omega^R)$.
2: Solve the optimal division scheme represented by $\{\kappa_p, \{\mathcal{G}_k\}_{k\in[\kappa_p]}\}$;
3: Construct the sampling sets $\{X^P_{\mathcal{G}_k}\}_{k\in[\kappa_p]}$;
4: **for** $k = 1, 2, \cdots, \kappa_p$ **do**
5:     Sample from each state-action pair $x \in X^P_{\mathcal{G}_k}$ for $N$ times to obtain $\{s^k_{x,i}\}_{x\in X^P_{\mathcal{G}_k}, i\in[N]}$.
6: **end for**
7: **Output:** Samples $\{s^k_{x,i}\}_{k\in[\kappa_p], x\in X^P_{\mathcal{G}_k}, i\in[N]}$.

---

To estimate the reward function components, we employ similar inclusive and exclusive scope strategies. By constructing a sampling set tailored to these strategies, we can then proceed with learning the reward function.

## 5. Model-Based RL with Approximate Factorization

In this section, we focus on model-based RL and propose the *model-based Q-value iteration with approximate factorization* algorithm, which offers improved sample complexity guarantees.

### 5.1. Algorithm Design

The algorithm is summarized in Algorithm 2, which consists of two steps: estimating the model parameters and performing Q-value iteration on the estimated model.

---

**Algorithm 2** Model-Based Q-Value Iteration with Approximate Factorization

1: **Input:** Positive integer $T$ and initialization $\widehat{Q}_0(s,a) = 0$ for all $(s,a) \in \mathcal{S} \times \mathcal{A}$.
2: Compute empirical transition kernel $\widehat{P}$ and the reward function $\widehat{r}$ through synchronous sampling Algorithm 1 and Eqs. (7) and (8).
3: **for** $t = 1, 2, \cdots, T$ **do**
4:     $\widehat{Q}_t(s,a) = \mathbb{E}_{s'\sim\widehat{P}(s'|s,a)}[\widehat{r}(s,a) + \gamma \max_{a'} \widehat{Q}_{t-1}(s',a')]$ for all $(s,a)$.
5: **end for**
6: **Output:** Estimated Q-value function $\widehat{Q}^*_\omega = \widehat{Q}_T$.

---

The first step involves estimating the transition kernel and the reward function (see Algorithm 2, Line 2). Specifically, in Section 4, we explained how to estimate the transition probabilities $\widehat{P}_k$ for each component $k \in [K_\omega]$. For any $s' \in \mathcal{S}$ and $x \in \mathcal{X}$, the overall transition probability $\widehat{P}(s' \mid x)$ is computed by combining the individual component estimates as follows:

$$\widehat{P}(s' \mid x) = \prod\nolimits_{k=1}^{K_\omega} \widehat{P}_k(s'[Z^S_k] \mid x[Z^P_k]). \qquad (7)$$

Similarly, for the reward function, we aggregate the estimated rewards $\widehat{r}_i$ for each component $i \in [\ell_\omega]$ to obtain the overall reward function

$$\widehat{r}(x) = \sum\nolimits_{i=1}^{\ell_\omega} \widehat{r}_i(x[Z^R_i]). \qquad (8)$$

The second step is to apply the value iteration method using these estimated model parameters (see Algorithm 2, Lines 3–5). In particular, Line 4 of Algorithm 2 represents the empirical version of the Bellman iteration. Through value iteration, we can compute the desired optimal-$Q$-value function and derive the corresponding greedy policy as the final solution to the RL problem.

### 5.2. Sample Complexity Guarantees

We now present the sample complexity guarantees of Algorithm 2; the proof can be found in Appendix D. Before proceeding, without loss of generality, we assume that $|\mathcal{X}[Z^P_1]| \geq |\mathcal{X}[Z^P_2]| \geq \cdots \geq |\mathcal{X}[Z^P_{K_\omega}]|$ and $|\mathcal{X}[Z^R_1]| \geq |\mathcal{X}[Z^R_2]| \geq \cdots \geq |\mathcal{X}[Z^R_{\ell_\omega}]|$, i.e., the component scope sets are ordered in descending order based on their cardinality.

**Theorem 5.1.** *Given any approximate factorization scheme $\omega$, let $\mathcal{E}_\omega = \gamma(1-\gamma)^{-2}\Delta^P_\omega + (1-\gamma)^{-1}\Delta^R_\omega$. For any confidence level $\delta > 0$ and the desired accuracy level $\epsilon \in (0,1)$, with probability at least $1 - \delta$, the output Q-function $\widehat{Q}^*_\omega$ from Algorithm 2 after $T \geq \bar{c}_2 \log(\epsilon^{-1}(1-\gamma)^{-1})$ iterations satisfies $\|\widehat{Q}^*_\omega - Q^*\|_\infty \leq \epsilon + \mathcal{E}_\omega$ provided that the*

*total number of samples, denoted by $D_\omega$, satisfies*

$$D_\omega \geq \frac{\bar{c}_0 \left( \sum_{k \in [\kappa_p]} |\mathcal{X}[Z_k^P]| \right) \log \left( \bar{c}_1 |\mathcal{X}[\cup_{k \in [K_\omega]} Z_k^P]| \delta^{-1} \right)}{\epsilon^2 (1 - \gamma)^3}$$
$$+ \sum_{i \in [\kappa_r]} |\mathcal{X}[Z_i^R]|, \qquad (9)$$

*where $\kappa_p \in [0, K_\omega]$ and $\kappa_r \in [0, \ell_\omega]$ are problem-dependent parameters, and $\bar{c}_0$, $\bar{c}_1$ $\bar{c}_2$ are absolute constants.*

Now we discuss further implications of our results.

**Model Misspecification Bias.** The parameter $\mathcal{E}_\omega$ is called the model misspecification bias, which consists of two terms: $\gamma(1-\gamma)^{-2}\Delta_\omega^P$ and $(1-\gamma)^{-1}\Delta_\omega^R$, both of which are linearly dependent on the approximation errors of the transition kernel and reward function, respectively. This bias arises from the inaccuracies introduced by the factorization of the transition kernel and reward function. If the factorization were accurate, these approximation errors would vanish, and $\mathcal{E}_\omega$ would reduce to zero, as in the FMDP case, which is discussed below.

**Sample Complexity Implications and Comparison to Prior Works.** For a fair comparison, we consider our results using a factorization scheme $\omega$ that is perfect with the bias $\mathcal{E}_\omega = 0$. In this case, our sample complexity required to achieve an $\varepsilon$-optimal policy is of the following order:

$$\widetilde{O}\left( \sum_{k \in [\kappa_p]} |\mathcal{X}[Z_k^P]| \epsilon^{-2} (1 - \gamma)^{-3} + \sum_{i \in [\kappa_r]} |\mathcal{X}[Z_i^R]| \right).$$

Compared with the minimax-optimal sample complexity of $\widetilde{O}(|\mathcal{S}||\mathcal{A}|\varepsilon^{-2}(1-\gamma)^{-3})$ (Azar et al., 2012) for solving standard MDPs, we have identical dependence on $\delta$, $\epsilon$, and the effective horizon $1/(1-\gamma)$. The key improvement lies in the dependence on the size of the state-action space, where we improve the minimax $\widetilde{\mathcal{O}}(|\mathcal{S}||\mathcal{A}|)$ dependence (Azar et al., 2012) to $\widetilde{\mathcal{O}}(\sum_{k \in [\kappa_p]} |\mathcal{X}[Z_k^P]|) \leq \widetilde{\mathcal{O}}(K_\omega \max_k |\mathcal{X}[Z_k^P]|)$. Notably, $\widetilde{\mathcal{O}}(K_\omega \max_k |\mathcal{X}[Z_k^P]|)$ is a problem-dependent sample complexity — almost proportional to the sample complexity of solving the largest individual component among the factorized parts of the entire transition kernel, which is exponentially smaller than $|\mathcal{S}||\mathcal{A}|$ due to reduced dimensionality. For example, consider an approximate factorization scheme that decomposes an MDP into 10 disjoint components with identical cardinalities. The corresponding sample complexity becomes $\widetilde{\mathcal{O}}([|\mathcal{S}||\mathcal{A}|]^{\frac{1}{10}})$, achieving an exponential reduction in sample complexity relative to the number of factorized components.

In addition, when the bias $\mathcal{E}_\omega = 0$, the MDPs with approximate factorization reduce to the well-known setting of FMDPs. Compared to the best-known sample complexity upper bound $\widetilde{\mathcal{O}}(\sum_{k=1}^{K_\omega} |\mathcal{X}[Z_k^P]| \epsilon^{-2}(1-\gamma)^{-3} +$

$\sum_{i=1}^{\ell_\omega} |\mathcal{X}[Z_i^R]|)$ for FMDPs from (Chen et al., 2020) [1], our result $\widetilde{\mathcal{O}}(\sum_{k=1}^{\kappa_p} |\mathcal{X}[Z_k^P]| \epsilon^{-2}(1-\gamma)^{-3} + \sum_{i=1}^{\kappa_r} |\mathcal{X}[Z_i^R]|)$ offers strictly better sample complexity, where $\kappa_p \in [K_\omega]$ and $\kappa_r \in [\ell_\omega]$ are instance-dependent parameters. Indeed, $\kappa_p$ reflects the sparsity of the dependence structure in the MDP. In many real-world applications such as UAV swarm control (Campion et al., 2018) and power system economic dispatch (Chen et al., 2022; Lu et al., 2023), the dynamics of different UAVs or different power generators are highly independent, then $\kappa_p \ll K_\omega$ and can be small constant if the dependence structure is sparse. Also, by connecting it to the *Graph Coloring Problem*, $\kappa_p \ll K_\omega$ is provable under mild conditions (can refer to the discussion in Section 4.2). As a result, it indicates our sample complexity can improve the prior arts in FMDPs by a factor of up to $\mathcal{O}(K_\omega)$. Similarly, $\kappa_r$ can be significantly smaller than $\ell_\omega$ if the reward function components have such sparse dependence.

The sample complexity lower bound for FMDPs is $\widetilde{\mathcal{O}}(\max_k |\mathcal{X}[Z_k^P]| \epsilon^{-2}(1-\gamma)^{-3} + \max_i |\mathcal{X}[Z_i^R]|)$, which is established in (Xu & Tewari, 2020; Chen et al., 2020). It worth noting that, our algorithm is the first to match this lower bound in an instance-dependent manner when $\kappa_p, \kappa_r = \mathcal{O}(1)$, demonstrating that we not only improve upon existing upper bounds but also achieve the theoretical minimum sample complexity under certain conditions.

**Trade-Off Between Sample Complexity and Model Misspecification Bias.** The approximate factorization scheme $\omega$ can be viewed as a tunable hyperparameter. In general, when the number of components $K_\omega$ increases, we get a finer decomposition of $P$ and the size of each component decays exponentially with $K_\omega$, which significantly reduces the sample complexity $D_\omega$. However, the drawback of increasing $K_\omega$ arbitrarily is that it may result in a larger misspecification bias $\mathcal{E}_\omega$ due to model mismatch. To illustrate this trade-off, consider the example depicted in Figure 1, where we decompose an MDP into three components by disregarding certain weak dependencies. If the induced error $\mathcal{E}_\omega$ is smaller than the desired accuracy level $\epsilon$, this imperfect factorization is sufficient to achieve the target solution. When we require higher accuracy (smaller $\varepsilon$), a more careful factorization scheme with generally fewer components should be chosen. For instance, grouping $w_t$ and $p_t$ within a single component will reduce the bias $\mathcal{E}_\omega$, though at the cost of increased sample requirements. This trade-off will be revisited again in our numerical simulations in Section H.1. As a final note, when $K_\omega = 1$, this theorem recovers the classical sample complexity result (without decomposition) established in (Azar et al., 2012).

---

[1]We translate their results to our setting for easy understanding and clear comparison.

# 6. Model-Free RL with Approximate Factorization

In this section, we focus on the model-free setting and introduce the *Variance-Reduced Q-Learning with Approximate Factorization* (VRQL-AF) algorithm with sample complexity guarantees.

## 6.1. Algorithm Design

In classical Q-learning (Watkins & Dayan, 1992), the Q-function estimate $\widehat{Q}_t$ is updated using a small-stepsize variant of the empirical Q-value iteration as follows:

$$\widehat{Q}_{t+1} = \widehat{Q}_t + \eta_t(\widehat{\mathcal{H}}_t(\widehat{Q}_t) - \widehat{Q}_t),$$

where $\eta_t \in (0, 1)$ is the learning rate, $\widehat{\mathcal{H}}_t(Q)$ is the empirical Bellman operator satisfying $[\widehat{\mathcal{H}}_t(Q)](s, a) = r(s, a) + \gamma \max_{a'} Q(s'_t(s, a), a')$ for all $(s, a)$ and $Q \in \mathbb{R}^{|\mathcal{S}||\mathcal{A}|}$, and $s'_t(s, a)$ denotes the next state sampled from $P(\cdot \mid s, a)$.

Compared with classical Q-learning, our VRQL-AF algorithm consists of two key modifications: (1) a factored empirical Bellman operator design and (2) a variance-reduced Q-iteration. The factored Bellman operator is designed to replace the standard empirical Bellman operator in order to enhance sampling efficiency by leveraging the structure. Its construction is outlined in Algorithm 3. Specifically, in vanilla Q-learning, we need to sample all state-action pairs to generate a single empirical Bellman operator, which requires $|\mathcal{S}||\mathcal{A}|$ samples and is highly inefficient. Our approach aims to reduce the number of required samples by exploiting the problem structure through a factorization scheme, thus enabling the repeated use of the same samples.

---

**Algorithm 3** Empirical Factored Bellman Operator Generation

---

1: **Input:** Factorization scheme $\omega = (\omega^P, \omega^R)$.
2: Solve the optimal Synchronous sampling scheme represented by $\{\kappa_p, \{\mathcal{G}_k\}_{k \in [\kappa_p]}\}$;
3: Construct the sampling sets $\{X^P_{\mathcal{G}_k}\}_{k \in [\kappa_p]}$;
4: **for** $k = 1, 2, \cdots, \kappa_p$ **do**
5:     Sample the transition from each state-action pair $x \in X^P_{\mathcal{G}_k}$ once, and obtain the next state by $\{s^k_x\}_{k \in [\kappa_p]}$.
6: **end for**
7: Get empirical Bellman operator $\widehat{\mathcal{H}}$ (for any $Q$) following Eq. (10);
8: **Output:** Empirical Bellman operator $\widehat{\mathcal{H}}$;

---

Specifically, we adopt the synchronous sampling scheme $\{\kappa_p, \{\mathcal{G}_k\}_{k \in [\kappa_p]}\}$ discussed in Section 4, and then obtain the corresponding $\kappa_p$ sampling sets $\{X^P_{\mathcal{G}_k}\}_{k \in [\kappa_p]}$. For each $k \in [\kappa_p]$, we sample over each entry $x$ from the set $\mathcal{X}^P_{\mathcal{G}_k}$ once, obtaining the next state $s^k_x$ randomly generated based on transition and corresponding reward. This gives us the

---

**Algorithm 4** Variance-Reduced Q-Learning with Approximate Factorization (VRQL-AF)

---

**Input**: Number of epochs $T$; Epoch length $M$; Reference sample size $N_\tau$ ($\tau \leq T$); Learning rate $\eta_t$.
**Output**: $\epsilon$-accurate Q Function estimation $\overline{Q}_M$ with $1 - \delta$ probability;
1: Initialize $\overline{Q}_0(s, a) = 0$ for all $s$ and $a$;
2: **for** epoch $\tau = 1, 2, ..., T$ **do**
3:     Generate $N_\tau$ factored empirical Bellman operators $\{\widehat{\mathcal{B}}_i\}_{i=1,2,...,N_\tau}$ though Algorithm 3;
4:     Calculate the reference Bellman operator $\overline{\mathcal{H}}_\tau(\overline{Q}_{\tau-1}) = \frac{1}{N_\tau} \sum_{i=1}^{N_\tau} \widehat{\mathcal{B}}_i(\overline{Q}_{\tau-1})$;
5:     Initialize $Q_0 = \overline{Q}_{\tau-1}$;
6:     **for** iteration $t = 0, ..., M - 1$ **do**
7:        Generate factored empirical Bellman operator $\widehat{\mathcal{H}}_t$ through Algorithm 3;
8:        Compute the variance-reduced update:
$$Q_{t+1} = Q_t + \eta_t\big(\widehat{\mathcal{H}}_t(Q_t) - \widehat{\mathcal{H}}_t(\overline{Q}_{\tau-1}) + \overline{\mathcal{H}}_\tau(\overline{Q}_{\tau-1}) - Q_t\big). \quad (11)$$
9:     **end for**
10:    $\overline{Q}_\tau = Q_M$;
11: **end for**
12: **Output:** $\overline{Q}_T$;

---

samples $\{s^k_x\}_{x \in \mathcal{X}^P_{\mathcal{G}_k}, k \in [\kappa_p]}$. These samples are sufficient to construct the factored empirical Bellman operator $\widehat{\mathcal{H}}(Q)$. Specifically, for any $x := (s, a) \in \mathcal{S} \times \mathcal{A}$, the corresponding entry $[\widehat{\mathcal{H}}(Q)](s, a)$ of $\widehat{\mathcal{H}}(Q)$ satisfies

$$[\widehat{\mathcal{H}}(Q)](s, a) = \gamma \max_{a'} Q(s'_{x,1}[Z^S_1], \ldots, s'_{x,K_\omega}[Z^S_{K_\omega}], a') + r(s, a), \quad (10)$$

where for each $j \in [K_\omega]$, $s'_{x,j}$ is the sample containing the correct transition of component $j$ satisfying

$$s'_{x,j} \in \{s^k_{\tilde{x}} \mid \tilde{x} \in \mathcal{X}^P_{\mathcal{G}_k}, \tilde{x}[Z^P_j] = x[Z^P_j]\}.$$

This ensures that the samples from different components are used to compute the transitions for all dimensions of the Bellman operator. By using synchronous sampling, we cover the transitions for all components, allowing us to estimate the entire factored Bellman operator efficiently. Note that, Eq. (10) simply extracts the relevant dimensions from each sample to update the corresponding factors, which makes the implementation straightforward and efficient.

The variance reduction technique aims to reduce the variance in the update process, thereby accelerating the convergence (Sidford et al., 2018; 2023; Wainwright, 2019b). Armed with both schemes, our proposed VRQL-AF is summarized in Algorithm 4. Specifically, the algorithm employs

a two-loop structure. In each outer epoch $\tau$, a new reference is constructed, followed by an inner loop that iteratively updates the Q-value estimate using both the reference and real-time empirical Bellman operators.

In particular, in each outer loop $\tau$, we first compute a *reference Bellman operator* $\overline{\mathcal{H}}(\overline{Q}_{\tau-1})$, which is an average of $N_\tau$ factored empirical Bellman operators $\widehat{\mathcal{B}}_i$ generated through a sampling process applied to the fixed Q-value function $\overline{Q}_{\tau-1}$ (obtained from previous steps). This reference Bellman operator provides a low-variance estimate of the Q-value function.

Next, in each inner loop $t$, a new factored empirical Bellman operator $\widehat{\mathcal{H}}_t$ is generated from fresh samples, and the Q-function is updated by combining a high-variance unbiased operator $\widehat{\mathcal{H}}_t(Q_t)$ with a reference-based variance-reduction term $\widetilde{\mathcal{H}}_t(\overline{Q}_{\tau-1}) - \overline{\mathcal{H}}_\tau(\overline{Q}_{\tau-1})$.

### 6.2. Sample Complexity Guarantees

We now provide a bound on the sample complexity of Algorithm 4; the proof is provided in Appendix E:

**Theorem 6.1.** *Given any approximate factorization scheme $\omega$, a confidence level $\delta > 0$, and the desired accuracy level $\epsilon \in (0,1)$. Let the bias $\mathcal{E}_\omega = \gamma(1-\gamma)^{-2}\Delta_\omega^P + (1-\gamma)^{-1}\Delta_\omega^R$, the number of epochs $T = c_1 \log\left((1-\gamma)^{-1}\epsilon^{-1}\right)$, the epoch length $M = c_2 \log\left(6T|\mathcal{X}[\cup_{k=1}^{K_\omega} Z_k^P]|(1-\gamma)^{-1}\delta^{-1}\right)(1-\gamma)^{-3}$, the reference sample size $N_\tau = c_3 4^\tau \log\left(6T|\mathcal{X}[\cup_{k=1}^{K_\omega} Z_k^P]|\right)(1-\gamma)^{-2}$, and the learning rate $\eta_t = (1 + (1-\gamma)(t+1))^{-1}$. With probability at least $1 - \delta$, the output Q-function $\widehat{Q}_\omega^*$ from Algorithm VRQL-AF satisfies $\|Q^* - \widehat{Q}_\omega^*\|_\infty \leq \epsilon + \mathcal{E}_\omega$, provided that the total number of samples, denoted by $D_\omega$, satisfies the following:*

$$D_\omega \geq \frac{c_0 (\sum_{k\in[\kappa_p]} |\mathcal{X}[Z_k^P]|) \log\left(\frac{c_1|\mathcal{X}[\cup_{k=1}^{K_\omega} Z_k^P]|}{(1-\gamma)\delta}\right) \log\left(\frac{c_2}{(1-\gamma)\epsilon}\right)}{\epsilon^2(1-\gamma)^3}$$
$$+ \sum_{i\in[\kappa_r]} |\mathcal{X}[Z_i^R]|,$$

*where $c_0, c_1, c_2 > 0$ are universal constants.*

In evaluating the performance of VRQL-AF, we note that the estimation bias $\mathcal{E}_\omega$ is the same as in our model-based algorithm in Theorem 5.1.

Therefore, we focus our discussion on the sample complexity, in the case of perfect factorization $\mathcal{E}_\omega = 0$ for fair comparisons. In this case, our sample complexity required to achieve an $\varepsilon$-optimal policy is of the following order:

$$\widetilde{O}\left(\frac{\sum_{k\in[\kappa_p]} |\mathcal{X}[Z_k^P]|}{\epsilon^2(1-\gamma)^3} + \sum_{i\in[\kappa_r]} |\mathcal{X}[Z_i^R]|\right), \quad (12)$$

which achieves the same order as our model-based algorithm. This indicates that our proposed model-free VRQL-AF can also address the persistent curse of dimensionality in standard RL, as already discussed for our model-based algorithm. For more details, please refer to Section 5.2.

Importantly, VRQL-AF is the first near-optimal model-free algorithm for our approximate factorization framework (including FMDPs), which breaks the curse of dimensionality by combining a tailored factored empirical Bellman operator, a synchronous sampling scheme, and advanced variance reduction. The improved sample complexity is achieved by a refined cross-dimensional variance analysis on the convergence process with factored empirical Bellman operators.

Note that for the factored MDP setting (Osband & Van Roy, 2014) with perfect factorization, our result can potentially be extended to the Markov-sampling regime under a fixed behavior policy $\pi$, provided that $\pi$ induces, on each factor with a sampling set, a uniformly ergodic Markov chain. The analysis can following techniques like (Li et al., 2020b). A fully rigorous treatment of this extension is left to future work.

## 7. Conclusion

Our work introduces a framework for approximate factorization in MDPs to address the curse of dimensionality in large-scale RL problems. By decomposing MDPs into smaller factorized components, we propose a synchronous sampling method to optimize component-wise sampling and a model-based RL algorithm that leverages the factorization structure. Additionally, we design variance-reduced Q-learning algorithms with a factored empirical Bellman operator for efficient online learning. Our theoretical results show significant improvements in sample complexity, outperforming existing bounds for both vanilla and FMDPs.

## Acknowledgements

We sincerely thank the anonymous area chair and reviewers for their insightful feedback. We are grateful to Hongyu Yi for proofreading the paper and to Kishan Panaganti for helpful discussions.

This work was supported in part by Tsinghua University during C. Lu's visit to the California Institute of Technology. L. Shi acknowledges support from the Resnick Institute and the Computing, Data, and Society Postdoctoral Fellowship at Caltech. C. Wu's work was supported in part by the National Natural Science Foundation of China under Grant 72271213, the Shenzhen Science and Technology Program under Grant RCYX20221008092927070. A. Wierman's work was supported in part by NSF grants CNS-2146814, CPS-2136197, CNS-2106403, and NGSDI-2105648.

## Impact Statement

This paper presents work whose goal is to advance the field of Machine Learning. There are many potential societal consequences of our work, none which we feel must be specifically highlighted here.

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

# Appendices

## A. Related Work

Our work contributes to a few key literatures within the RL community. We discuss each in turn below.

**Finite-Sample Analysis for Model-Based Algorithms.** Our proposed algorithm (cf. Algorithm 2) follows the model-based RL approach, where the learning process involves model estimation and planning. Model-based approaches have been extensively studied (Azar et al., 2012; Agarwal et al., 2020; Gheshlaghi Azar et al., 2013; Sidford et al., 2018; Azar et al., 2017; Jin et al., 2020; Li et al., 2024b), achieving minimax-optimal sample complexity of $\widetilde{\mathcal{O}}(|\mathcal{S}||\mathcal{A}|\epsilon^{-2}(1-\gamma)^{-3})$ in the generative model setting (Li et al., 2020a). This minimax-optimal bound is established by considering all possible MDPs in a worst-case manner, without leveraging any additional structure in the problem. In contrast, by leveraging the structure for algorithm design, we achieve sample complexity with exponentially reduced dependency on the size of the state and action space (cf. Theorem 5.1) when the MDP is perfectly factorizable.

**Finite-Sample Analysis for Model-Free Algorithms.** Our proposed algorithm also aligns with model-free RL, which does not estimate the model but directly optimize the policy (Sutton & Barto, 2018). A vast body of literature focuses on Q-learning (Tsitsiklis, 1994; Jaakkola et al., 1993; Szepesvári, 1997; Kearns & Singh, 1998; Even-Dar et al., 2003; Wainwright, 2019a; Chen et al., 2024a; Li et al., 2023; 2024a; Shi et al., 2022; Woo et al., 2024) with various sampling settings, demonstrating a minimax sample complexity of $\widetilde{\mathcal{O}}(|\mathcal{S}||\mathcal{A}|\epsilon^{-2}(1-\gamma)^{-4})$. With further advancements like variance reduction, Q-learning has been shown to achieve a minimax-optimal sample complexity of $\widetilde{\mathcal{O}}(|\mathcal{S}||\mathcal{A}|\epsilon^{-2}(1-\gamma)^{-3})$ in the generative model setting (Wainwright, 2019b). In contrast, our work leverages the approximate factorization structure of MDPs to further enhance sample efficiency. By designing a factored empirical Bellman operator with variance reduction, we achieve exponentially reduced sample complexity with respect to the state-action space size (cf. Theorem 6.1) with matching minimax dependence on the other parameters.

**Factored MDPs.** Our model generalizes the framework of FMDPs (Boutilier et al., 1995; 1999), extending it to account for approximation errors. Most existing work on FMDPs is set in an episodic framework and primarily analyzes regret performance (Guestrin et al., 2003; Osband & Van Roy, 2014; Xu & Tewari, 2020; Tian et al., 2020; Chen et al., 2020). In particular, the state-of-the-art results translate into a sample complexity of $\widetilde{\mathcal{O}}(\sum_{k=1}^{K}|\mathcal{X}_k|\epsilon^{-2}(1-\gamma)^{-3})$, where the complexity scales with the sum of the state-action space sizes $|\mathcal{X}_k|$ across all factored components. Building on this line of work, we propose a factorized synchronous sampling technique that enables simultaneous updates for multiple components using a single sample. By coupling this with refined cross-component variance analysis, we reduce the sample complexity to as low as $\widetilde{\mathcal{O}}\left(\max_k |\mathcal{X}_k|\epsilon^{-2}(1-\gamma)^{-3}\right)$ in an instance-dependent manner, which only depends on the maximal component size rather than the sum. This result matches the lower bounds established in prior work (Xu & Tewari, 2020; Chen et al., 2020), up to logarithmic factors. Crucially, our approach does not require the MDP to exhibit perfect factorizability, allowing for broader applicability to general MDPs.

**RL with Function Approximation.** To make RL problem sample efficient for large-scale problems, a common approach is to employ function approximation (Sutton & Barto, 2018). Intuitively, the key idea is to limit the searching space of an RL problem to a predefined function class, in which each function can be specified with a parameter that is low-dimensional. This approach has achieved significant empirical success (Mnih et al., 2015; Silver et al., 2017). However, RL with function approximation is not theoretically well understood except under strong structural assumptions on the approximating function class, such as the function class being linear (Tsitsiklis & Van Roy, 1996; Bhandari et al., 2018; Srikant & Ying, 2019; Chen et al., 2023; 2024b), the Bellman completeness being satisfied (Fan et al., 2020), or others (Dai et al., 2018; Wang et al., 2020). Also, the function approximation often targets to approximate the Q-values, instead of exploiting the inherent transition kernel and reward function structures. In this work, we take a different approach by leveraging approximate factorization structures instead of implementing function approximation. It is also worth noting that our approach is highly flexible and can be further extended by incorporating function approximation techniques, providing an even broader framework for tackling large-scale RL problems.

## B. An Illustrative Example of Approximate Factorization

To provide intuition, we present an example of applying the approximate factorization scheme in a real-world application, demonstrating that it offers additional opportunities to achieve a better balance between desired solution accuracy and sample efficiency. In particular, we consider the storage control problem (Xu et al., 2024) in wind farms. The storage controller (the agent) aims to align the real-time wind power generation output (the state) with the desired prediction values by flexibly charging or discharging the energy storage system.

This problem can be modeled as an MDP, where the state at time $t$, $s_t = (w_t, p_t, c_t)$, captures the wind power generation $w_t$, electricity price $p_t$, and the state of charge (SoC) of the storage $c_t$. The action $a_t$ represents the storage charging decision at time $t$. The transition dependence structure can be represented as a bipartite graph, as illustrated in Figure 1(a). In the graph, nodes on the left-hand side (LHS) represent the state and action at time $t$, while nodes on the right-hand side (RHS) represent the state at time $t + 1$. A solid blue line between nodes indicates a strong dependence. For instance, the electricity price $p_{t+1}$ strongly depends on the previous price $p_t$. While a dashed blue line represents weak dependence. For example, while the wind power generation $w_t$ can influence the next step's electricity price $p_{t+1}$ in the market, its impact is weaker compared to the direct influence of the previous price $p_t$.

Figure 1(b) illustrates an approximate factorization scheme on this MDP, where the system's dynamics are divided into three smaller components — the dynamics of the wind power generation, electricity price, and storage levels. Strong transition dependencies are preserved within each component, but weaker dependencies, such as the influence of wind power generation on electricity price, are disregarded. This leads to the following approximation of the transition probability:

$$\widehat{P}(s_{t+1}|s_t, a_t) = P(w_{t+1}|w_t) \cdot P(p_{t+1}|p_t) \cdot P(c_{t+1}|c_t, a_t).$$

This approximation simplifies the model by factoring the transition dynamics into smaller, more manageable components while retaining key dependencies. Note that this problem does not fall under a FMDP, as the dynamics cannot be perfectly broken down into smaller components. However, our approximate factorization scheme offers a more flexible framework, allowing imperfectly factorizable transition dynamics to be divided into smaller components while maintaining sufficient model accuracy. This expands the factorization step's action space, enabling a more effective search for optimal trade-offs between solution accuracy and sample efficiency.

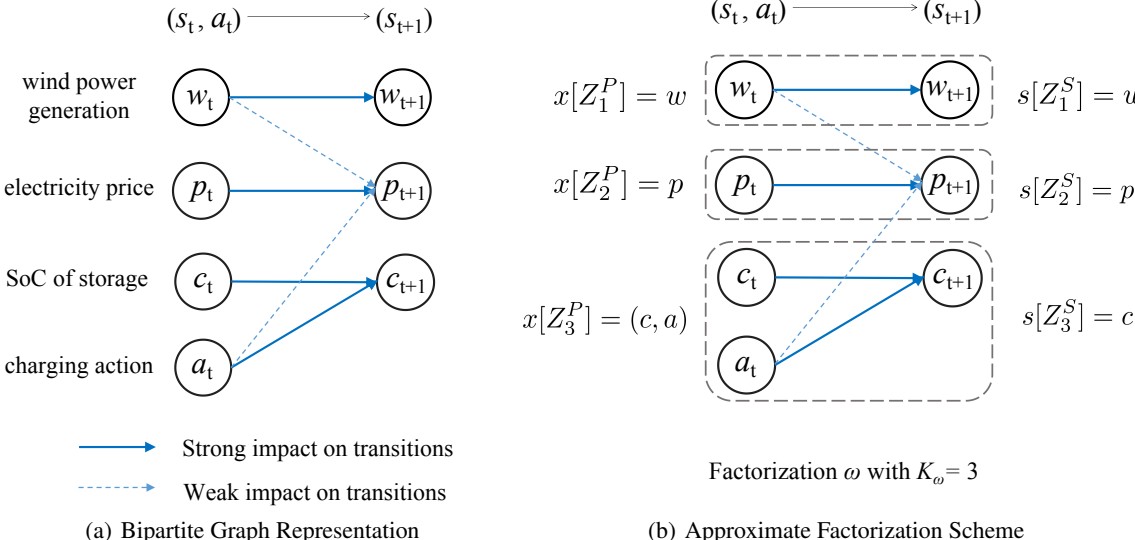

Figure 1. Bipartite Graph Representation and Approximate Factorization.

## C. Illustration of Synchronous Sampling with Exclusive Scopes

An example of synchronous sampling with exclusive scopes is illustrated in Fig. 2. Specifically, this factorization contains two components 1 and 2, the transition of state in the first component depends on $x[Z_1^P] = x[\{1, 2\}]$, while for the second

component, its associated state's transition depends on $x[Z_2^P] = x[\{3\}]$, where $Z_1^P \cap Z_2^P = \emptyset$, which means the transitions of the two components don't have common dependence. The classical sampling process within the FMDPs involves sampling all possible entries from the sampling sets of both components, leading to a sampling cost $|\mathcal{X}_1^P| + |\mathcal{X}_2^P|$. While equipped with the exclusive scope property, we can simultaneously sample the entries of both components in a single sample. Thus, the total sampling cost equals the size of the larger sampling set, i.e., $\max(|\mathcal{X}_1^P|, |\mathcal{X}_2^P|)$.

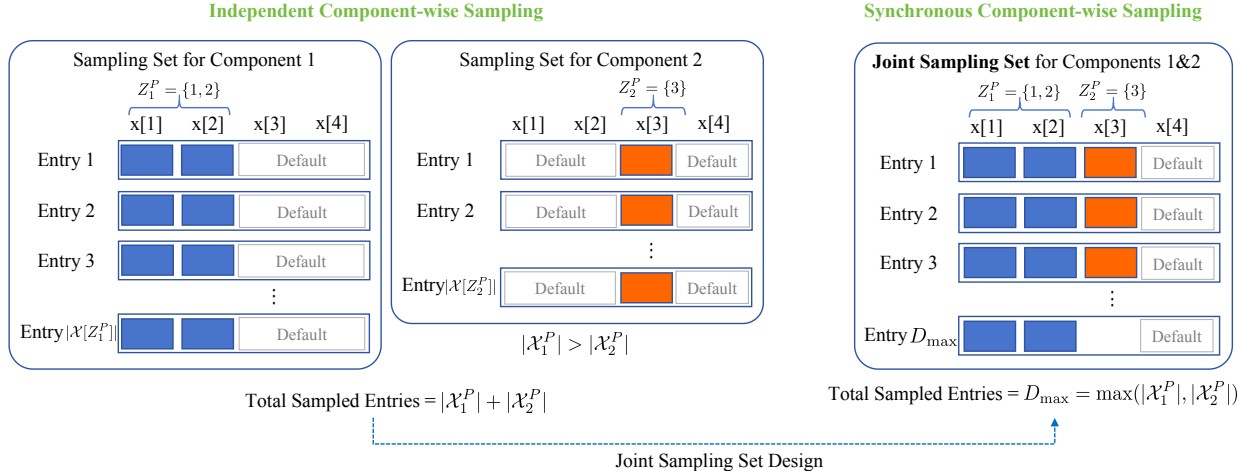

*Figure 2.* Synchronous Sampling with Exclusive Scopes.

# D. Proof of Theorem 5.1

We aim to ensure the total number of samples $D_\omega$ satisfies the following lower bound:

$$D_\omega \geq N_{\text{entry}} \cdot N + D_r, \tag{13}$$

where $N_{\text{entry}}$ denotes the number of unique state-action pairs that must be sampled, $N$ is the sampling frequency for each pair, and $D_r$ is the sample complexity required to estimate the exact reward function.

The term $N_{\text{entry}}$ is directly specified in Algorithm 1. It is defined as the total number of state-action pairs across all components to be sampled. Mathematically:

$$N_{\text{entry}} = \sum_{i \in [\kappa_p]} \max_{k \in \mathcal{G}_i} |\mathcal{X}[Z_k^P]| \leq \sum_{k \in [\kappa_p]} |\mathcal{X}[Z_k^P]|,$$

where $\kappa_p$ is the total number of sampling sets, $\mathcal{G}_i$ is the set of component indices associated with the $i$-th sampling set, $|\mathcal{X}[Z_k^P]|$ denotes the size of the state-action space for component $k$.

For the sample complexity associated with estimating the reward function, $D_r$, it is sufficient to sample all necessary state-action pairs once for each reward component $r_i$ to obtain the exact reward values. Similar to the transition kernel sampling, the sample complexity of the reward function is bounded by:

$$D_r \leq \max_{i \in [\kappa_r]} |\mathcal{X}[Z_i^R]|$$

where $\kappa_r$ is the minimal number of sampling sets required to estimate the reward function, and $|\mathcal{X}[Z_i^R]|$ is the size of the state-action space for reward component $r_i$.

Therefore, we only need to determine the required sampling frequency $N$. Recall that, the error $\|Q^* - \widehat{Q}_\omega^*\|_\infty$ comes from two aspects: 1) the computation error in Algorithm 2 due to finite value function iterations, and 2) finite sample error due to inaccurate estimation of $\widehat{P}$. Since the value iteration algorithm converges exponentially fast, according to Section 5.2 in (Shi et al., 2024), $T = \bar{c}_0 \log(\frac{1}{(1-\gamma)\epsilon})$ is enough to guarantee the computation error $\leq \mathcal{O}(\epsilon)$. Instead, we focus on the estimation error due to finite samples. To do so, we decompose the estimation error of the $Q$-value function into two terms, the bias $\mathcal{E}_\omega$

and the finite sample error $\alpha_N$. By applying the triangle inequality to the error, we get:

$$\|Q^* - \widehat{Q}_\omega^*\|_\infty \leq \underbrace{\|Q^* - Q_\omega^*\|_\infty}_{\text{Bias } \mathcal{E}_\omega} + \underbrace{\|Q_\omega^* - \widehat{Q}_\omega^*\|_\infty}_{\text{Finite Sample Error } \alpha_N}, \tag{14}$$

where

- $Q^*$ is the optimal $Q$-value function induced by the actual optimal policy $\pi^*$, transition kernel $P$, and reward function $r$.

- $\widehat{Q}_\omega^*$ is the estimated $Q$-value function based on the estimated policy $\widehat{\pi}_\omega^*$, estimated transition kernel $\widehat{P}_\omega$, and estimated reward function $r_\omega$, where

$$\widehat{P}_\omega(s'|x) = \prod_{k=1}^{K_\omega} \widehat{P}_k(s'[Z_k^S] \mid x[Z_k^P]), \forall s' \in \mathcal{S}, \forall x \in \mathcal{X},$$

$$r_\omega(x) = \sum_{i=1}^{\ell_\omega} r_i(x[Z_i^R]), \forall x \in \mathcal{X}.$$

- $Q_\omega^*$ is the optimal $Q$-value function based on the factorization scheme $\omega$, which assumes the utilization of infinite samples. It is induced by the policy $\pi_\omega$, transition kernel $P_\omega$, and reward function $r_\omega$, where $P_\omega(s'|x)$ is defined as:

$$P_\omega(s'|x) := \lim_{N \to \infty} \widehat{P}_\omega(s'|x).$$

The limits exists due to the fixed sampling algorithm and the law of large numbers. DUe to the deterministic mapping between $P$ and $Q$, we have $\lim_{N \to \infty} \widehat{Q}_\omega^* = Q_\omega^*$ with probability 1.

Thus, the finite sample error term $\alpha_N := \|Q_\omega^* - \widehat{Q}_\omega^*\|_\infty$ in the decomposition can be made arbitrarily small as $N$ increases. Instead, the bias $\mathcal{E}_\omega$ doesn't vanish due to the approximation errors from factorization.

For the bias term $\mathcal{E}_\omega$, it can be bounded as follows:

**Lemma D.1** (Proof in Appendix F.2). *Given any factorization scheme $\omega$, the following condition holds:*

$$\mathcal{E}_\omega := \|Q^* - Q_\omega^*\|_\infty \leq \frac{\Delta_\omega^R}{1 - \gamma} + \frac{\gamma \Delta_\omega^P}{(1 - \gamma)^2}. \tag{15}$$

Then, we focus on the finite sample error term $\alpha_N$. Using the result in Lemma F.4, we have:

$$Q_\omega^* - \widehat{Q}_\omega^* \leq \underbrace{\gamma(I - \gamma \widehat{P}_\omega^{\pi_\omega^*})^{-1}(P_\omega - \widehat{P}_\omega)V_\omega^*}_{:=\Delta_1}, \tag{16}$$

$$Q_\omega^* - \widehat{Q}_\omega^* \geq \underbrace{\gamma(I - \gamma \widehat{P}_\omega^{\widehat{\pi}_\omega^*})^{-1}(P_\omega - \widehat{P}_\omega)V_\omega^*}_{:=\Delta_2}, \tag{17}$$

where $\widehat{P}_\omega^{\pi_\omega^*} \in \mathbb{R}^{|\mathcal{S}||\mathcal{A}| \times |\mathcal{S}||\mathcal{A}|}$ represents the transition matrix of Markov chain $\{(s_t, a_t)\}$ induced by policy $\pi_\omega^*$ under transition kernel $\widehat{P}_\omega$, and $\widehat{P}_\omega^{\widehat{\pi}_\omega^*} \in \mathbb{R}^{|\mathcal{S}||\mathcal{A}| \times |\mathcal{S}||\mathcal{A}|}$ is induced by policy $\widehat{\pi}_\omega^*$ under transition kernel $\widehat{P}_\omega$.

Based on Eq. (16) and (17), the error term $\alpha_N$ satisfies

$$\alpha_N := \|Q_\omega^* - \widehat{Q}_\omega^*\|_\infty \leq \max\{\|\Delta_1\|_\infty, \|\Delta_2\|_\infty\}, \tag{18}$$

Now we only need to control $\|\Delta_1\|_\infty$ and $\|\Delta_2\|_\infty$. We first control the absolute value of the common term $(P_\omega - \widehat{P}_\omega)V_\omega^*$ for both $\Delta_1$ and $\Delta_2$ in Lemma D.2.

**Lemma D.2** (Proof in Appendix F.3). *Given sample size $N$, then with probability at least $1 - \delta$, the following condition holds:*

$$|(P_\omega - \widehat{P}_\omega)V_\omega^*| \leq \frac{2\log(2|\mathcal{X}[\cup_{k=1}^{K_\omega} Z_k^P]|)}{3N(1-\gamma)} \cdot \mathbf{1} + \sqrt{\frac{2\log(2|\mathcal{X}[\cup_{k=1}^{K_\omega} Z_k^P]|)Var_{P_\omega}(V_\omega^*)}{N}},$$

*where $|\cdot|$ denotes the absolute value function, $Var_{P_\omega}(V_\omega^*)$ denotes the value function variance with transition kernel $P_\omega$, as defined in Definition F.1.*

Using the naive bound $\|\gamma(I - \gamma\widehat{P}_\omega^\pi)^{-1}\|_\infty \leq \frac{\gamma}{1-\gamma}$ for any $\pi$, and combining Lemma D.2 with Eq. (16) and (17), we can show that

$$|\Delta_1| \leq \frac{2\gamma\log(2|\mathcal{X}[\cup_{k=1}^{K_\omega} Z_k^P]|)}{3N(1-\gamma)^2} \cdot \mathbf{1} + \gamma\sqrt{\frac{2\log(2|\mathcal{X}[\cup_{k=1}^{K_\omega} Z_k^P]|)}{N}}(I - \gamma\widehat{P}_\omega^{\pi_\omega^*})^{-1}\sqrt{Var_{P_\omega}(V_\omega^*)}, \tag{19}$$

$$|\Delta_2| \leq \frac{2\gamma\log(2|\mathcal{X}[\cup_{k=1}^{K_\omega} Z_k^P]|)}{3N(1-\gamma)^2} \cdot \mathbf{1} + \gamma\sqrt{\frac{2\log(2|\mathcal{X}[\cup_{k=1}^{K_\omega} Z_k^P]|)}{N}}(I - \gamma\widehat{P}_\omega^{\widehat{\pi}_\omega^*})^{-1}\sqrt{Var_{P_\omega}(V_\omega^*)}. \tag{20}$$

The remaining challenge is to control $(I - \gamma\widehat{P}_\omega^{\widehat{\pi}_\omega^*})^{-1}\sqrt{Var_{P_\omega}(V_\omega^*)}$ and $(I - \gamma\widehat{P}_\omega^{\pi_\omega^*})^{-1}\sqrt{Var_{P_\omega}(V_\omega^*)}$ for $\Delta_1$ and $\Delta_2$, respectively. We first bound the term $(I - \gamma\widehat{P}_\omega^{\widehat{\pi}_\omega^*})^{-1}\sqrt{Var_{P_\omega}(V_\omega^*)}$ for $\Delta_1$, which is slightly more complex, and bounding another term is analogous. Specifically,

$$|(I - \gamma\widehat{P}_\omega^{\widehat{\pi}_\omega^*})^{-1}\sqrt{Var_{P_\omega}(V_\omega^*)}|$$

$$= (I - \gamma\widehat{P}_\omega^{\widehat{\pi}_\omega^*})^{-1}\sqrt{Var_{P_\omega}(V_\omega^*) - Var_{\widehat{P}_\omega}(V_\omega^*) + Var_{\widehat{P}_\omega}(V_\omega^*)}$$

$$= (I - \gamma\widehat{P}_\omega^{\widehat{\pi}_\omega^*})^{-1}\sqrt{P_\omega(V_\omega^*)^2 - (P_\omega V^*)^2 - \widehat{P}_\omega(V_\omega^*)^2 + (\widehat{P}_\omega V_\omega^*)^2 + Var_{\widehat{P}_\omega}(V_\omega^*)}$$

$$= (I - \gamma\widehat{P}_\omega^{\widehat{\pi}_\omega^*})^{-1}\sqrt{(P_\omega - \widehat{P}_\omega)(V_\omega^*)^2 - ((P_\omega V_\omega^*)^2 - (\widehat{P}_\omega V_\omega^*)^2) + Var_{\widehat{P}_\omega}(V_\omega^*)}$$

$$\leq (I - \gamma\widehat{P}_\omega^{\widehat{\pi}_\omega^*})^{-1}\left(\sqrt{|(P_\omega - \widehat{P}_\omega)(V_\omega^*)^2|} + \sqrt{|(PV^*)^2 - (\widehat{P}_\omega V_\omega^*)^2|} + \sqrt{Var_{\widehat{P}_\omega}(V_\omega^*)}\right)$$

$$= \underbrace{(I - \gamma\widehat{P}_\omega^{\widehat{\pi}_\omega^*})^{-1}\sqrt{|(P_\omega - \widehat{P}_\omega)(V_\omega^*)^2|}}_{T_1} + \underbrace{(I - \gamma\widehat{P}_\omega^{\widehat{\pi}_\omega^*})^{-1}\sqrt{|(P_\omega V_\omega^*)^2 - (\widehat{P}_\omega V_\omega^*)^2|}}_{T_2}$$

$$+ \underbrace{(I - \gamma\widehat{P}_\omega^{\widehat{\pi}_\omega^*})^{-1}\sqrt{Var_{\widehat{P}_\omega}(V_\omega^*)}}_{T_3}.$$

This splition allows us to focus on $T_1$, $T_2$ and $T_3$, which can be controlled by the following lemmas:

**Lemma D.3** (Proof in Appendix F.4). *With a probability at least $1 - \delta$, $T_1$ satisfies*

$$T_1 \leq \frac{1}{(1-\gamma)^2}\sqrt[4]{\frac{2\log\left(2|\mathcal{X}[\cup_{k=1}^{K_\omega} Z_k^P]|\right)}{N}} \cdot \mathbf{1}.$$

**Lemma D.4** (Proof in Appendix F.5). *With a probability at least $1 - \delta$, $T_2$ satisfies*

$$T_2 \leq \frac{\sqrt{2}}{(1-\gamma)^2}\sqrt[4]{\frac{2\log\left(2|\mathcal{X}[\cup_{k=1}^{K_\omega} Z_k^P]|\right)}{N}} \cdot \mathbf{1}.$$

**Lemma D.5** (Proof in Appendix F.6). *The term $T_3$ can be bounded as follows:*

$$T_3 \leq \left(\frac{4\gamma}{(1-\gamma)^3}\sqrt{\frac{2\log\left(4|\mathcal{X}[\cup_{k=1}^{K_\omega} Z_k^P]|\right)}{N}} + \sqrt{\frac{1+\gamma}{(1-\gamma)^3}}\right) \cdot \mathbf{1}.$$

Combining Lemmas D.2-D.5 yields the bound of $\Delta_2$:

**Lemma D.6** (Proof in Appendix F.7). *With probability at least $1 - \delta$, the estimated Q-function $\widehat{Q}^*$ satisfies*

$$|\Delta_2| \leq 18 \left( \frac{\log \left( 12 |\mathcal{X}[\cup_{k=1}^{K_\omega} Z_k^P]| \right)}{N(1-\gamma)^3} \right) + 6 \left( \frac{\log \left( 12 |\mathcal{X}[\cup_{k=1}^{K_\omega} Z_k^P]| \right)}{N(1-\gamma)^3} \right)^{\frac{1}{2}}.$$

Analogously, we can show that with probability at least $1 - \delta$, $|\Delta_1|$ also satisfies

$$|\Delta_1| \leq 18 \left( \frac{\log \left( 12 |\mathcal{X}[\cup_{k=1}^{K_\omega} Z_k^P]| \right)}{N(1-\gamma)^3} \right) + 6 \left( \frac{\log \left( 12 |\mathcal{X}[\cup_{k=1}^{K_\omega} Z_k^P]| \right)}{N(1-\gamma)^3} \right)^{\frac{1}{2}}.$$

Combining the bounds on $|\Delta_1|$ and $|\Delta_2|$, we have that with probability at least $1 - \delta$,

$$\begin{aligned}
\|Q_\omega^* - \widehat{Q}_\omega^*\|_\infty &\leq \max(|\Delta_1|, |\Delta_2|) \\
&\leq 18 \left( \frac{\log \left( 24 |\mathcal{X}[\cup_{k=1}^{K_\omega} Z_k^P]| \right)}{N(1-\gamma)^3} \right) + 6 \left( \frac{\log \left( 24 |\mathcal{X}[\cup_{k=1}^{K_\omega} Z_k^P]| \right)}{N(1-\gamma)^3} \right)^{\frac{1}{2}}.
\end{aligned}$$

Taking any $\epsilon = \|Q_\omega^* - \widehat{Q}_\omega^*\|_\infty \leq 1$, we can first verify that $\frac{\log \left( 24 |\mathcal{X}[\cup_{k=1}^{K_\omega} Z_k^P]| \right)}{N(1-\gamma)^3} \leq 1$. Therefore,

$$\epsilon \leq (18 + 6) \left( \frac{\log \left( 24 |\mathcal{X}[\cup_{k=1}^{K_\omega} Z_k^P]| \right)}{N(1-\gamma)^3} \right)^{\frac{1}{2}}.$$

It directly yields the bound for $N$ with probability at least $1 - \delta$:

$$N \geq \frac{576 \log \left( 24 |\mathcal{X}[\cup_{k=1}^{K_\omega} Z_k^P]| \right)}{(1-\gamma)^3 \epsilon^2}. \tag{21}$$

Substituting Eq. (21) into Eq. (13) yields:

$$D_\omega \geq \frac{576 \left( \kappa_p \max_{k \in [K_\omega]} |\mathcal{X}[Z_k^P]| \right) \log \left( 24 |\mathcal{X}[\cup_{k \in [K_\omega]} Z_k^P]| \delta^{-1} \right)}{\epsilon^2 (1-\gamma)^3} + \kappa_r \max_{i \in [\ell_\omega]} |\mathcal{X}[Z_i^R]|.$$

Letting $\overline{c}_0 = 576$ and $\overline{c}_1 = 24$ leads to our result.

## E. Proof of Theorem 6.1

*Proof.* Similar to the model-based case, we show the total amount of samples $D_\omega$ should satisfy:

$$D_\omega \geq N_{\text{entry}} \cdot N + D_r, \tag{22}$$

where $N_{\text{entry}}$ denotes the number of sampled state-action pairs for generating a single empirical Bellman operator in Algorithm 3, and $N$ denotes the number of generated empirical Bellman operators. Notation $D_r$ denotes the sample complexity of finding the exact reward function.

The term $N_{\text{entry}}$ can be directly obtained in Algorithm 3, which equals the sum of numbers of all state-action pairs in all decomposed components, satisfying:

$$N_{\text{entry}} = \sum_{i \in [\kappa_p]} \max_{k \in \mathcal{G}_i} |\mathcal{X}[Z_k^P]| \leq \sum_{k \in [\kappa_p]} |\mathcal{X}[Z_k^P]|,$$

where $\kappa_p$ is the total number of sampling sets, $\mathcal{G}_i$ is the set of component indices associated with the $i$-th sampling set, $|\mathcal{X}[Z_k^P]|$ denotes the size of the state-action space for component $k$.

For the sample complexity associated with estimating the reward function, $D_r$, it is sufficient to sample all necessary state-action pairs once for each reward component $r_i$ to obtain the exact reward values. Similar to the transition kernel sampling, the sample complexity of the reward function is bounded by:

$$D_r \leq \sum\nolimits_{i \in [\kappa_r]} |\mathcal{X}[Z_i^R]|$$

where $\kappa_r$ is the minimal number of sampling sets required to estimate the reward function, and $|\mathcal{X}[Z_i^R]|$ is the size of the state-action space for reward component $r_i$.

For the sampling times $N$, according to Algorithm 4, we can conclude that:

$$N = \sum\nolimits_{\tau=1}^{T} (N_\tau + M) = \sum\nolimits_{\tau=1}^{T} N_\tau + MT, \tag{23}$$

where $T$ is the number of epochs, $N_\tau$ is the sampling frequency for estimating the reference Bellman operator in the $\tau$-th epoch, $M$ is the number of variance-reduced updates in a single epoch.

The key of the proof is to show the estimation error decays exponentially when the number of epochs increases, i.e.,

**Lemma E.1** (Proof in Appendix F.8). *For any $\delta \geq 0$, with probability at least $1 - \delta/T$, the Q-function estimate $\overline{Q}_\tau$ after $\tau$ epochs satisfies*

$$\|\overline{Q}_\tau - Q_\omega^*\|_\infty \leq \frac{1}{(1-\gamma)2^\tau}, \quad \forall \tau = 1, \dots, T, \tag{24}$$

*provided that the number of samples $M$ and the number of iterations $N_\tau$ satisfy the following conditions:*

$$M = c_2 \frac{\log\left(\frac{6T|\mathcal{X}[\cup_{k=1}^{K_\omega} Z_k^P]|}{(1-\gamma)\delta}\right)}{(1-\gamma)^3}, \tag{25}$$

$$N_\tau = c_3 4^\tau \frac{\log\left(6T|\mathcal{X}[\cup_{k=1}^{K_\omega} Z_k^P]|\right)}{(1-\gamma)^2}, \tag{26}$$

*where $\eta_t = \frac{1}{1+(1-\gamma)(t+1)}$, $c_2, c_3 > 0$ are sufficiently large constants.*

We first ensure the error decreases to a relatively small value $\frac{1}{\sqrt{1-\gamma}}$ after $T_1$ epochs. Specifically, we set:

$$T_1 = \left\lceil \log_2\left(\frac{1}{\sqrt{1-\gamma}}\right) \right\rceil. \tag{27}$$

Using the union bound over all $\tau \leq T_1$ in Eq. (24), we obtain that with probability at least $1 - \delta$:

$$\|\overline{Q}_{T_1} - Q^*\|_\infty \leq \frac{1}{\sqrt{1-\gamma}}.$$

Substituting $N_\tau$, $T_1$, and $M$ into Eq. (23), the total number of samples for this phase, denoted $N_{[1]}$, is bounded by:

$$N_{[1]} \leq c_4 \cdot \frac{\log\left(\frac{6T_1|\mathcal{X}[\cup_{k=1}^{K_\omega} Z_k^P]|}{(1-\gamma)\delta}\right) \cdot \log\left(\frac{1}{1-\gamma}\right)}{(1-\gamma)^3}, \tag{28}$$

where $c_4$ is a constant depending on $c_2$ and $c_3$.

We now show that from initial point $\overline{Q}_0 = \overline{Q}_{T_1}$, an additional $T_2 = \left\lceil c \log_2\left(\frac{1}{\sqrt{1-\gamma}\epsilon}\right) \right\rceil$ epochs are sufficient to reduce the error to $\epsilon$, where $c > 0$ is a constant.

We utilize the following lemma:

**Lemma E.2** (Proof in Appendix F.16). *Given* $\|\overline{Q}_0 - Q^*\|_\infty \leq \frac{1}{\sqrt{1-\gamma}}$, *and setting:*

$$M = c_2 \cdot \frac{\log\left(\frac{6T|\mathcal{X}[\cup_{k=1}^{K_\omega} Z_k^P]|}{(1-\gamma)\delta}\right)}{(1-\gamma)^3}, \tag{29}$$

$$N_\tau = c_3 \cdot 4^\tau \cdot \frac{\log\left(6T|\mathcal{X}[\cup_{k=1}^{K_\omega} Z_k^P]|\right)}{(1-\gamma)^2}, \tag{30}$$

*then, with probability at least* $1 - \delta$:

$$\|\overline{Q}_\tau - Q_\omega^*\|_\infty \leq \frac{1}{2^\tau \sqrt{1-\gamma}}, \quad \forall \tau \geq 0.$$

The proof of Lemma E.2 follows a similar routine to that of Lemma E.1 but leverages the better initial estimate $\overline{Q}_0$. This lemma indicates that starting from an initial error of $\frac{1}{\sqrt{1-\gamma}}$, the variance-reduced iteration halves the error at each step.

After $T_2 = \left\lceil c' \log_2\left(\frac{1}{\sqrt{1-\gamma}\epsilon}\right)\right\rceil$ epochs (with $c' > 0$), the error reduces to $\epsilon$. The total number of samples for this phase, denoted $N_{[2]}$, is bounded by:

$$N_{[2]} \leq c_5 \cdot \frac{\log\left(\frac{6T_1|\mathcal{X}[\cup_{k=1}^{K_\omega} Z_k^P]|}{(1-\gamma)\delta}\right) \cdot \log\left(\frac{1}{1-\gamma}\right)}{(1-\gamma)^3 \epsilon^2}, \tag{31}$$

where $c_5$ is a sufficiently large constant.

Combining the samples from both phases, the total number of samples required is:

$$N = N_{[1]} + N_{[2]}.$$

Substituting back into Eq. (22), we conclude that the total sample complexity $D_\omega$ satisfies the desired bound:

$$D_\omega \geq N_{\text{entry}} \cdot (N_{[1]} + N_{[2]}) + D_r.$$

This completes the proof.

$\square$

# F. Proof for Auxiliary Lemmas

## F.1. Preliminary Definitions and Lemmas

In the following, we provide all necessary definition, preliminary lemmas. For self-contained, we provide the proofs to the lemmas according to the notations of this work.

**Definition F.1.** Let $\text{Var}_P(V) \in \mathbb{R}^{|\mathcal{S}||\mathcal{A}|}$ be the value function variance with transition kernel $P$, which is given by:

$$\text{Var}_P(V) = P(V)^2 - (PV)^2,$$

where $P \in \mathbb{R}^{|\mathcal{S}||\mathcal{A}|\times|\mathcal{S}|}$ is the transition kernel, and $(V)^2 \in \mathbb{R}^{|\mathcal{S}|}$ is the element-wise product of $V$, i.e., $(V)^2 = V \circ V$.

**Definition F.2.** We define $\Sigma_M^\pi(s,a)$ as the variance of discounted reward under policy $\pi$ and MDP $M$ given state $s$ and action $a$, i.e.,

$$\Sigma_M^\pi(s,a) = \mathbb{E}\left[\left(\sum_{t=0}^\infty \gamma^t r(s_t, a_t) - Q_M^\pi(s_0, a_0)\right)^2 \middle| s_0 = s, a_0 = a\right],$$

where $Q_M^\pi$ denotes the Q-function induced by policy $\pi$ under MDP $M$.

**Lemma F.3.** *(Lemma 2.2 in (Agarwal et al., 2019)) For any policy $\pi$, it holds that:*

$$Q^\pi - \widehat{Q}^\pi = \gamma(I - \gamma\widehat{P}^\pi)^{-1}(P - \widehat{P})V^\pi,$$

*where $Q^\pi$ and $\widehat{Q}^\pi$ are Q-functions induced by the policy $\pi$ under transition kernels $P$ and $\widehat{P}$, respectively, and $\widehat{P}^\pi \in \mathbb{R}^{|\mathcal{S}||\mathcal{A}|\times|\mathcal{S}||\mathcal{A}|}$ represents the transition matrix of the Markov chain $\{(s_t, a_t)\}$ induced by $\pi$ with transition kernel $\widehat{P}$.*

*Proof.* Recall that $Q^\pi$ is the unique solution of the Bellman equation:

$$Q^\pi = r + \gamma P^\pi Q^\pi.$$

Since $I - \gamma P^\pi$ is invertible when $0 < \gamma < 1$, we have

$$Q^\pi = (I - \gamma P^\pi)^{-1}r. \tag{32}$$

It follows that

$$\begin{aligned}
Q^\pi - \widehat{Q}^\pi &= (I - \gamma P^\pi)^{-1}r - (I - \gamma\widehat{P}^\pi)^{-1}r \\
&= (I - \gamma\widehat{P}^\pi)^{-1}((I - \gamma\widehat{P}^\pi) - (I - \gamma P^\pi))(I - \gamma P^\pi)^{-1}r \\
&= \gamma(I - \gamma\widehat{P}^\pi)^{-1}(P^\pi - \widehat{P}^\pi)(I - \gamma P^\pi)^{-1}r \\
&= \gamma(I - \gamma\widehat{P}^\pi)^{-1}(P^\pi - \widehat{P}^\pi)Q^\pi \\
&= \gamma(I - \gamma\widehat{P}^\pi)^{-1}(P - \widehat{P})V^\pi,
\end{aligned}$$

where the fourth equality is due to Eq. (32). The last equality is because $P^\pi Q^\pi = PV^\pi$. $\qquad\square$

**Lemma F.4.** *(Lemma 2.5 in (Agarwal et al., 2019)) For any two optimal Q functions $Q^*$ and $\widehat{Q}^*$, which are induced by the same reward function $r$, but different transition kernels $P$ and $\widehat{P}$. It holds that:*

$$Q^* - \widehat{Q}^* \leq \gamma(I - \gamma\widehat{P}^{\pi^*})^{-1}(P - \widehat{P})V^*,$$
$$Q^* - \widehat{Q}^* \geq \gamma(I - \gamma\widehat{P}^{\widehat{\pi}^*})^{-1}(P - \widehat{P})V^*,$$

*where $\pi^*$ and $\widehat{\pi}^*$ are the optimal policies induced by Q-value functions $Q^*$ and $\widehat{Q}^*$. The matrix $\widehat{P}^{\pi^*} \in \mathbb{R}^{|\mathcal{S}||\mathcal{A}|\times|\mathcal{S}||\mathcal{A}|}$ represents the transition matrix of the Markov chain $\{(s_t, a_t)\}$ induced by the policy $\pi^*$ under the transition kernel $\widehat{P}$. Similarly, $\widehat{P}^{\widehat{\pi}^*} \in \mathbb{R}^{|\mathcal{S}||\mathcal{A}|\times|\mathcal{S}||\mathcal{A}|}$ represents the transition matrix of the Markov chain $\{(s_t, a_t)\}$ induced by the policy $\widehat{\pi}^*$ under the transition kernel $\widehat{P}$. The optimal value function $V^*$ is induced by $Q^*$, which satisfies $V^*(s) = \max_a Q^*(s, a)$ for all states $s \in \mathcal{S}$.*

*Proof.* The two conditions can be proved using Lemma F.3. Specifically, the first inequality can be proved as follows:

$$Q^* - \widehat{Q}^* = Q^{\pi^*} - \widehat{Q}^{\widehat{\pi}*} \leq Q^{\pi^*} - \widehat{Q}^{\pi^*} = \gamma(I - \gamma\widehat{P}^{\pi^*})^{-1}(P - \widehat{P})V^*,$$

where the inequality is because $\widehat{Q}^{\widehat{\pi}*} \geq \widehat{Q}^\pi$ for any policy $\pi$. The last equality comes from Lemma F.3.

For the second condition, we have:

$$\begin{aligned}
Q^* - \widehat{Q}^* &= Q^{\pi^*} - \widehat{Q}^{\widehat{\pi}^*} \\
&= Q^{\pi^*} - (I - \gamma\widehat{P}^{\widehat{\pi}^*})^{-1}r \\
&= (I - \gamma\widehat{P}^{\widehat{\pi}^*})^{-1}(I - \gamma\widehat{P}^{\widehat{\pi}^*})Q^{\pi^*} - (I - \gamma\widehat{P}^{\widehat{\pi}^*})^{-1}(I - \gamma P^{\pi^*})Q^{\pi^*} \\
&= (I - \gamma\widehat{P}^{\widehat{\pi}^*})^{-1}((I - \gamma\widehat{P}^{\widehat{\pi}^*}) - (I - \gamma P^{\pi^*}))Q^{\pi^*} \\
&= \gamma(I - \gamma\widehat{P}^{\widehat{\pi}^*})^{-1}(P^{\pi^*} - \widehat{P}^{\widehat{\pi}^*})Q^{\pi^*} \\
&\geq \gamma(I - \gamma\widehat{P}^{\widehat{\pi}^*})^{-1}(P^{\pi^*} - \widehat{P}^{\pi^*})Q^{\pi^*} \\
&= \gamma(I - \gamma\widehat{P}^{\widehat{\pi}^*})^{-1}(P - \widehat{P})V^*.
\end{aligned}$$

The second equality is because $Q^\pi = (I - \gamma P^\pi)^{-1}r$, and the inequality is because $\widehat{P}^{\widehat{\pi}^*}Q^{\pi^*} \leq \widehat{P}^{\pi^*}Q^{\pi^*}$ due to the optimality of policy $\pi^*$ regarding Q-function $Q^{\pi^*}$. $\qquad\square$

**Lemma F.5.** *(Adapted from Lemma 5 in (Azar et al., 2012)) Given a MDP $M$ with transition kernel $P$ and reward function $r$, the following identity holds for all policy $\pi$:*

$$\Sigma_M^\pi = \gamma^2 (1 - \gamma^2 P^\pi)^{-1} Var_P(V_M^\pi)$$

*Proof.* We start with the definition of $\Sigma_M^\pi(s, a)$:

$$\Sigma_M^\pi(s, a) = \mathbb{E}_{P,\pi}\left[\left(\sum_{t=0}^\infty \gamma^t r(s_t, a_t) - Q_M^\pi(s_0, a_0)\right)^2 \middle| s_0 = s, a_0 = a\right]$$

$$= \mathbb{E}_{P,\pi}\left[\left(\sum_{t=1}^\infty \gamma^t r(s_t, a_t) - \gamma Q_M^\pi(s_1, a_1) - (Q_M^\pi(s_0, a_0) - r(s_0, a_0) - \gamma Q_M^\pi(s_1, a_1))\right)^2 \middle| s_0 = s, a_0 = a\right]$$

$$= \mathbb{E}_{P,\pi}\left[\left(\sum_{t=1}^\infty \gamma^t r(s_t, a_t) - \gamma Q_M^\pi(s_1, a_1)\right)^2 \middle| s_0 = s, a_0 = a\right]$$

$$- 2\mathbb{E}_{P,\pi}\left[\left(\sum_{t=1}^\infty \gamma^t r(s_t, a_t) - \gamma Q_M^\pi(s_1, a_1)\right)(Q_M^\pi(s_0, a_0) - r(s_0, a_0) - \gamma Q_M^\pi(s_1, a_1)) \middle| s_0 = s, a_0 = a\right]$$

$$+ \mathbb{E}_{P,\pi}\left[(Q_M^\pi(s_0, a_0) - r(s_0, a_0) - \gamma Q_M^\pi(s_1, a_1))^2 \middle| s_0 = s, a_0 = a\right]$$

$$= \gamma^2 \mathbb{E}_{P,\pi}\left[\left(\sum_{t=1}^\infty \gamma^{t-1} r(s_t, a_t) - Q_M^\pi(s_1, a_1)\right)^2 \middle| s_0 = s, a_0 = a\right]$$

$$- 2\mathbb{E}_{P,\pi}\left[\mathbb{E}\left[\sum_{t=1}^\infty \gamma^t r(s_t, a_t) - \gamma Q_M^\pi(s_1, a_1) \middle| s_1, a_1\right](Q_M^\pi(s_0, a_0) - r(s_0, a_0) - \gamma Q_M^\pi(s_1, a_1)) \middle| s_0 = s, a_0 = a\right]$$

$$+ \mathbb{E}_{P,\pi}\left[(Q_M^\pi(s_0, a_0) - r(s_0, a_0) - \gamma Q_M^\pi(s_1, a_1))^2 \middle| s_0 = s, a_0 = a\right]$$

$$= \gamma^2 \mathbb{E}_{P,\pi}\left[\left(\sum_{t=1}^\infty \gamma^{t-1} r(s_t, a_t) - Q_M^\pi(s_1, a_1)\right)^2 \middle| s_0 = s, a_0 = a\right]$$

$$+ \mathbb{E}_{P,\pi}\left[(Q_M^\pi(s_0, a_0) - r(s_0, a_0) - \gamma Q_M^\pi(s_1, a_1))^2 \middle| s_0 = s, a_0 = a\right]$$

$$= \gamma^2 \mathbb{E}_{P,\pi}\left[\left(\sum_{t=1}^\infty \gamma^{t-1} r(s_t, a_t) - Q_M^\pi(s_1, a_1)\right)^2 \middle| s_0 = s, a_0 = a\right]$$

$$+ \gamma^2 \mathbb{E}_{P,\pi}\left[\left(\mathbb{E}_{s_1, a_1 \sim P(\cdot|s_0, a_0)}[Q_M^\pi(s_1, a_1)] - Q_M^\pi(s_1, a_1)\right)^2 \middle| s_0 = s, a_0 = a\right]$$

$$= \gamma^2 \sum_{s_1, a_1} P^\pi(s_1, a_1|s, a)\Sigma_M^\pi(s_1, a_1) + \gamma^2 Var_P(V_M^\pi)(s, a),$$

where the third equality is obtained by dividing the quadratic term; the fourth equality is derived by the law of total expectation; the fifth equality holds due to $\mathbb{E}\left[\sum_{t=1}^\infty \gamma^t r(s_t, a_t) - \gamma Q_M^\pi(s_1, a_1) \middle| s_1, a_1\right] = 0$. The last equality is derived based on the definitions of $\Sigma_M^\pi(s, a)$ and $Var_P(V_M^\pi)$. □

**Lemma F.6.** *(Lemma 6 in (Wainwright, 2019b)) Given two Q-functions $Q_r^*$ and $\tilde{Q}^*$, which are the induced by the same transition kernel $P$, but different reward functions $r$ and $\tilde{r} = r + \Delta r$. It holds that,*

$$|Q^* - \tilde{Q}^*| \le \max\left\{(I - \gamma P^{\pi^*})^{-1}|\Delta r|, (I - \gamma P^{\tilde{\pi}^*})^{-1}|\Delta r|\right\},$$

*where $\pi^*$ and $\tilde{\pi}^*$ denote the optimal policy induced by $Q^*$ and $\tilde{Q}^*$, respectively.*

*Proof.* This lemma can be proved by showing the following two conditions:

$$\max(Q^* - \tilde{Q}^*, \mathbf{0}) \le (I - \gamma P^{\pi^*})|\Delta r|, \tag{33}$$

$$\max(\tilde{Q}^* - Q^*, \mathbf{0}) \leq (I - \gamma P^{\tilde{\pi}^*})|\Delta r|. \tag{34}$$

For condition (33), we can prove the following:

$$\begin{aligned}
Q^* - \tilde{Q}^* &= r + \gamma P^{\pi^*} Q^* - (r + \Delta r + \gamma P^{\tilde{\pi}^*} \tilde{Q}^*) \\
&\leq |\Delta r| + \gamma P^{\pi^*}(Q^* - \tilde{Q}^*) \\
&\leq |\Delta r| + \gamma P^{\pi^*} \max(Q^* - \tilde{Q}^*, \mathbf{0}),
\end{aligned}$$

where the second inequality comes from $P^{\pi^*}\tilde{Q}^* \leq P^{\tilde{\pi}^*}\tilde{Q}^*$.

Since the right-hand-side term is positive in all entries. Thus, we have:

$$\max(Q^* - \tilde{Q}^*, \mathbf{0}) \leq |\Delta r| + \gamma P^{\pi^*} \max(Q^* - \tilde{Q}^*, \mathbf{0}).$$

Rearranging the inequality yields Eq. (33).

For proving Eq. (34), we follow a similar routine and get:

$$\begin{aligned}
\tilde{Q}^* - Q^* &= (r + \Delta r + \gamma P^{\tilde{\pi}^*} \tilde{Q}^*) - (r + \gamma P^{\pi^*} Q^*) \\
&\leq |\Delta r| + \gamma P^{\tilde{\pi}^*}(\tilde{Q}^* - Q^*) \\
&\leq |\Delta r| + \gamma P^{\tilde{\pi}^*} \max(\tilde{Q}^* - Q^*, \mathbf{0}).
\end{aligned}$$

Hence, we have:

$$\max(\tilde{Q}^* - Q^*, \mathbf{0}) \leq |\Delta r| + \gamma P^{\pi^*} \max(\tilde{Q}^* - Q^*, \mathbf{0}).$$

Rearranging the inequality yields Eq. (34). Using $|Q^* - \tilde{Q}^*| = \max(\max(Q^* - \tilde{Q}^*, \mathbf{0}), \max(\tilde{Q}^* - Q^*, \mathbf{0}))$ and combining Eq. (33)-(34) yields the desired result. $\qquad\square$

### F.2. Proof for Lemma D.1

*Proof.* We first write the Bellman equation of $Q^*$ and $Q_\omega^*$ as follows:

$$Q^*(s, a) = r(s, a) + \gamma \sum_{s'} P(s'|s, a) \max_{a'} Q^*(s', a'), \tag{35}$$

$$Q_\omega^*(s, a) = r_\omega(s, a) + \gamma \sum_{s'} P_\omega(s'|s, a) \max_{a'} Q_\omega^*(s', a'). \tag{36}$$

Subtracting Eq. (35) with (36) yields:

$$\begin{aligned}
Q^*(s, a) - Q_\omega^*(s, a) &= r(s, a) - r_\omega(s, a) + \gamma \sum_{s'} (P(s'|s, a) \max_{a'} Q^*(s', a') - P_\omega(s'|s, a) \max_{a'} Q_\omega^*(s', a')) \\
&= r(s, a) - r_\omega(s, a) + \gamma \sum_{s'} (P(s'|s, a)(\max_{a'} Q^*(s', a') - \max_{a'} Q_\omega^*(s', a')) \\
&\quad + (P(s'|s, a) - P_\omega(s'|s, a)) \max_{a'} Q_\omega^*(s', a'))
\end{aligned}$$

Taking absolute value on both side yields:

$$\begin{aligned}
|Q^*(s, a) - Q_\omega^*(s, a)| &\leq \|r - r_\omega\|_\infty + \gamma \sum_{s'} (P(s'|s, a)(\max_{a'} Q^*(s', a') - \max_{a'} Q_\omega^*(s', a')) \\
&\quad + \gamma \|P - P_\omega\|_\infty \max_{s', a'} Q_\omega^*(s', a') \\
&\leq \|r - r_\omega\|_\infty + \gamma \|Q^* - Q_\omega^*\|_\infty + \gamma \|P - P_\omega\|_\infty \max_{s', a'} Q_\omega^*(s', a'), \tag{37}
\end{aligned}$$

where the last inequality comes from $|\max_a Q^*(s,a) - \max_a Q^*_\omega(s,a)| \le \max_a |Q^*(s,a) - Q^*_\omega(s,a)| \le \|Q^* - Q^*_\omega\|_\infty$. Due to $Q^*_\omega(s',a') \le \frac{1}{1-\gamma}$ for any state-action pair $(s',a')$, we have:

$$\|Q^* - Q^*_\omega\|_\infty = \max_{s,a} |Q^*(s,a) - Q^*_\omega(s,a)|$$

$$\le \|r - r_\omega\|_\infty + \gamma\|Q^* - Q^*_\omega\|_\infty + \frac{\gamma}{1-\gamma}\|P - P_\omega\|_\infty. \tag{38}$$

Standard mathematical manipulation on Eq. (38) yields:

$$\|Q^* - Q^*_\omega\|_\infty \le \frac{\|r - r_\omega\|_\infty}{1-\gamma} + \frac{\gamma\|P - P_\omega\|_\infty}{(1-\gamma)^2}. \tag{39}$$

Applying the definitions of approximation errors, we have:

$$\|Q^* - Q^*_\omega\|_\infty \le \frac{\Delta^R_\omega}{1-\gamma} + \frac{\gamma\Delta^P_\omega}{(1-\gamma)^2}. \tag{40}$$

This concludes our proof. $\square$

### F.3. Proof for Lemma D.2

*Proof.* We leverage the structure of the factorized transition kernel and show that the vector $(P_\omega - \widehat{P}_\omega)V^*_\omega$ contains multiple identical entries. By identifying the distinct entries, we can focus our analysis on a subset of state-action pairs.

Recall that $(P_\omega - \widehat{P}_\omega)V^*_\omega \in \mathbb{R}^{|\mathcal{S}||\mathcal{A}| \times 1}$ represents the difference between the actual and estimated reward vector. Due to the factorized form of the transition kernel $P_\omega$, the estimated transition probability $\widehat{P}(s' \mid x)$ is given by:

$$\widehat{P}(s' \mid x) = \prod_{k=1}^{K_\omega} \widehat{P}_k(s'[Z^S_k] \mid x[Z^P_k]),$$

where each $\widehat{P}_k(s'[Z^S_k] \mid x[Z^P_k])$ depends only on the subset $Z^P_k$ of the state-action pair $x$. As a result, only the state-action components $x[Z^P_k]$ for $k \in [K_\omega]$ determine the transition probabilities. Thus, there are at most $|\mathcal{X}[\cup_{k=1}^{K_\omega} Z^P_k]|$ distinct rows in the matrix $(P_\omega - \widehat{P}_\omega)$.

We use $\mathcal{X}^*$ to denote a set of state-action pairs indicating distinct entries, satisfying:

$$\mathcal{X}^* = \left\{ x \in \mathcal{X} \,\middle|\, \forall x', x'' \in \mathcal{X}^*,\ x' \ne x'' \Rightarrow \exists\, k \in [K_\omega] \text{ such that } x'[Z^P_k] \ne x''[Z^P_k] \right\}. \tag{41}$$

For any $x \in \mathcal{X}^*$, we can easily verify that the estimation $\widehat{P}(s' \mid x)$ is unbiased:

$$\mathbb{E}[\widehat{P}(s' \mid x) - P(s' \mid x)]$$

$$=\mathbb{E}\left[\prod_{i \in [\kappa_p]}\left(\prod_{k \in \mathcal{G}_i} \widehat{P}_k(s'[Z^S_k] \mid x[Z^P_k])\right) - \prod_{i \in [\kappa_p]}\left(\prod_{k \in \mathcal{G}_i} P_k(s'[Z^S_k] \mid x[Z^P_k])\right)\right]$$

$$=\mathbb{E}\left[\prod_{i \in [\kappa_p]} \widehat{P}(s'[\cup_{k \in \mathcal{G}_i} Z^S_k] \mid x[\cup_{k \in \mathcal{G}_i} Z^P_k]) - \prod_{i \in [\kappa_p]} P(s'[\cup_{k \in \mathcal{G}_i} Z^S_k] \mid x[\cup_{k \in \mathcal{G}_i} Z^P_k])\right]$$

$$=\prod_{i \in [\kappa_p]} \mathbb{E}[\widehat{P}(s'[\cup_{k \in \mathcal{G}_i} Z^S_k] \mid x[\cup_{k \in \mathcal{G}_i} Z^P_k])] - \prod_{i \in [\kappa_p]} P(s'[\cup_{k \in \mathcal{G}_i} Z^S_k] \mid x[\cup_{k \in \mathcal{G}_i} Z^P_k])$$

$$=0.$$

The third equality is due to the independent sampling of different sampling sets, and the last equality is due to $\mathbb{E}[\widehat{P}(s'[\cup_{k \in \mathcal{G}_i} Z^S_k] \mid x[\cup_{k \in \mathcal{G}_i} Z^P_k])] = P(s'[\cup_{k \in \mathcal{G}_i} Z^S_k] \mid x[\cup_{k \in \mathcal{G}_i} Z^P_k])$ within each sampling set due to the law of large numbers.

Also, since $\mathbf{0} \leq V_\omega^* \leq \frac{1}{1-\gamma} \cdot \mathbf{1}$, we have $\|(P_\omega - \widehat{P}_\omega)V_\omega^*\|_\infty \leq \frac{1}{1-\gamma}$. Combining with the Bernstein's inequality (Vershynin, 2018) with sample size $N$ yields that:

$$P\left(|(P_\omega(\cdot|x) - \widehat{P}_\omega(\cdot|x))V_\omega^*| \geq t\right) \leq 2\exp\left(-\frac{\frac{N^2}{2}t^2}{N\mathrm{Var}_{P_\omega}(V_\omega^*)_{(x)} + N\frac{t}{3(1-\gamma)}}\right), \tag{42}$$

where $\mathrm{Var}_{P_\omega}(V_\omega^*)_{(x)}$ denotes the entry of $\mathrm{Var}_{P_\omega}(V_\omega^*)$ corresponding to $x$.

Letting the right-hand-side of Eq. (42) equal $\frac{\delta}{|\mathcal{X}[\cup_{k=1}^{K_\omega} Z_k^P]|}$ leads to the following inequality:

$$\frac{\delta}{|\mathcal{X}[\cup_{k=1}^{K_\omega} Z_k^P]|} = 2\exp\left(-\frac{\frac{N^2}{2}t^2}{N\mathrm{Var}_{P_\omega}(V_\omega^*)_{(x)} + N\frac{t}{3(1-\gamma)}}\right).$$

Rearranging this term yields that

$$t \leq \frac{2\log(|\mathcal{X}[\cup_{k=1}^{K_\omega} Z_k^P]|)}{3N(1-\gamma)} + \sqrt{\frac{2\log(|\mathcal{X}[\cup_{k=1}^{K_\omega} Z_k^P]|)\mathrm{Var}_{P_\omega}(V_\omega^*)_{(x)}}{N}}.$$

Taking the union bound across all state-action pairs $x \in \mathcal{X}^*$, and using the identical entry property yields that, with probability at least $1 - \delta$:

$$\left|(P_\omega - \widehat{P}_\omega)V^*\right| \leq \sqrt{\frac{2\log(|\mathcal{X}[\cup_{k=1}^{K_\omega} Z_k^P]|)\mathrm{Var}_{P_\omega}(V_\omega^*)}{N}} + \frac{2\log(|\mathcal{X}[\cup_{k=1}^{K_\omega} Z_k^P]|)}{3N(1-\gamma)} \cdot \mathbf{1}.$$

This concludes our proof. $\qquad\square$

### F.4. Proof for Lemma D.3

*Proof.* We have that

$$\|(P_\omega - \widehat{P}_\omega)(V_\omega^*)^2\|_\infty = \left\|\left(\prod_{k=1}^{K_\omega} P_k(s'[Z_k^S] \mid x[Z_k^P]) - \prod_{k=1}^{K_\omega} \widehat{P}_k(s'[Z_k^S] \mid x[Z_k^P])\right)(V_\omega^*)^2\right\|_\infty.$$

Following the same routine in the proof of Lemma D.2, we known $(P_\omega - \widehat{P}_\omega)(V_\omega^*)^2$ contains at most $|\mathcal{X}[\cup_{k=1}^{K_\omega} Z_k^P]|$ distinct entries.

Meanwhile, for each single entry of $(P_\omega - \widehat{P}_\omega)(V_\omega^*)^2$, denoted by $(P_\omega(\cdot|x) - \widehat{P}_\omega(\cdot|x))(V_\omega^*)^2$, we can show the following conditions:

$$\mathbb{E}((P_\omega(\cdot|x) - \widehat{P}_\omega(\cdot|x))(V_\omega^*)^2) = 0, \forall x \in \mathcal{X},$$

$$\|(P_\omega - \widehat{P}_\omega)(V_\omega^*)^2\|_\infty \leq \|\widehat{P}_\omega(V_\omega^*)^2\|_\infty + \|P_\omega(V_\omega^*)^2\|_\infty \leq \frac{2}{(1-\gamma)^2}.$$

With $N$ *i.i.d.* samples for estimating $(P_\omega(\cdot|x) - \widehat{P}_\omega(\cdot|x))(V_\omega^*)^2 \in \mathbb{R}$, we apply the standard Hoeffding's inequality to $(P_\omega(\cdot|x) - \widehat{P}_\omega(\cdot|x))(V_\omega^*)^2$ as follows:

$$P\left(|(P_\omega(\cdot|x) - \widehat{P}_\omega(\cdot|x))(V_\omega^*)^2| \geq \epsilon\right) \leq 2\exp\left(-\frac{2N\epsilon^2}{(\frac{2}{(1-\gamma)^2})^2 \cdot N}\right) = 2\exp\left(-\frac{(1-\gamma)^4 N\epsilon^2}{2}\right), \forall x \in \mathcal{X},$$

Letting the right-hand-side term be $\frac{\delta}{|\mathcal{X}[\cup_{k=1}^{K_\omega} Z_k^P]|}$ and applying the union bound across all distinct state-action pair $x \in \mathcal{X}^*$ (with definition in Eq. (41)) yield:

$$\|(P_\omega - \widehat{P}_\omega)(V_\omega^*)^2\|_\infty \leq \frac{1}{(1-\gamma)^2} \cdot \sqrt{\frac{2\log\left(2|\mathcal{X}[\cup_{k=1}^{K_\omega} Z_k^P]|\right)}{N}}.$$

Thus, $T_1$ satisfies

$$
\begin{aligned}
T_1 &= (I - \gamma \widehat{P}_\omega^{\widehat{\pi}_\omega^*})^{-1} \sqrt{|(P_\omega - \widehat{P}_\omega)(V_\omega^*)^2|} \\
&\leq \|(I - \gamma \widehat{P}_\omega^{\widehat{\pi}_\omega^*})^{-1}\|_\infty \sqrt{\left\|(P_\omega - \widehat{P}_\omega)(V_\omega^*)^2\right\|_\infty} \cdot \mathbf{1} \\
&\leq \frac{1}{(1-\gamma)^2} \sqrt[4]{\frac{2\log\left(2|\mathcal{X}[\cup_{k=1}^{K_\omega} Z_k^P]|\right)}{N}} \cdot \mathbf{1}.
\end{aligned}
$$

This concludes our proof. $\qquad\square$

### F.5. Proof for Lemma D.4

*Proof.* We first bound $\|(P_\omega V_\omega^*)^2 - (\widehat{P}_\omega V_\omega^*)^2\|_\infty$ as follows:

$$
\begin{aligned}
\|(P_\omega V_\omega^*)^2 - (\widehat{P}_\omega V_\omega^*)^2\|_\infty &= \|(P_\omega V_\omega^* + \widehat{P}_\omega V_\omega^*)(P_\omega V_\omega^* - \widehat{P}_\omega V_\omega^*)\|_\infty \\
&\leq \|P_\omega V_\omega^* + \widehat{P}_\omega V_\omega^*\|_\infty \|P_\omega V_\omega^* - \widehat{P}_\omega V_\omega^*\|_\infty \\
&\leq 2\|V_\omega^*\|_\infty \|(P_\omega - \widehat{P}_\omega)V_\omega^*\|_\infty.
\end{aligned}
$$

Applying the Hoeffding's inequality to $\|(P_\omega - \widehat{P}_\omega)V_\omega^*\|_\infty$ yields that, with probability at least $1 - \delta$:

$$
\|(P_\omega - \widehat{P}_\omega)V_\omega^*\|_\infty \leq \frac{1}{1-\gamma} \sqrt{\frac{2\log\left(2|\mathcal{X}[\cup_{k=1}^{K_\omega} Z_k^P]|\right)}{N}}. \tag{43}
$$

Therefore, $T_2$ satisfies

$$
\begin{aligned}
T_2 &= (I - \gamma \widehat{P}_\omega^{\widehat{\pi}_\omega^*})^{-1} \sqrt{|(P_\omega V_\omega^*)^2 - (\widehat{P}_\omega V_\omega^*)^2|} \\
&\leq \|(I - \gamma \widehat{P}_\omega^{\widehat{\pi}_\omega^*})^{-1}\|_\infty \sqrt{\left\|(P_\omega V_\omega^*)^2 - (\widehat{P}_\omega V_\omega^*)^2\right\|_\infty} \cdot \mathbf{1} \\
&\leq \frac{1}{1-\gamma} \sqrt{2\|V_\omega^*\|_\infty \|(P_\omega - \widehat{P}_\omega)V_\omega^*\|_\infty} \cdot \mathbf{1} \\
&\leq \frac{\sqrt{2}}{(1-\gamma)^2} \sqrt[4]{\frac{2\log\left(2|\mathcal{X}[\cup_{k=1}^{K_\omega} Z_k^P]|\right)}{N}} \cdot \mathbf{1}.
\end{aligned}
$$

$\qquad\square$

### F.6. Proof for Lemma D.5

*Proof.* We decompose $T_3$ as follows:

$$
\begin{aligned}
T_3 &= (I - \gamma \widehat{P}_\omega^{\widehat{\pi}_\omega^*})^{-1} \sqrt{\mathrm{Var}_{\widehat{P}_\omega}(V_\omega^*)} \\
&= (I - \gamma \widehat{P}_\omega^{\widehat{\pi}_\omega^*})^{-1} \sqrt{\mathrm{Var}_{\widehat{P}_\omega}(V_\omega^* - \widehat{V}_\omega^{\pi_\omega^*} + \widehat{V}_\omega^{\pi_\omega^*} - \widehat{V}_\omega^{\widehat{\pi}_\omega^*} + \widehat{V}_\omega^{\widehat{\pi}_\omega^*})} \\
&\leq (I - \gamma \widehat{P}_\omega^{\widehat{\pi}_\omega^*})^{-1} \sqrt{2\mathrm{Var}_{\widehat{P}_\omega}(V_\omega^* - \widehat{V}_\omega^{\pi_\omega^*}) + 2\mathrm{Var}_{\widehat{P}_\omega}(\widehat{V}_\omega^{\widehat{\pi}_\omega^*}) + 2\mathrm{Var}_{\widehat{P}_\omega}(\widehat{V}_\omega^{\widehat{\pi}_\omega^*} - \widehat{V}_\omega^{\pi_\omega^*})} \\
&\leq \underbrace{(I - \gamma \widehat{P}_\omega^{\widehat{\pi}_\omega^*})^{-1} \sqrt{2\mathrm{Var}_{\widehat{P}_\omega}(V_\omega^* - \widehat{V}_\omega^{\pi_\omega^*})}}_{T_{31}} + \underbrace{(I - \gamma \widehat{P}_\omega^{\widehat{\pi}_\omega^*})^{-1} \sqrt{2\mathrm{Var}_{\widehat{P}_\omega}(\widehat{V}_\omega^{\widehat{\pi}_\omega^*})}}_{T_{32}} \\
&\quad + \underbrace{(I - \gamma \widehat{P}_\omega^{\widehat{\pi}_\omega^*})^{-1} \sqrt{2\mathrm{Var}_{\widehat{P}_\omega}(\widehat{V}_\omega^{\widehat{\pi}_\omega^*} - \widehat{V}_\omega^{\pi_\omega^*})}}_{T_{33}}.
\end{aligned}
$$

Then, we bound $T_{31}$, $T_{32}$ and $T_{33}$ separately.

**Step 1. Bounding $T_{31}$:**

For $T_{31}$, the following condition holds:

$$
\begin{aligned}
T_{31} &= (I - \gamma \widehat{P}_\omega^{\widehat{\pi}_\omega^*})^{-1} \sqrt{2 \mathrm{Var}_{\widehat{P}_\omega}(V_\omega^* - \widehat{V}_\omega^{\pi_\omega^*})} \\
&\leq \|(I - \gamma \widehat{P}_\omega^{\widehat{\pi}_\omega^*})^{-1}\|_\infty \sqrt{2 \|\mathrm{Var}_{\widehat{P}_\omega}(V_\omega^* - \widehat{V}_\omega^{\pi_\omega^*})\|_\infty} \cdot \mathbf{1} \\
&\leq \frac{\sqrt{2}}{1 - \gamma} \sqrt{\|V_\omega^* - \widehat{V}_\omega^{\pi_\omega^*}\|_\infty^2} \cdot \mathbf{1} \\
&\leq \frac{\sqrt{2}}{1 - \gamma} \sqrt{\|Q_\omega^* - \widehat{Q}_\omega^{\pi_\omega^*}\|_\infty^2} \cdot \mathbf{1}.
\end{aligned}
$$

Applying Lemma F.3 yields:

$$
\begin{aligned}
\|Q_\omega^* - \widehat{Q}_\omega^{\pi_\omega^*}\|_\infty^2 &= \|\gamma(I - \gamma \widehat{P}_\omega^{\pi_\omega^*})^{-1}(P_\omega - \widehat{P}_\omega)V_\omega^*\|_\infty^2 \\
&\leq \frac{\gamma^2}{(1 - \gamma)^2} \|(P_\omega - \widehat{P}_\omega)V_\omega^*\|_\infty^2.
\end{aligned}
$$

Applying Eq. (43), we have that, with probability at least $1 - \delta$, the term $\|Q_\omega^* - \widehat{Q}_\omega^{\pi_\omega^*}\|_\infty^2$ satisfies

$$
\begin{aligned}
\|Q_\omega^* - \widehat{Q}_\omega^{\pi_\omega^*}\|_\infty^2 &\leq \frac{\gamma^2}{(1 - \gamma)^4} \left( \sqrt{\frac{2 \log\left(2|\mathcal{X}[\cup_{k=1}^{K_\omega} Z_k^P]|\right)}{N}} \right)^2 \\
&= \frac{2\gamma^2 \log\left(2|\mathcal{X}[\cup_{k=1}^{K_\omega} Z_k^P]|\right)}{N(1 - \gamma)^4}.
\end{aligned} \tag{44}
$$

Therefore, with probability at least $1 - \delta$, $T_{31}$ satisfies that:

$$
T_{31} \leq \frac{2\gamma}{(1 - \gamma)^3} \sqrt{\frac{2 \log\left(2|\mathcal{X}[\cup_{k=1}^{K_\omega} Z_k^P]|\right)}{N}} \cdot \mathbf{1}.
$$

$\square$

**Step 2. Bounding $T_{32}$:**

Note that, $(1 - \gamma)(I - \gamma \widehat{P}_\omega^{\widehat{\pi}_\omega^*})^{-1}$ is a matrix of probability with each row being a probability distribution. For a positive vector $\boldsymbol{v}$ and distribution $\nu$, Jensen's inequality implies that $\nu\sqrt{\boldsymbol{v}} \leq \sqrt{\nu \cdot \boldsymbol{v}}$ (inequality of expectation). This implies:

$$
\begin{aligned}
\|T_{32}\|_\infty &= \left\| (I - \gamma \widehat{P}_\omega^{\widehat{\pi}_\omega^*})^{-1} \sqrt{\mathrm{Var}_{\widehat{P}_\omega}(\widehat{V}_\omega^{\widehat{\pi}_\omega^*})} \right\|_\infty \\
&= \frac{1}{1 - \gamma} \left\| (1 - \gamma)(I - \gamma \widehat{P}_\omega^{\widehat{\pi}_\omega^*})^{-1} \sqrt{\mathrm{Var}_{\widehat{P}_\omega}(\widehat{V}_\omega^{\widehat{\pi}_\omega^*})} \right\|_\infty \\
&\leq \sqrt{\left\| \frac{1}{1 - \gamma}(I - \gamma \widehat{P}_\omega^{\widehat{\pi}_\omega^*})^{-1} \mathrm{Var}_{\widehat{P}_\omega}(\widehat{V}_\omega^{\widehat{\pi}_\omega^*}) \right\|_\infty}.
\end{aligned} \tag{45}
$$

Also, we can reformulate $\|(I - \gamma \widehat{P}_\omega^{\widehat{\pi}_\omega^*})^{-1} \mathrm{Var}_{\widehat{P}}(\widehat{V}_\omega^{\widehat{\pi}_\omega^*})\|_\infty$ as follows:

$$
\|(I - \gamma \widehat{P}_\omega^{\widehat{\pi}_\omega^*})^{-1} \mathrm{Var}_{\widehat{P}_\omega}(\widehat{V}_\omega^{\widehat{\pi}_\omega^*})\|_\infty = \|(I - \gamma \widehat{P}_\omega^{\widehat{\pi}_\omega^*})^{-1}(I - \gamma^2 \widehat{P}_\omega^{\widehat{\pi}_\omega^*})(I - \gamma^2 \widehat{P}_\omega^{\widehat{\pi}_\omega^*})^{-1} \mathrm{Var}_{\widehat{P}_\omega}(\widehat{V}_\omega^{\widehat{\pi}_\omega^*})\|_\infty
$$

$$= \|(I - \gamma \widehat{P}_\omega^{\widehat{\pi}_\omega^*})^{-1}(I - \gamma \widehat{P}_\omega^{\widehat{\pi}_\omega^*})(I + \gamma \widehat{P}_\omega^{\widehat{\pi}_\omega^*})(I - \gamma^2 \widehat{P}_\omega^{\widehat{\pi}_\omega^*})^{-1}\mathrm{Var}_{\widehat{P}_\omega}(\widehat{V}_\omega^{\widehat{\pi}_\omega^*})\|_\infty$$

$$= \|(I + \gamma \widehat{P}_\omega^{\widehat{\pi}_\omega^*})(I - \gamma^2 \widehat{P}_\omega^{\widehat{\pi}_\omega^*})^{-1}\mathrm{Var}_{\widehat{P}_\omega}(\widehat{V}_\omega^{\widehat{\pi}_\omega^*})\|_\infty$$

$$\leq \|(I + \gamma \widehat{P}_\omega^{\widehat{\pi}_\omega^*})\|_\infty \|(I - \gamma^2 \widehat{P}_\omega^{\widehat{\pi}_\omega^*})^{-1}\mathrm{Var}_{\widehat{P}_\omega}(\widehat{V}_\omega^{\widehat{\pi}_\omega^*})\|_\infty$$

$$\leq (1 + \gamma)\|(I - \gamma^2 \widehat{P}_\omega^{\widehat{\pi}_\omega^*})^{-1}\mathrm{Var}_{\widehat{P}_\omega}(\widehat{V}_\omega^{\widehat{\pi}_\omega^*})\|_\infty.$$

Thus, we have

$$\left\| (I - \gamma \widehat{P}^\pi)^{-1}\sqrt{\mathrm{Var}_{\widehat{P}_\omega}(\widehat{V}_\omega^{\widehat{\pi}_\omega^*})} \right\|_\infty \leq \sqrt{\left\| \frac{1 + \gamma}{1 - \gamma}(I - \gamma^2 \widehat{P}_\omega^{\widehat{\pi}_\omega^*})^{-1}\mathrm{Var}_{\widehat{P}_\omega}(\widehat{V}_\omega^{\widehat{\pi}_\omega^*}) \right\|_\infty}.$$

We now connect this result with the definition of $\Sigma_M^\pi$ in Lemma F.5:

$$\Sigma_M^{\widehat{\pi}_\omega^*} = \gamma^2(1 - \gamma^2 \widehat{P}_\omega^{\widehat{\pi}_\omega^*})^{-1}\mathrm{Var}_{\widehat{P}_\omega}(\widehat{V}_\omega^{\widehat{\pi}_\omega^*}),$$

where the transition kernel of MDP $M$ is $\widehat{P}_\omega$. Therefore,

$$(I - \gamma^2 \widehat{P}_\omega^{\widehat{\pi}_\omega^*})^{-1}\mathrm{Var}_{\widehat{P}_\omega}(\widehat{V}_\omega^{\widehat{\pi}_\omega^*}) = \frac{\Sigma_M^{\widehat{\pi}_\omega^*}}{\gamma^2} \leq \frac{1}{(1 - \gamma)^2} \cdot \mathbf{1},$$

where the inequality comes from $\|\Sigma_M^\pi\|_\infty \leq \frac{\gamma^2}{(1-\gamma)^2}$ for any policy $\pi$, which can be easily verified according to Definition F.2:

$$\|\Sigma_M^\pi\|_\infty = \max_{(s,a)} \left\| \mathbb{E}\left[ \left( \sum_{t=0}^\infty \gamma^t r(s_t, a_t) - Q_M^\pi(s_0, a_0) \right)^2 \Bigg| s_0 = s, a_0 = a \right] \right\|_\infty$$

$$\leq \max_{(s,a)} \mathbb{E}\left[ \left\| \sum_{t=0}^\infty \gamma^t r(s_t, a_t) - Q_M^\pi(s_0, a_0) \right\|_\infty^2 \Bigg| s_0 = s, a_0 = a \right]$$

$$\leq \frac{\gamma^2}{(1 - \gamma)^2}.$$

Substituting this condition into Eq. (45) yields:

$$T_{32} \leq \sqrt{\frac{1 + \gamma}{(1 - \gamma)^3}} \cdot \mathbf{1}.$$

**Step 3. Bounding $T_{33}$:** The term $T_{33}$ satisfies

$$T_{33} = (I - \gamma \widehat{P}_\omega^{\widehat{\pi}_\omega^*})^{-1}\sqrt{2\mathrm{Var}_{\widehat{P}_\omega}(\widehat{V}_\omega^{\widehat{\pi}_\omega^*} - \widehat{V}_\omega^{\pi_\omega^*})}$$

$$\leq \|(I - \gamma \widehat{P}_\omega^{\widehat{\pi}_\omega^*})^{-1}\|_\infty \sqrt{2\|\mathrm{Var}_{\widehat{P}_\omega}(\widehat{V}_\omega^{\widehat{\pi}_\omega^*} - \widehat{V}_\omega^{\pi_\omega^*})\|_\infty} \cdot \mathbf{1}$$

$$\leq \frac{\sqrt{2}}{1 - \gamma}\sqrt{\|\widehat{V}_\omega^{\widehat{\pi}_\omega^*} - \widehat{V}_\omega^{\pi_\omega^*}\|_\infty^2} \cdot \mathbf{1}$$

$$\leq \frac{\sqrt{2}}{1 - \gamma}\sqrt{\|\widehat{Q}_\omega^* - \widehat{Q}_\omega^{\pi_\omega^*}\|_\infty^2} \cdot \mathbf{1}.$$

Similar to the bound of $T_{31}$ in Eq. (44), we have that with probability at least $1 - \delta$,

$$\|\widehat{Q}_\omega^* - \widehat{Q}_\omega^{\pi_\omega^*}\|_\infty^2 \leq \frac{2\gamma^2 \log\left( 2|\mathcal{X}[\cup_{k=1}^{K_\omega} Z_k^P]| \right)}{N(1 - \gamma)^4}.$$

Thus, with probability at least $1 - \delta$, $T_{33}$ satisfies

$$T_{33} \leq \frac{2\gamma}{(1-\gamma)^3} \sqrt{\frac{2 \log \left(2|\mathcal{X}[\cup_{k=1}^{K_\omega} Z_k^P]|\right)}{N}} \cdot \mathbf{1}.$$

**Step 4. Combining the Results:**

Combining the upper bounds for $T_{31}$, $T_{32}$ and $T_{33}$, we can bound $T_3$ that, with probability at least $1 - \delta$,

$$T_3 \leq \left( \frac{4\gamma}{(1-\gamma)^3} \sqrt{\frac{2 \log \left(4|\mathcal{X}[\cup_{k=1}^{K_\omega} Z_k^P]|\right)}{N}} + \sqrt{\frac{1+\gamma}{(1-\gamma)^3}} \right) \cdot \mathbf{1}.$$

### F.7. Proof for Lemma D.6

*Proof.* Taking $\delta$ to be $\frac{\delta}{3}$, and applying Lemma D.2-D.5 yields:

$$|\Delta_2| \leq \frac{3}{(1-\gamma)^3} \left( \frac{\log \left(12|\mathcal{X}[\cup_{k=1}^{K_\omega} Z_k^P]|\right)}{N} \right) + \frac{7}{(1-\gamma)^2} \left( \frac{\log \left(12|\mathcal{X}[\cup_{k=1}^{K_\omega} Z_k^P]|\right)}{N} \right)^{\frac{3}{4}}$$
$$+ \sqrt{\frac{2}{(1-\gamma)^3}} \left( \frac{\log \left(12|\mathcal{X}[\cup_{k=1}^{K_\omega} Z_k^P]|\right)}{N} \right)^{1/2}. \tag{46}$$

Applying Cauchy–Schwarz inequality yields:

$$\frac{3}{(1-\gamma)^3} \left( \frac{\log \left(12|\mathcal{X}[\cup_{k=1}^{K_\omega} Z_k^P]|\right)}{N} \right) + \sqrt{\frac{2}{(1-\gamma)^3}} \left( \frac{\log \left(12|\mathcal{X}[\cup_{k=1}^{K_\omega} Z_k^P]|\right)}{N} \right)^{1/2}$$
$$\geq \sqrt{\frac{3}{(1-\gamma)^3} \left( \frac{\log \left(12|\mathcal{X}[\cup_{k=1}^{K_\omega} Z_k^P]|\right)}{N} \right) \cdot \sqrt{\frac{2}{(1-\gamma)^3}} \left( \frac{\log \left(12|\mathcal{X}[\cup_{k=1}^{K_\omega} Z_k^P]|\right)}{N} \right)^{1/2}}$$
$$= \sqrt{3}(1-\gamma)^{-\frac{9}{4}} \left( \frac{\log \left(12|\mathcal{X}[\cup_{k=1}^{K_\omega} Z_k^P]|\right)}{N} \right)^{\frac{3}{4}}$$
$$\geq \frac{\sqrt{3}}{(1-\gamma)^2} \left( \frac{\log \left(12|\mathcal{X}[\cup_{k=1}^{K_\omega} Z_k^P]|\right)}{N} \right)^{\frac{3}{4}}.$$

Applying this condition to Eq. (46) yields:

$$\|Q_\omega^* - \widehat{Q}_\omega^*\|_\infty \leq \frac{3(1+\frac{7}{\sqrt{3}})}{(1-\gamma)^3} \left( \frac{\log \left(12|\mathcal{X}[\cup_{k=1}^{K_\omega} Z_k^P]|\right)}{N} \right) + \frac{\sqrt{2}(1+\frac{7}{\sqrt{3}})}{\sqrt{(1-\gamma)^3}} \left( \frac{\log \left(12|\mathcal{X}[\cup_{k=1}^{K_\omega} Z_k^P]|\right)}{N} \right)^{1/2}$$
$$\leq 18 \left( \frac{\log \left(12|\mathcal{X}[\cup_{k=1}^{K_\omega} Z_k^P]|\right)}{N(1-\gamma)^3} \right) + 6 \left( \frac{\log \left(12|\mathcal{X}[\cup_{k=1}^{K_\omega} Z_k^P]|\right)}{N(1-\gamma)^3} \right)^{\frac{1}{2}}.$$

This concludes our proof.

$\square$

## F.8. Proof for Lemma E.1

We prove this lemma by induction. We first show that the base case ($\tau = 1$) satisfies Eq. (24), and then prove the inductive condition when $\tau \geq 2$.

**Step 1. Showing Base Case with $\tau = 1$:**

Due to the initialization, we have $\overline{Q}_{\tau-1}(s, a) = \overline{Q}_0(s, a) = 0$ for any state-action pair $(s, a)$. Therefore, for $\tau = 1$, both the empirical and reference Bellman operators equal to the immediate reward:

$$\widehat{\mathcal{H}}_t(\overline{Q}_{\tau-1})_{s,a} = r(s, a) \quad \text{and} \quad \overline{\mathcal{H}}_\tau(\overline{Q}_{\tau-1})_{s,a} = r(s, a), \quad \forall (s, a).$$

Given this, the variance-reduced update in Eq. (11) simplifies to the standard Q-learning update:

$$Q_t = (1 - \eta_{t-1})Q_{t-1} + \eta_{t-1}\widehat{\mathcal{H}}_{t-1}(Q_{t-1}), \quad \forall t = 1, \ldots, M.$$

Let $\Delta_t = Q_t - Q_\omega^*$ denote the estimation error at iteration $t$, where $Q_\omega^*$ is the unique fixed-point solution of the Bellman equation $\mathcal{H}(Q) = Q$. Then, we have:

$$\begin{aligned}
\Delta_t &= (1 - \eta_{t-1})\Delta_{t-1} + \eta_{t-1}(\widehat{\mathcal{H}}_{t-1}(Q_\omega^* + \Delta_{t-1}) - \mathcal{H}(Q_\omega^*)) \\
&= (1 - \eta_{t-1})\Delta_{t-1} + \eta_{t-1}(\widehat{\mathcal{H}}_{t-1}(Q_\omega^* + \Delta_{t-1}) - \widehat{\mathcal{H}}_{t-1}(Q_\omega^*) + \widehat{\mathcal{H}}_{t-1}(Q_\omega^*) - \mathcal{H}(Q_\omega^*)) \\
&= (1 - \eta_{t-1})\Delta_{t-1} + \eta_{t-1}\underbrace{(\widehat{\mathcal{H}}_{t-1}(Q_\omega^* + \Delta_{t-1}) - \widehat{\mathcal{H}}_{t-1}(Q_\omega^*))}_{\text{Contractive Error: } \mathcal{W}_{t-1}(\Delta_{t-1})} + \eta_{t-1}\underbrace{(\widehat{\mathcal{H}}_{t-1}(Q_\omega^*) - \mathcal{H}(Q_\omega^*))}_{\text{Random Error: } \mathcal{E}_{t-1}}.
\end{aligned} \quad (47)$$

Observe that the error iteration $\Delta_t$ consists of three components: the decaying term $(1 - \eta_{t-1})\Delta_{t-1}$, the contractive error term $\mathcal{W}_{t-1}(\Delta_{t-1})$, and the random error $\mathcal{E}_{t-1}$.

The contractive error $\mathcal{W}_{t-1}(\Delta_{t-1})$ depends on both $\Delta_{t-1}$ and the empirical Bellman operator $\widehat{\mathcal{H}}_{t-1}$, and it is bounded due to the $\gamma$-contractiveness of $\widehat{\mathcal{H}}_{t-1}$:

$$\|\mathcal{W}_{t-1}(\Delta_{t-1})\|_\infty \leq \gamma\|\Delta_{t-1}\|_\infty. \quad (48)$$

The random error $\mathcal{E}_{t-1}$ is *i.i.d.* for different values of $t$. By applying the iteration in Eq. (47) and using the contraction condition in Eq. (48), we can express $\Delta_t$ fully as follows:

**Lemma F.7** (Proof in Appendix F.9). *For any $t \geq 1$, the estimation error $\Delta_t$ is bounded above and below by:*

$$\Delta_t \leq \prod_{k=0}^{t-1}(1 - (1-\gamma)\eta_k)\|\Delta_0\|_\infty \mathbf{1} + \gamma\eta_{t-1}\|P_{t-1}\|_\infty \mathbf{1} + \gamma\sum_{i=1}^{t-2}\left(\left(\prod_{j=i+1}^{t-1}(1 - (1-\gamma)\eta_j)\right)\eta_i\|P_i\|_\infty\right)\mathbf{1} + P_t, \quad (49)$$

$$\Delta_t \geq -\prod_{k=0}^{t-1}(1 - (1-\gamma)\eta_k)\|\Delta_0\|_\infty \mathbf{1} - \gamma\eta_{t-1}\|P_{t-1}\|_\infty \mathbf{1} - \gamma\sum_{i=1}^{t-2}\left(\left(\prod_{j=i+1}^{t-1}(1 - (1-\gamma)\eta_j)\right)\eta_i\|P_i\|_\infty\right)\mathbf{1} + P_t, \quad (50)$$

*where $P_t$ represents a discounted sum of the random error $\mathcal{E}_t$, defined as:*

$$P_t = \begin{cases} \mathbf{0} & \text{if } t = 0, \\ \sum_{k=0}^{t-1}\left(\left(\prod_{j=k+1}^{t-1}(1 - \eta_j)\right)\eta_k\mathcal{E}_k\right) & \text{if } t \geq 1. \end{cases} \quad (51)$$

To effectively manage the product term $\prod_{k=0}^{t-1}(1 - (1-\gamma)\eta_k)$, we apply a step size defined as $\eta_k = \frac{1}{1+(1-\gamma)(k+1)}$. Under this choice of step size, we can demonstrate that the expression $1 - (1-\gamma)\eta_k$ satisfies

$$\begin{aligned}
1 - (1-\gamma)\eta_k &= 1 - \frac{1-\gamma}{1+(1-\gamma)(k+1)} \\
&= \frac{1+(1-\gamma)k}{1+(1-\gamma)(k+1)}
\end{aligned}$$

$$= \frac{\eta_k}{\eta_{k-1}}, \quad \forall k \geq 1.$$

Using this result, we can simplify the product-form coefficients in Eq. (49) as follows:

$$\prod_{k=0}^{t-1}(1-(1-\gamma)\eta_k) = (1-(1-\gamma)\eta_0)\prod_{k=1}^{t-1}\frac{\eta_k}{\eta_{k-1}} = \eta_{t-1}, \quad \forall t, \tag{52}$$

$$\left(\prod_{j=i+1}^{t-1}(1-(1-\gamma)\eta_j)\right)\eta_i = \left(\prod_{j=i+1}^{t-1}\frac{\eta_j}{\eta_{j-1}}\right)\eta_i = \eta_{t-1}, \quad \forall t. \tag{53}$$

Substituting Eq. (52) and (53) into the bounds in Eq. (49) and (50), we obtain:

$$\|\Delta_t\|_\infty \leq \eta_{t-1}\|\Delta_0\|_\infty + \gamma\eta_{t-1}\sum_{k=0}^{t-1}\|P_k\|_\infty + \|P_t\|_\infty. \tag{54}$$

To complete the proof, we need to control $\|P_k\|_\infty$ for each $k$. The following lemma provides the necessary bound:

**Lemma F.8** (Proof in Appendix F.10). *Given the step size* $\eta_k = \frac{1}{1+(1-\gamma)k}$*, for any* $k \geq 1$*, with probability at least* $1 - \delta$*, the weighted error sum* $P_k$ *satisfies*

$$\|P_k\|_\infty \leq \frac{2}{3(1-\gamma)(1+(1-\gamma)k)}\log\left(\frac{2|\mathcal{X}[\cup_{k=1}^{K_\omega}Z_k^P]|}{\delta}\right) + \frac{2\sqrt{2\|\sigma_{\mathcal{E}}^2\|_\infty\log\left(\frac{2|\mathcal{X}[\cup_{k=1}^{K_\omega}Z_k^P]|}{\delta}\right)}}{\sqrt{1+(1-\gamma)k}}, \tag{55}$$

*where* $\sigma_{\mathcal{E}}^2$ *is the variance of each random error* $\mathcal{E}_t$*, satisfying* $\sigma_{\mathcal{E}}^2(s,a) = \text{Var}\left(\widehat{\mathcal{H}}(Q_\omega^*)(s,a)\right)$.

This lemma shows that $\|P_k\|_\infty$ is decreasing in the iteration number $k$. Substituting the bound on $\|P_k\|_\infty$ into Eq. (54) yields the following results:

**Lemma F.9** (Proof in Appendix F.11). *With probability at least* $1 - \delta$*, the estimation error* $\Delta_M$ *after* $M$ *iterations in epoch* $\tau = 1$ *satisfies*

$$\|\Delta_M\|_\infty \leq \frac{3+2\log\left(\frac{2M|\mathcal{X}[\cup_{k=1}^{K_\omega}Z_k^P]|}{\delta}\right)}{3(1-\gamma)^2 M} + \frac{2\log\left(\frac{2M|\mathcal{X}[\cup_{k=1}^{K_\omega}Z_k^P]|}{\delta}\right)\log(1+(1-\gamma)M)}{3(1-\gamma)^3 M}$$
$$+ \frac{6\sqrt{2\|\sigma_{\mathcal{E}}^2\|_\infty\log\left(\frac{2M|\mathcal{X}[\cup_{k=1}^{K_\omega}Z_k^P]|}{\delta}\right)}}{(1-\gamma)^{\frac{3}{2}}M^{\frac{1}{2}}},$$

Let $M = c_2\dfrac{\log\left(\frac{6T|\mathcal{X}[\cup_{k=1}^{K_\omega}Z_k^P]|}{(1-\gamma)\delta}\right)}{(1-\gamma)^3}$ with a sufficiently large $c_2$, we have:

$$\|\overline{Q}_1 - Q_\omega^*\|_\infty = \|\Delta_M\|_\infty \leq \frac{\sqrt{\|\sigma_{\mathcal{E}}^2\|_\infty}+1}{2} \leq \frac{1}{2(1-\gamma)} \tag{56}$$

holds with probability at least $1 - \frac{\delta}{T}$, where the last inequality is due to $\|\sigma_{\mathcal{E}}^2\|_\infty \leq \frac{\gamma^2}{(1-\gamma)^2}$. This finishes the proof of the basic case with $\tau = 1$.

**Step 2. Showing the Inductive Case with $\tau \geq 2$:**

We assume that the input $\overline{Q}_\tau$ in epoch $\tau$ satisfies the bound

$$\|\overline{Q}_\tau - Q_\omega^*\|_\infty \leq \frac{\sqrt{\|\sigma_{\mathcal{E}}^2\|_\infty}+1}{2^\tau} := b_\tau,$$

and our goal is to show that $\|\overline{Q}_{\tau+1} - Q^*\|_\infty \leq \frac{b_\tau}{2}$ with probability at least $1 - \frac{\delta}{T}$.

Specifically, $\overline{Q}_{\tau+1}$ is equivalent to the output $Q_M$ of running $M$ rounds of variance-reduced Q-learning from the initialization $Q_0 = \overline{Q}_\tau$. The reference Bellman operator $\overline{\mathcal{H}}_\tau$ is obtained using $N_\tau$ empirical samples.

The variance-reduced update can be rewritten into:

$$Q_{t+1} - Q^* = (1 - \eta_t)(Q_t - Q^*) + \eta_t \underbrace{(\widehat{\mathcal{H}}_t(Q_t) - \widehat{\mathcal{H}}_t(Q^*))}_{\text{Contractive Error: } \mathcal{W}_t(\Delta_t)} + \eta_t \underbrace{(\widehat{\mathcal{H}}_t(Q^*) - \mathcal{H}(Q^*_\omega) - \widehat{\mathcal{H}}_t(\overline{Q}_\tau) + \overline{\mathcal{H}}_{\tau+1}(\overline{Q}_\tau))}_{\text{Variance-reduced Error: } \overline{\mathcal{E}}_t}.$$

Observe that, the update form contains three components: the decaying term $(1 - \eta_t)(Q_t - Q^*)$, the contractive error term $\mathcal{W}_t(\Delta_t)$ and the variance reduced error term $\overline{\mathcal{E}}_t$. The only difference between the variance-reduced iteration and the vanilla iteration comes from the variance-reduced error $\overline{\mathcal{E}}_t$, which can be further split into the following form:

$$\begin{aligned}
&\widehat{\mathcal{H}}_t(Q^*) - \mathcal{H}(Q^*_\omega) - \widehat{\mathcal{H}}_t(\overline{Q}_\tau) + \overline{\mathcal{H}}_{\tau+1}(\overline{Q}_\tau) \\
=&\widehat{\mathcal{H}}_t(Q^*) - \widehat{\mathcal{H}}_t(\overline{Q}_\tau) + \overline{\mathcal{H}}_{\tau+1}(\overline{Q}_\tau) - \overline{\mathcal{H}}_{\tau+1}(Q^*_\omega) + \overline{\mathcal{H}}_{\tau+1}(Q^*_\omega) - \mathcal{H}(Q^*_\omega) \\
=&\underbrace{\mathcal{H}(\overline{Q}_\tau) - \mathcal{H}(Q^*_\omega) - \widehat{\mathcal{H}}_t(\overline{Q}_\tau) + \widehat{\mathcal{H}}_t(Q^*_\omega)}_{\overline{\mathcal{E}}^a_t} + \underbrace{\overline{\mathcal{H}}_{\tau+1}(\overline{Q}_\tau) - \overline{\mathcal{H}}_{\tau+1}(Q^*_\omega) - \mathcal{H}(\overline{Q}_\tau) + \mathcal{H}(Q^*_\omega)}_{\overline{\mathcal{E}}^b} + \underbrace{\overline{\mathcal{H}}_{\tau+1}(Q^*_\omega) - \mathcal{H}(Q^*_\omega)}_{\overline{\mathcal{E}}^c}
\end{aligned}$$

An important observation is that, $\overline{\mathcal{E}}^a_t$ depends on the empirical Bellman operator sampled in each iteration $t$, while $\overline{\mathcal{E}}^b$ and $\overline{\mathcal{E}}^c$ are independent of each iteration, but only dependent on the reference Bellman operator $\overline{T}_\tau$ sampled in the begining of iteration.

We apply the result in Lemma F.7 for the bound on the error accumulation in iterative updates, and get the following bound on $\Delta_t = Q_t - Q^*$:

$$\|\Delta_t\|_\infty \leq \eta_{t-1}\|\Delta_0\|_\infty + \gamma\eta_{t-1}\sum_{k=0}^{t-1}(\|P^a_k\|_\infty + \|P^b_k\|_\infty + \|P^c_k\|_\infty) + \|P^a_t\|_\infty + \|P^b_t\|_\infty + \|P^c_t\|_\infty, \quad (57)$$

where $P^a_t$, $P^b_t$ and $P^c_t$ are the discounted sums of $\overline{E}^a_t$, $\overline{E}^b$ and $\overline{E}^c$ satisfying:

$$P^a_t = \begin{cases} \mathbf{0} & \text{if } t = 0, \\ \sum_{k=0}^{t-1}\left(\left(\prod_{j=k+1}^{t-1}(1-\eta_j)\right)\eta_k\overline{\mathcal{E}}^a_k\right) & \text{if } t \geq 1. \end{cases} \quad (58)$$

$$P^b_t = \begin{cases} \mathbf{0} & \text{if } t = 0, \\ \sum_{k=0}^{t-1}\left(\left(\prod_{j=k+1}^{t-1}(1-\eta_j)\right)\eta_k\overline{\mathcal{E}}^b\right) & \text{if } t \geq 1. \end{cases} \quad (59)$$

$$P^c_t = \begin{cases} \mathbf{0} & \text{if } t = 0, \\ \sum_{k=0}^{t-1}\left(\left(\prod_{j=k+1}^{t-1}(1-\eta_j)\right)\eta_k\overline{\mathcal{E}}^c\right) & \text{if } t \geq 1. \end{cases} \quad (60)$$

To simplify, we have that $\|P^b_t\|_\infty \leq \|\overline{\mathcal{E}}^b\|_\infty$ and $\|P^c_t\|_\infty \leq \|\overline{\mathcal{E}}^c\|_\infty$. Since $\|\overline{\mathcal{E}}^b\|_\infty$ and $\|\overline{\mathcal{E}}^c\|_\infty$ are independent of $t$, by substituting $\eta_{t-1} = \frac{1}{1+(1-\gamma)t}$, we have

$$\begin{aligned}
&\gamma\eta_{t-1}\sum_{k=0}^{t-1}(\|\overline{\mathcal{E}}^b\|_\infty + \|\overline{\mathcal{E}}^c\|_\infty) + \|\overline{\mathcal{E}}^b\|_\infty + \|\overline{\mathcal{E}}^c\|_\infty \\
=&\left(1 + \frac{\gamma t}{1 + (1-\gamma)t}\right)(\|\overline{\mathcal{E}}^b\|_\infty + \|\overline{\mathcal{E}}^c\|_\infty) \\
\leq&\frac{2}{1-\gamma}(\|\overline{\mathcal{E}}^b\|_\infty + \|\overline{\mathcal{E}}^c\|_\infty).
\end{aligned}$$

Hence, $\Delta_t$ satisfies

$$\|\Delta_t\|_\infty \leq \eta_{t-1}\|\Delta_0\|_\infty + \gamma\eta_{t-1}\sum_{k=0}^{t-1}\|P^a_k\|_\infty + \|P^a_t\|_\infty + \frac{2}{1-\gamma}(\|\overline{\mathcal{E}}^b\|_\infty + \|\overline{\mathcal{E}}^c\|_\infty). \quad (61)$$

The remaining challenge is to bound the following three terms: $\|\overline{\mathcal{E}}^b\|$, $\|\overline{\mathcal{E}}^c\|$ and $\|P^a_k\|$.

**Lemma F.10** (Proof in Appendix F.12). *In $\tau$-th iteration, with probability at least $1 - \frac{\delta}{3T}$, $\overline{\mathcal{E}}^b$ satisfies*

$$\|\overline{\mathcal{E}}^b\|_\infty \leq b_\tau \sqrt{\frac{2\log\left(\frac{6T|\mathcal{X}[\cup_{k=1}^{K_\omega} Z_k^P]|}{\delta}\right)}{N_\tau}}.$$

**Lemma F.11** (Proof in Appendix F.13). *In $\tau$-th iteration, with probability at least $1 - \frac{\delta}{3T}$, $\overline{\mathcal{E}}^c$ satisfies*

$$\|\overline{\mathcal{E}}^c\|_\infty \leq \overline{c}_1 \sqrt{\frac{\log\left(\frac{2T|\mathcal{X}[\cup_{k=1}^{K_\omega} Z_k^P]|}{\delta}\right)}{N_\tau}} \left(\sqrt{\|\sigma_{\mathcal{E}}^2\|_\infty} + 1\right),$$

*where $\overline{c}_1$ is an absolute constant.*

**Lemma F.12** (Proof in Appendix F.14). *In the $\tau$-th iteration, with probability at least $1 - \frac{\delta}{3T}$, it satisfies*

$$\gamma\eta_{t-1}\sum_{k=0}^{t-1}\|P_k^a\|_\infty + \|P_t^a\|_\infty \leq \overline{c}_2 b_\tau \left(\frac{\log\left(\frac{6TM|\mathcal{X}[\cup_{k=1}^{K_\omega} Z_k^P]|}{\delta}\right)\log(1+(1-\gamma)t)}{(1-\gamma)^2 t} + \frac{\sqrt{\log\left(\frac{6TM|\mathcal{X}[\cup_{k=1}^{K_\omega} Z_k^P]|}{\delta}\right)}}{(1-\gamma)^{\frac{3}{2}} t^{\frac{1}{2}}}\right),$$

*where $\overline{c}_2$ is an absolute constant.*

Combining Lemmas F.10, F.11 and F.12 yields:

$$\|\Delta_M\|_\infty \leq \frac{b_\tau}{1+(1-\gamma)M} + \frac{\overline{c}_1\sqrt{\frac{\log\left(\frac{2T|\mathcal{X}[\cup_{k=1}^{K_\omega} Z_k^P]|}{\delta}\right)}{N_\tau}}\left(\sqrt{\|\sigma_{\mathcal{E}}^2\|_\infty}+1\right) + b_\tau\sqrt{\frac{\log\left(\frac{2T|\mathcal{X}[\cup_{k=1}^{K_\omega} Z_k^P]|}{\delta}\right)}{N_\tau}}}{1-\gamma}$$

$$+ \overline{c}_2 b_\tau\left(\frac{\log\left(\frac{6TM|\mathcal{X}[\cup_{k=1}^{K_\omega} Z_k^P]|}{\delta}\right)\log(1+(1-\gamma)M)}{(1-\gamma)^2 M} + \frac{\sqrt{\log\left(\frac{6TM|\mathcal{X}[\cup_{k=1}^{K_\omega} Z_k^P]|}{\delta}\right)}}{(1-\gamma)^{\frac{3}{2}} M^{\frac{1}{2}}}\right).$$

Substituting $M = c_2 \frac{\log\left(\frac{6T|\mathcal{X}[\cup_{k=1}^{K_\omega} Z_k^P]|}{(1-\gamma)\delta}\right)}{(1-\gamma)^3}$ and $N_\tau = c_3 4^\tau \frac{\log\left(6T|\mathcal{X}[\cup_{k=1}^{K_\omega} Z_k^P]|\right)}{(1-\gamma)^2}$ with large enough $c_2$ and $c_3$ yields that:

$$\|\Delta_M\|_\infty \leq \frac{\sqrt{\|\sigma_{\mathcal{E}}^2\|_\infty}+1}{2^{\tau+1}} \leq \frac{1}{(1-\gamma)2^{\tau+1}}.$$

This concludes our proof.

### F.9. Proof to Lemma F.7

*Proof.* We will prove the result by induction. First, by applying Eq. (47), we have that $\Delta_1$ satisfies

$$\begin{aligned}
\Delta_1 &= (1-\eta_0)\Delta_0 + \eta_0(\widehat{\mathcal{H}}_0(Q_\omega^* + \Delta_0) - \widehat{\mathcal{H}}_0(Q_\omega^*)) + \eta_0(\widehat{\mathcal{H}}_0(Q_\omega^*) - \mathcal{H}(Q_\omega^*)) \\
&\leq (1-\eta_0)\|\Delta_0\|_\infty \mathbf{1} + \eta_0\gamma\|\Delta_0\|_\infty \mathbf{1} + \eta_0\mathcal{E}_0 \\
&= (1-(1-\gamma)\eta_0)\|\Delta_0\|_\infty \mathbf{1} + \eta_0\mathcal{E}_0 \\
&= (1-(1-\gamma)\eta_0)\|\Delta_0\|_\infty \mathbf{1} + P_1,
\end{aligned} \tag{62}$$

where the inequality follows from the contraction property $\widehat{\mathcal{H}}_0(Q_\omega^* + \Delta_0) - \widehat{\mathcal{H}}_0(Q_\omega^*) \leq \gamma\|\Delta_0\|_\infty\mathbf{1}$.

Similarly, $\Delta_1$ can be lower bounded as follows:

$$\Delta_1 = (1-\eta_0)\Delta_0 + \eta_0(\widehat{\mathcal{H}}_0(Q_\omega^* + \Delta_0) - \widehat{\mathcal{H}}_0(Q_\omega^*)) + \eta_0(\widehat{\mathcal{H}}_0(Q_\omega^*) - \mathcal{H}(Q_\omega^*))$$

$$\geq -(1-\eta_0)\|\Delta_0\|_\infty \mathbf{1} - \eta_0\gamma\|\Delta_0\|_\infty \mathbf{1} + \eta_0\mathcal{E}_0$$
$$= -(1-(1-\gamma)\eta_0)\|\Delta_0\|_\infty \mathbf{1} + \eta_0\mathcal{E}_0$$
$$= -(1-(1-\gamma)\eta_0)\|\Delta_0\|_\infty \mathbf{1} + P_1. \tag{63}$$

The expressions in Eq. (62) and (63) are consistent with the upper and lower bounds provided by Eq. (49) and (50) in Lemma F.7.

Now, suppose that the conditions in Eq. (49) and (50) hold for $\Delta_t$, that is:

$$\Delta_t \leq a_t\|\Delta_0\|_\infty \mathbf{1} + b_t\mathbf{1} + P_t, \tag{64}$$
$$\Delta_t \geq -a_t\|\Delta_0\|_\infty \mathbf{1} - b_t\mathbf{1} + P_t, \tag{65}$$

where

$$a_t = \prod_{k=0}^{t-1}(1-(1-\gamma)\eta_k), \quad b_t = \gamma\eta_{t-1}\|P_{t-1}\|_\infty + \gamma\sum_{i=1}^{t-2}\left(\left(\prod_{j=i+1}^{t-1}(1-(1-\gamma)\eta_j)\right)\eta_i\|P_i\|_\infty\right).$$

Applying Eq. (47), $\Delta_{t+1}$ satisfies

$$\Delta_{t+1} = (1-\eta_t)\Delta_t + \eta_t\mathcal{W}_t(\Delta_t) + \eta_t\mathcal{E}_t$$
$$\leq (1-\eta_t)(a_t\|\Delta_0\|_\infty\mathbf{1} + b_t\mathbf{1} + P_t) + \eta_t\gamma\|\Delta_t\|_\infty\mathbf{1} + \eta_t\mathcal{E}_t \tag{66}$$
$$\leq (1-\eta_t)(a_t\|\Delta_0\|_\infty\mathbf{1} + b_t\mathbf{1} + P_t) + \eta_t\gamma(a_t\|\Delta_0\|_\infty + b_t + \|P_t\|_\infty)\mathbf{1} + \eta_t\mathcal{E}_t \tag{67}$$
$$= \underbrace{(1-(1-\gamma)\eta_t)a_t}_{a_{t+1}}\|\Delta_0\|_\infty\mathbf{1} + \underbrace{(\gamma\eta_t\|P_t\|_\infty + (1-(1-\gamma)\eta_t)b_t)}_{b_{t+1}}\mathbf{1} + \underbrace{(1-\eta_t)P_t + \eta_t\mathcal{E}_t}_{P_{t+1}}, \tag{68}$$

where inequality (66) is obtained by applying the upper bound condition (64) and the contraction property $\mathcal{W}_t(\Delta_t) \leq \gamma\|\Delta_t\|_\infty\mathbf{1}$; Inequality (67) is obtained by using conditions (64); Equality (68) is achieved by rearranging the terms.

Following the same steps, we can establish the corresponding lower bound for $\Delta_{t+1}$ in Eq. (50). This completes the proof. $\qquad\square$

### F.10. Proof to Lemma F.8

*Proof.* Recall the definition in Eq. (51). For any $t \geq 2$, $P_t$ can be expressed as a weighted sum:

$$P_t = \sum_{k=0}^{t-1} a_{t,k}\mathcal{E}_k,$$

where $a_{t,k} = \left(\prod_{j=k+1}^{t-1}(1-\eta_j)\right)\eta_k$.

Denote the $(s,a)$-th entry of $P_t$ as $P_t(s,a)$. Since all $\mathcal{E}_k$ are independent and identically distributed, we can apply Bernstein's inequality (Vershynin, 2018) to $P_t(s,a)$. This gives us the following probability bound:

$$\mathbb{P}\left(|P_t(s,a)| \geq \epsilon_t\right) \leq 2\exp\left(-\frac{\epsilon_t^2}{2\sum_{k=0}^{t-1}a_{t,k}^2\sigma_\mathcal{E}^2(s,a) + \frac{2}{3}(\max_k|a_{t,k}|M_{\mathcal{E}(s,a)})\epsilon_t}\right), \quad \forall(s,a),$$

where $\sigma_\mathcal{E}^2(s,a)$ is the variance of $\mathcal{E}(s,a)$, and $M_{\mathcal{E}(s,a)}$ is the maximum absolute value of $\mathcal{E}(s,a)$.

Recall that $P_t$ is the weighted sum of different $\mathcal{E}_k$'s, and all $\mathcal{E}_k$ are *i.i.d.* random variables. Each $\mathcal{E}_k$ has at most $|\mathcal{X}[\cup_{k=1}^{K_\omega}Z_k^P]|$ distinct entries due to the sampling scheme, where $\mathcal{X}[\cup_{k=1}^{K_\omega}Z_k^P]$ represents the set of possible state-action pairs involved in the components' scopes. We denote these distinct state-action pairs by $\mathcal{X}^*$ defined in Eq. (41).

Taking the union bound over all state-action pairs $x := (s,a) \in \mathcal{X}^*$, we have:

$$\mathbb{P}\left(\|P_t\|_\infty \geq \epsilon_t\right) \leq 2|\mathcal{X}[\cup_{k=1}^{K_\omega}Z_k^P]|\exp\left(-\frac{\epsilon_t^2}{2\sum_{k=0}^{t-1}a_{t,k}^2\sigma_\mathcal{E}^2(s,a) + \frac{2}{3}(\max_k|a_{t,k}|M_{\mathcal{E}(s,a)})\epsilon_t}\right).$$

To set this probability to $\delta$, let $\mathbb{P}\left(\|P_t\|_\infty \geq \epsilon_t\right) = \delta$. Then, we can conclude:

$$\delta \leq 2|\mathcal{X}[\cup_{k=1}^{K_\omega} Z_k^P]| \exp\left(-\frac{\epsilon_t^2}{2\sum_{k=0}^{t-1} a_{t,k}^2 \sigma_{\mathcal{E}}^2(s,a) + \frac{2}{3}(\max_k |a_{t,k}|M_{\mathcal{E}(s,a)})\epsilon_t}\right).$$

Taking the natural logarithm on both sides and rearranging terms gives:

$$\epsilon_t^2 - \log\left(\frac{2|\mathcal{X}[\cup_{k=1}^{K_\omega} Z_k^P]|}{\delta}\right)\left(2\sum_{k=0}^{t-1} a_{t,k}^2 \sigma_{\mathcal{E}}^2(s,a) + \frac{2}{3}\left(\max_k |a_{t,k}|M_{\mathcal{E}(s,a)}\right)\epsilon_t\right) \leq 0.$$

To solve for $\epsilon_t$, we treat this as a quadratic inequality:

$$\epsilon_t^2 - \left(\frac{2}{3}\max_k |a_{t,k}|M_{\mathcal{E}(s,a)}\right)\log\left(\frac{2|\mathcal{X}[\cup_{k=1}^{K_\omega} Z_k^P]|}{\delta}\right)\epsilon_t - 2\log\left(\frac{2|\mathcal{X}[\cup_{k=1}^{K_\omega} Z_k^P]|}{\delta}\right)\sum_{k=0}^{t-1} a_{t,k}^2 \sigma_{\mathcal{E}}^2(s,a) \leq 0.$$

To simplify further:

$$\epsilon_t \leq \frac{2}{3}\log\left(\frac{2|\mathcal{X}[\cup_{k=1}^{K_\omega} Z_k^P]|}{\delta}\right)\max_k |a_{t,k}|M_{\mathcal{E}(s,a)} + 2\sqrt{\log\left(\frac{2|\mathcal{X}[\cup_{k=1}^{K_\omega} Z_k^P]|}{\delta}\right)\sum_{k=0}^{t-1} a_{t,k}^2 \sigma_{\mathcal{E}}^2(s,a)}. \tag{69}$$

To complete the analysis, we need to bound $\max_k |a_{t,k}|$, $M_{\mathcal{E}(s,a)}$, and $\sum_{k=0}^{t-1} a_{t,k}^2$.

**Step 1: Bounding $M_{\mathcal{E}(s,a)}$**

For any $k$, since the empirical Bellman operator $\widehat{\mathcal{H}}_k(Q^*)$ and the true Bellman operator $\mathcal{H}(Q_\omega^*)$ are both bounded between 0 and $\frac{1}{1-\gamma}$, we have:

$$M_{\mathcal{E}(s,a)} = \sup \|\mathcal{E}_k\|_\infty = \sup \|\widehat{\mathcal{H}}_k(Q^*) - \mathcal{H}(Q_\omega^*)\|_\infty \leq \frac{1}{1-\gamma}.$$

**Step 2: Bounding $\max_k |a_{t,k}|$**

Next, we analyze the ratio $\frac{a_{t,k}}{a_{t,k-1}}$ to find $\max_k |a_{t,k}|$. Recall the definition of $a_{t,k}$:

$$a_{t,k} = \left(\prod_{j=k+1}^{t-1}(1-\eta_j)\right)\eta_k.$$

To compute $\frac{a_{t,k}}{a_{t,k-1}}$, we have:

$$\frac{a_{t,k}}{a_{t,k-1}} = \frac{\eta_k}{\eta_{k-1}(1-\eta_k)}.$$

Using the step size $\eta_k = \frac{1}{1+(1-\gamma)(k+1)}$, we can express:

$$\frac{a_{t,k}}{a_{t,k-1}} = \frac{1-(1-\gamma)\eta_k}{1-\eta_k}.$$

Given that $0 < \gamma < 1$, we have $\eta_k \leq \frac{1}{1+(1-\gamma)k} < 1$, and thus:

$$\frac{a_{t,k}}{a_{t,k-1}} \geq 1, \quad \forall k.$$

This means that $a_{t,k}$ is non-decreasing with respect to $k$. Hence, the maximum value $\max_k |a_{t,k}|$ occurs at $k = t-1$:

$$\max_k |a_{t,k}| = a_{t,t-1} = \eta_{t-1} = \frac{1}{1+(1-\gamma)t}.$$

**Step 3: Bounding $\sum_{k=0}^{t-1} a_{t,k}^2$**

We also need to bound $\sum_{k=0}^{t-1} a_{t,k}^2$. The following lemma provides this bound:

**Lemma F.13** (Proof in Appendix F.15). *For any $t \geq 1$, the sum $\sum_{k=0}^{t-1} a_{t,k}^2$ satisfies*

$$\sum_{k=0}^{t-1} a_{t,k}^2 \leq \frac{2}{1 + (1-\gamma)t}.$$

Combining the results from Steps 1, 2, and 3, we can now derive the bound for $\epsilon_t$. Substituting these bounds into Eq. (69), we get that with probability at least $1 - \delta$:

$$\|P_t\|_\infty \leq \epsilon_t \leq \frac{2}{3(1-\gamma)(1+(1-\gamma)t)} \log\left(\frac{2|\mathcal{X}[\cup_{k=1}^{K_\omega} Z_k^P]|}{\delta}\right) + \frac{2\sqrt{2\|\sigma_\mathcal{E}^2\|_\infty \log\left(\frac{2|\mathcal{X}[\cup_{k=1}^{K_\omega} Z_k^P]|}{\delta}\right)}}{\sqrt{1+(1-\gamma)t}}.$$

This concludes our proof. $\qquad\square$

### F.11. Proof of Lemma F.9

*Proof.* Letting the upper bound of each $\|P_k\|_\infty$ (in Eq. (55)) with $k = 1, \ldots, M$ holds with probability $1 - \frac{\delta}{M}$, we have:

$$\|\Delta_M\|_\infty \leq \frac{\|\Delta_0\|_\infty}{1+(1-\gamma)M} + \frac{2\log\left(\frac{2M|\mathcal{X}[\cup_{k=1}^{K_\omega} Z_k^P]|}{\delta}\right)}{3(1-\gamma)^2 M} + \frac{2\sqrt{2\|\sigma_\mathcal{E}^2\|_\infty \log\left(\frac{2M|\mathcal{X}[\cup_{k=1}^{K_\omega} Z_k^P]|}{\delta}\right)}}{\sqrt{1+(1-\gamma)M}}$$

$$+ \frac{2\log\left(\frac{2M|\mathcal{X}[\cup_{k=1}^{K_\omega} Z_k^P]|}{\delta}\right)}{3(1-\gamma)^2 M} \sum_{i=1}^{M} \frac{1}{1+(1-\gamma)i}$$

$$+ \frac{2\sqrt{2\|\sigma_\mathcal{E}^2\|_\infty \log\left(\frac{2M|\mathcal{X}[\cup_{k=1}^{K_\omega} Z_k^P]|}{\delta}\right)}}{1+(1-\gamma)M} \sum_{i=1}^{M} \frac{1}{\sqrt{1+(1-\gamma)i}}.$$

Note that,

$$\sum_{i=1}^{M} \frac{1}{1+(1-\gamma)i} \leq \int_0^M \frac{1}{1+(1-\gamma)x} dx = \frac{\log\left(1+(1-\gamma)M\right)}{1-\gamma},$$

$$\sum_{i=1}^{M} \frac{1}{\sqrt{1+(1-\gamma)i}} \leq \int_0^M \frac{1}{\sqrt{1+(1-\gamma)x}} dx = \frac{2\left(\sqrt{1+(1-\gamma)M}-1\right)}{1-\gamma} \leq \frac{2\sqrt{1+(1-\gamma)M}}{1-\gamma}.$$

Hence, $\Delta_M$ satisfies

$$\|\Delta_M\|_\infty \leq \frac{\|\Delta_0\|_\infty}{1+(1-\gamma)M} + \frac{2\log\left(\frac{2M|\mathcal{X}[\cup_{k=1}^{K_\omega} Z_k^P]|}{\delta}\right)}{3(1-\gamma)^2 M} + \frac{2\sqrt{2\|\sigma_\mathcal{E}^2\|_\infty \log\left(\frac{2M|\mathcal{X}[\cup_{k=1}^{K_\omega} Z_k^P]|}{\delta}\right)}}{\sqrt{1+(1-\gamma)M}}$$

$$+ \frac{2\log\left(\frac{2M|\mathcal{X}[\cup_{k=1}^{K_\omega} Z_k^P]|}{\delta}\right) \log(1+(1-\gamma)M)}{3(1-\gamma)^3 M} + \frac{4\sqrt{2\|\sigma_\mathcal{E}^2\|_\infty \log\left(\frac{2M|\mathcal{X}[\cup_{k=1}^{K_\omega} Z_k^P]|}{\delta}\right)}}{(1-\gamma)^{\frac{3}{2}} M^{\frac{1}{2}}}$$

$$\leq \frac{3 + 2\log\left(\frac{2M|\mathcal{X}[\cup_{k=1}^{K_\omega} Z_k^P]|}{\delta}\right)}{3(1-\gamma)^2 M} + \frac{2\log\left(\frac{2M|\mathcal{X}[\cup_{k=1}^{K_\omega} Z_k^P]|}{\delta}\right) \log(1+(1-\gamma)M)}{3(1-\gamma)^3 M}$$

$$+ \frac{6\sqrt{2\|\sigma_{\mathcal{E}}^2\|_\infty \log\left(\frac{2M|\mathcal{X}[\cup_{k=1}^{K_\omega} Z_k^P]|}{\delta}\right)}}{(1-\gamma)^{\frac{3}{2}} M^{\frac{1}{2}}}.$$

This concludes our proof.

$\square$

### F.12. Proof of Lemma F.10

*Proof.* Recall the definition of $\overline{\mathcal{E}}^b$, which is given by:

$$\overline{\mathcal{E}}^b = \overline{\mathcal{H}}_{\tau+1}(\overline{Q}_\tau) - \overline{\mathcal{H}}_{\tau+1}(Q_\omega^*) - \mathcal{H}(\overline{Q}_\tau) + \mathcal{H}(Q_\omega^*).$$

We first note that the expectation of the difference between the empirical Bellman operators is:

$$\mathbb{E}_{\overline{\mathcal{H}}_{\tau+1}}\left[\overline{\mathcal{H}}_{\tau+1}(\overline{Q}_\tau) - \overline{\mathcal{H}}_{\tau+1}(Q_\omega^*)\right] = \mathcal{H}(\overline{Q}_\tau) - \mathcal{H}(Q_\omega^*).$$

Due to the $\gamma$-contraction property of both the empirical Bellman operator $\overline{\mathcal{H}}_{\tau+1}$ and the true Bellman operator $\mathcal{H}$, we have:

$$\|\overline{\mathcal{H}}_{\tau+1}(\overline{Q}_\tau) - \overline{\mathcal{H}}_{\tau+1}(Q_\omega^*) - \mathcal{H}(\overline{Q}_\tau) + \mathcal{H}(Q_\omega^*)\|_\infty \le \|\overline{\mathcal{H}}_{\tau+1}(\overline{Q}_\tau) - \overline{\mathcal{H}}_{\tau+1}(Q_\omega^*)\|_\infty + \|\mathcal{H}(\overline{Q}_\tau) - \mathcal{H}(Q_\omega^*)\|_\infty.$$

Since both operators are $\gamma$-contractions, it follows that:

$$\|\overline{\mathcal{H}}_{\tau+1}(\overline{Q}_\tau) - \overline{\mathcal{H}}_{\tau+1}(Q_\omega^*)\|_\infty \le \gamma\|\overline{Q}_\tau - Q_\omega^*\|_\infty \quad \text{and} \quad \|\mathcal{H}(\overline{Q}_\tau) - \mathcal{H}(Q_\omega^*)\|_\infty \le \gamma\|\overline{Q}_\tau - Q_\omega^*\|_\infty.$$

Combining these, we get:

$$\|\overline{\mathcal{H}}_{\tau+1}(\overline{Q}_\tau) - \overline{\mathcal{H}}_{\tau+1}(Q_\omega^*) - \mathcal{H}(\overline{Q}_\tau) + \mathcal{H}(Q_\omega^*)\|_\infty \le 2\gamma\|\overline{Q}_\tau - Q_\omega^*\|_\infty \le 2\gamma b_\tau.$$

Applying Hoeffding's inequality to the $(s, a)$-th entry of $\overline{\mathcal{E}}^b$, denoted as $\overline{\mathcal{E}}^b(s, a)$, we obtain:

$$\mathbb{P}\left(|\overline{\mathcal{E}}^b(s, a)| \ge \epsilon\right) \le 2\exp\left(-\frac{2N_\tau \epsilon^2}{4b_\tau^2}\right) = 2\exp\left(-\frac{N_\tau \epsilon^2}{2b_\tau^2}\right).$$

Letting $2\exp\left(-\frac{N_\tau \epsilon^2}{2b_\tau^2}\right) = \frac{\delta}{3T|\mathcal{X}[\cup_{k=1}^{K_\omega} Z_k^P]|}$, and taking union bound for $x \in \mathcal{X}^*$, we have that:

$$\|\overline{\mathcal{E}}^b\|_\infty \le b_\tau \sqrt{\frac{2\log\left(\frac{6T|\mathcal{X}[\cup_{k=1}^{K_\omega} Z_k^P]|}{\delta}\right)}{N_\tau}}.$$

This completes the proof. $\square$

### F.13. Proof of Lemma F.11

*Proof.* Recall that $\overline{E}^c = \overline{T}(Q^*) - \mathcal{H}(Q_\omega^*)$ represents the estimation error using $N_\tau$ empirical Bellman operators. We know that $\mathbb{E}[\overline{T}(Q^*)] = \mathcal{H}(Q_\omega^*)$.

Applying Bernstein's inequality (Vershynin, 2018) to the $(s, a)$-th entry of $\overline{E}^c$, denoted as $\overline{\mathcal{E}}_{(s,a)}^c$, we obtain:

$$\mathbb{P}\left(|\overline{\mathcal{E}}_{(s,a)}^c| \ge \epsilon\right) \le 2\exp\left(-\frac{\frac{1}{2}N_\tau \epsilon^2}{\|\sigma_{\mathcal{E}}^2\|_\infty + \frac{\epsilon}{3(1-\gamma)}}\right).$$

We set the right-hand side probability equal to $\frac{\delta}{3T|\mathcal{X}[\cup_{k=1}^{K_\omega} Z_k^P]|}$, which gives us:

$$2\exp\left(-\frac{\frac{1}{2}N_\tau \epsilon^2}{\|\sigma_{\overline{\mathcal{E}}}^2\|_\infty + \frac{\epsilon}{3(1-\gamma)}}\right) = \frac{\delta}{3T|\mathcal{X}[\cup_{k=1}^{K_\omega} Z_k^P]|}.$$

Rearranging this equation for $\epsilon$, we get:

$$\frac{1}{2}N_\tau \epsilon^2 - \frac{\epsilon \log(6T|\mathcal{X}[\cup_{k=1}^{K_\omega} Z_k^P]|/\delta)}{3(1-\gamma)} - \|\sigma_{\overline{\mathcal{E}}}^2\|_\infty \log(6T|\mathcal{X}[\cup_{k=1}^{K_\omega} Z_k^P]|/\delta) = 0.$$

Solving this quadratic equation for $\epsilon$, we have:

$$\epsilon \leq \frac{\frac{2\log(6T|\mathcal{X}[\cup_{k=1}^{K_\omega} Z_k^P]|/\delta)}{3(1-\gamma)} + \sqrt{\frac{4\log^2(6T|\mathcal{X}[\cup_{k=1}^{K_\omega} Z_k^P]|/\delta)}{9(1-\gamma)^2} + 2N_\tau \|\sigma_{\overline{\mathcal{E}}}^2\|_\infty \log(6T|\mathcal{X}[\cup_{k=1}^{K_\omega} Z_k^P]|/\delta)}}{N_\tau}.$$

Simplifying further, we find:

$$\epsilon \leq \frac{2\log(6T|\mathcal{X}[\cup_{k=1}^{K_\omega} Z_k^P]|/\delta)}{3N_\tau(1-\gamma)} + \sqrt{\frac{2\|\sigma_{\overline{\mathcal{E}}}^2\|_\infty \log(6T|\mathcal{X}[\cup_{k=1}^{K_\omega} Z_k^P]|/\delta)}{N_\tau}}.$$

Applying the union bound over all $x := (s,a) \in \mathcal{X}^*$, we conclude that with probability at least $1 - \frac{\delta}{3T}$, the infinity norm $\|\overline{\mathcal{E}}^c\|_\infty$ satisfies

$$\|\overline{\mathcal{E}}^c\|_\infty \leq \frac{2\log(6T|\mathcal{X}[\cup_{k=1}^{K_\omega} Z_k^P]|/\delta)}{3N_\tau(1-\gamma)} + \sqrt{\frac{2\|\sigma_{\overline{\mathcal{E}}}^2\|_\infty \log(6T|\mathcal{X}[\cup_{k=1}^{K_\omega} Z_k^P]|/\delta)}{N_\tau}}. \tag{70}$$

Given that $N_\tau \geq c \cdot 4^\tau \frac{\log(8T|\mathcal{X}[\cup_{k=1}^{K_\omega} Z_k^P]|/\delta)}{(1-\gamma)^2}$, it follows that $\frac{\log(2T|\mathcal{X}[\cup_{k=1}^{K_\omega} Z_k^P]|/\delta)}{N_\tau(1-\gamma)} \leq 1$ for a large enough $c$. Thus, $\|\overline{\mathcal{E}}^c\|_\infty$ is further bounded by:

$$\|\overline{\mathcal{E}}^c\|_\infty \leq \overline{c}_1 \left(\sqrt{\frac{\|\sigma_{\overline{\mathcal{E}}}^2\|_\infty \log\left(\frac{6T|\mathcal{X}[\cup_{k=1}^{K_\omega} Z_k^P]|}{\delta}\right)}{N_\tau}} + 1\right)$$

$$\leq \overline{c}_1 \sqrt{\frac{\log\left(\frac{6T|\mathcal{X}[\cup_{k=1}^{K_\omega} Z_k^P]|}{\delta}\right)}{N_\tau}} \left(\sqrt{\|\sigma_{\overline{\mathcal{E}}}^2\|_\infty} + 1\right),$$

where $\overline{c}_1$ is an absolute constant. This completes the proof. $\qquad\square$

### F.14. Proof of Lemma F.12

*Proof.* The term $P_k^a$ evolves in an exactly the same way as $P_k$ discussed before. Hence, we apply the result in Lemma F.8 and get that, with probability at least $1 - \frac{\delta}{3TM}$:

$$\|P_t^a\|_\infty \leq \frac{2M_{\mathcal{E},a}}{3(1+(1-\gamma)t)} \log\left(\frac{6TM|\mathcal{X}[\cup_{k=1}^{K_\omega} Z_k^P]|}{\delta}\right) + \frac{2\sqrt{2\|\sigma_{\mathcal{E},a}^2\|_\infty \log\left(\frac{6TM|\mathcal{X}[\cup_{k=1}^{K_\omega} Z_k^P]|}{\delta}\right)}}{\sqrt{1+(1-\gamma)t}}, \forall t.$$

Here $M_{\mathcal{E},a}$ and $\sigma_{\mathcal{E},a}^2$ denote the maximal absolute value and variance of error $\overline{\mathcal{E}}_t^a$.

First, we observe that $\overline{\mathcal{E}}_t^a$ satisfies

$$
\begin{aligned}
\|\overline{\mathcal{E}}_t^a\|_\infty &= \|\mathcal{H}(\overline{Q}_\tau) - \mathcal{H}(Q_\omega^*) - \widehat{\mathcal{H}}_t(\overline{Q}_\tau) + \widehat{\mathcal{H}}_t(Q_\omega^*)\|_\infty \\
&\leq \|\mathcal{H}(\overline{Q}_\tau) - \mathcal{H}(Q_\omega^*)\|_\infty + \|\widehat{\mathcal{H}}_t(\overline{Q}_\tau) - \widehat{\mathcal{H}}_t(Q_\omega^*)\|_\infty \\
&\leq 2\gamma\|\overline{Q}_\tau - Q^*\|_\infty \\
&\leq 2b_\tau.
\end{aligned}
$$

Thus, we have $M_{\mathcal{E},a} \leq 2b_\tau$ and $\sqrt{\|\sigma_{\mathcal{E},a}^2\|_\infty} \leq 2b_\tau$. And $\|P_t^a\|_\infty$ then satisfies

$$
\|P_t^a\|_\infty \leq b_\tau \left( \frac{4\log\left(\frac{6TM|\mathcal{X}[\cup_{k=1}^{K_\omega} Z_k^P]|}{\delta}\right)}{3(1+(1-\gamma)t)} + \sqrt{\frac{8\log\left(\frac{6TM|\mathcal{X}[\cup_{k=1}^{K_\omega} Z_k^P]|}{\delta}\right)}{1+(1-\gamma)t}} \right)
$$

Thus, we can control the whole term by:

$$
\begin{aligned}
\gamma\eta_{t-1}\sum_{k=0}^{t-1}\|P_k^a\|_\infty + \|P_t^a\|_\infty \leq{} &\gamma\eta_{t-1}\sum_{i=0}^{t-1} b_\tau \left( \frac{4\log\left(\frac{6TM|\mathcal{X}[\cup_{k=1}^{K_\omega} Z_k^P]|}{\delta}\right)}{3(1+(1-\gamma)i)} + \sqrt{\frac{8\log\left(\frac{6TM|\mathcal{X}[\cup_{k=1}^{K_\omega} Z_k^P]|}{\delta}\right)}{1+(1-\gamma)i}} \right) \\
&+ b_\tau \left( \frac{4\log\left(\frac{6TM|\mathcal{X}[\cup_{k=1}^{K_\omega} Z_k^P]|}{\delta}\right)}{3(1+(1-\gamma)t)} + \sqrt{\frac{8\log\left(\frac{6TM|\mathcal{X}[\cup_{k=1}^{K_\omega} Z_k^P]|}{\delta}\right)}{1+(1-\gamma)t}} \right).
\end{aligned}
$$

For analyzing the coefficients:

$$
\begin{aligned}
\gamma\eta_{t-1}\sum_{i=0}^{t-1}\frac{1}{1+(1-\gamma)i} + \frac{1}{1+(1-\gamma)t} &\leq \frac{\gamma}{1+(1-\gamma)t}\int_0^t \frac{1}{1+(1-\gamma)x}dx + \frac{1}{1+(1-\gamma)t} \\
&\leq \frac{\gamma}{1+(1-\gamma)t}\cdot\frac{\log(1+(1-\gamma)t)}{1-\gamma} + \frac{1}{1+(1-\gamma)t} \\
&\leq \frac{2\log(1+(1-\gamma)t)}{(1-\gamma)^2 t}.
\end{aligned}
$$

$$
\begin{aligned}
\gamma\eta_{t-1}\sum_{i=0}^{t-1}\frac{1}{\sqrt{1+(1-\gamma)i}} + \frac{1}{\sqrt{1+(1-\gamma)t}} &\leq \frac{\gamma}{1+(1-\gamma)t}\int_0^t \frac{1}{\sqrt{1+(1-\gamma)x}}dx + \frac{1}{\sqrt{1+(1-\gamma)t}} \\
&\leq \frac{\gamma}{1+(1-\gamma)t}\frac{2(\sqrt{1+(1-\gamma)t}-1)}{1-\gamma} + \frac{1}{\sqrt{1+(1-\gamma)t}} \\
&\leq \frac{3}{(1-\gamma)^{\frac{3}{2}}t^{\frac{1}{2}}}.
\end{aligned}
$$

Thus, the whole term satisfies

$$
\gamma\eta_{t-1}\sum_{k=0}^{t-1}\|P_k^a\|_\infty + \|P_t^a\|_\infty \leq \frac{8b_\tau\log(1+(1-\gamma)t)\log\left(\frac{6TM|\mathcal{X}[\cup_{k=1}^{K_\omega} Z_k^P]|}{\delta}\right)}{3(1-\gamma)^2 t} + \frac{6b_\tau\sqrt{2\log\left(\frac{6TM|\mathcal{X}[\cup_{k=1}^{K_\omega} Z_k^P]|}{\delta}\right)}}{(1-\gamma)^{\frac{3}{2}}t^{\frac{1}{2}}}. \tag{71}
$$

Absorbing the constants concludes our proof. □

### F.15. Proof of Lemma F.13

*Proof.* Recall the expression for the ratio $\frac{a_{t,k-1}}{a_{t,k}}$:

$$\frac{a_{t,k-1}}{a_{t,k}} = \frac{1 - \eta_k}{1 - (1 - \gamma)\eta_k} = \frac{(1 - \gamma)(k + 1)}{(1 - \gamma)(k + 1) + \gamma} = \frac{k + 1}{k + 1 + \frac{\gamma}{1-\gamma}}.$$

Using this, we can express $a_{t,k-i}$ for any $1 \leq i < k$ as follows:

$$a_{t,k-i} = a_{t,k} \prod_{j=0}^{i-1} \frac{a_{t,k-j-1}}{a_{t,k-j}} = a_{t,k} \prod_{j=0}^{i-1} \frac{k + 1 - j}{k + 1 - j + \frac{\gamma}{1-\gamma}}.$$

To simplify this product, note that:

$$\frac{k + 1 - j}{k + 1 - j + \frac{\gamma}{1-\gamma}} \leq \frac{k + 1}{k + 1 + \frac{\gamma}{1-\gamma}} \quad \forall j.$$

Thus, we can further bound $a_{t,k-i}$ by:

$$a_{t,k-i} \leq a_{t,k} \prod_{j=0}^{i-1} \frac{k + 1}{k + 1 + \frac{\gamma}{1-\gamma}} = a_{t,k} \left( \frac{k + 1}{k + 1 + \frac{\gamma}{1-\gamma}} \right)^i.$$

Letting $k = t - 1$, we can now bound $\sum_{k=0}^{t-1} a_{t,k}^2$ by considering the following square form:

$$\sum_{k=0}^{t-1} a_{t,k}^2 \leq a_{t,t-1}^2 \left( 1 + \sum_{k=1}^{t-1} \left( \frac{t}{t + \frac{\gamma}{1-\gamma}} \right)^{2k} \right).$$

We analyze this term by considering different $\gamma$.

**Case 1:** $\gamma \geq \frac{1}{2}$

This series is geometric with a ratio $\left( \frac{t}{t+\frac{\gamma}{1-\gamma}} \right)^2 < 1$. The sum of a geometric series can be calculated as:

$$\sum_{k=0}^{\infty} r^k = \frac{1}{1 - r} \quad \text{for } |r| < 1.$$

Applying this to our series:

$$\sum_{k=0}^{t-1} a_{t,k}^2 \leq a_{t,t-1}^2 \left( 1 + \frac{\left( \frac{t}{t+\frac{\gamma}{1-\gamma}} \right)^2}{1 - \left( \frac{t}{t+\frac{\gamma}{1-\gamma}} \right)^2} \right) = a_{t,t-1}^2 \left( 1 + \frac{(1-\gamma)^2 t^2}{\gamma(2t(1-\gamma) + \gamma)} \right).$$

Since $\gamma \geq \frac{1}{2}$, we have:

$$a_{t,t-1}^2 \left( 1 + \frac{(1-\gamma)^2 t^2}{\gamma(2t(1-\gamma) + \gamma)} \right) \leq a_{t,t-1}^2 \left( 1 + \frac{(1-\gamma)^2 t^2}{t(1-\gamma)} \right) = a_{t,t-1}^2 (1 + (1-\gamma)t)$$

Using $a_{t,t-1} = \frac{1}{1+(1-\gamma)t}$ yields that,

$$\sum_{k=0}^{t-1} a_{t,k}^2 \leq \frac{1}{1 + (1 - \gamma)t}. \tag{72}$$

**Case 2:** $\gamma < \frac{1}{2}$

We use $\frac{t}{t+\frac{\gamma}{1-\gamma}} \leq 1$ and $1 - \gamma \geq \frac{1}{2}$ to obtain that:

$$a_{t,t-1}^2 \left(1 + \sum_{k=1}^{t-2}\left(\frac{t}{t + \frac{\gamma}{1-\gamma}}\right)^{2k}\right) \leq a_{t,t-1}^2 \left(1 + \sum_{k=1}^{t-2} 1\right) \leq a_{t,t-1}^2(1 + t).$$

Since $\gamma < \frac{1}{2}$, we have $1 - \gamma > \frac{1}{2}$ and $1 < 2(1 - \gamma)$. Thus

$$a_{t,t-1}^2(1 + t) \leq a_{t,t-1}^2(1 + 2(1 - \gamma)t) \leq 2a_{t,t-1}^2(1 + (1 - \gamma)t).$$

Using $a_{t,t-1} = \frac{1}{1+(1-\gamma)(t-1)}$ yields that,

$$a_{t,t-1}^2 \left(1 + \frac{(1-\gamma)^2 t^2}{\gamma(2(1 - \gamma)t + \gamma)}\right) \leq \frac{2}{1 + (1 - \gamma)t}. \tag{73}$$

Combing Eq. (72) and (73) yields our result. $\qquad\square$

### F.16. Proof to Lemma E.2

*Proof.* We proceed by induction on the epoch index $\tau$. The base case when $\tau = 0$ is straightforward, as the initial approximation $\overline{Q}_0$ can satisfy the desired bound. Our focus is on the inductive step: assuming that the input $\overline{Q}_\tau$ in epoch $\tau$ satisfies

$$\|\overline{Q}_\tau - Q_\omega^*\|_\infty \leq \frac{1}{2^\tau \sqrt{1 - \gamma}} := b_\tau',$$

we aim to show that the output $\overline{Q}_{\tau+1}$ satisfies $\|\overline{Q}_{\tau+1} - Q^*\|_\infty \leq \frac{b_\tau'}{2}$ with probability at least $1 - \frac{\delta}{T}$.

To achieve this, we analyze the variance-reduced update rule, which can be expressed as

$$Q_{t+1} = (1 - \eta_t)Q_t + \eta_t \widehat{\mathcal{F}}_t(Q_t),$$

where $\widehat{\mathcal{F}}_t(Q_t)$ is the variance-reduced Bellman operator defined by

$$\widehat{\mathcal{F}}_t(Q_t) = \widehat{\mathcal{H}}_t(Q_t) - \widehat{\mathcal{H}}_t(\overline{Q}_\tau) + \overline{\mathcal{H}}_{\tau+1}(\overline{Q}_\tau).$$

Here, $\widehat{\mathcal{H}}_t$ is the empirical Bellman operator based on samples at iteration $t$, and $\overline{\mathcal{H}}_{\tau+1}$ is the reference Bellman operator estimated using a larger reference dataset collected in epoch $\tau + 1$.

We additionally define the refined variance-reduced Bellman operator, eliminating the randomness due to sampling:

$$\mathcal{F}(Q_t) = \mathcal{H}(Q_t) - \mathcal{H}(\overline{Q}_\tau) + \overline{\mathcal{H}}_{\tau+1}(\overline{Q}_\tau),$$

where $\mathcal{H}$ is the true Bellman operator. Due to independent and identically distributed sampling, we have $\mathbb{E}[\widehat{\mathcal{F}}_t(Q_t)] = \mathcal{F}(Q_t)$. Moreover, since $\mathcal{H}$ is a $\gamma$-contraction, it follows that $\mathcal{F}$ is also a $\gamma$-contraction mapping.

We introduce a reference $Q$-function $\widetilde{Q}$, defined as the fixed point of $\mathcal{F}$:

$$\widetilde{Q} = \mathcal{F}(\widetilde{Q}) = \mathcal{H}(\widetilde{Q}) - \mathcal{H}(\overline{Q}_\tau) + \overline{\mathcal{H}}_{\tau+1}(\overline{Q}_\tau).$$

This function $\widetilde{Q}$ serves as an intermediary between $Q_M$ and $Q^*$, potentially closer to $Q_M$ than $Q^*$ is.

Our strategy is to bound the error $\|Q_M - Q_\omega^*\|_\infty$ by decomposing it into two components:

$$\|Q_M - Q_\omega^*\|_\infty \leq \underbrace{\|Q_M - \widetilde{Q}\|_\infty}_{=:B_1} + \underbrace{\|\widetilde{Q} - Q_\omega^*\|_\infty}_{=:B_2}.$$

The term $B_1$ quantifies the distance between the iteratively computed $Q_M$ and the reference $\widetilde{Q}$, while $B_2$ measures the distance between $\widetilde{Q}$ and the optimal $Q_\omega^*$.

We aim to bound $B_1$ and $B_2$ separately. Intuitively, $B_1$ will decrease with the number of iterations $M$ within the epoch, and $B_2$ will be controlled by the size of the reference dataset $N_\tau$. We show they can be well bounded when $N_\tau$ and $M$ are properly selected:

**Lemma F.14** (Proof in Appendix F.17). *In the $\tau$-th epoch, with probability at least $1 - \frac{\delta}{2T}$, $B_1$ satisfies*

$$B_1 \le \frac{b_\tau' + B_2}{5},$$

*if $N_\tau = c_1' \frac{4^\tau \log(8T|\mathcal{X}[\cup_{k=1}^{K_\omega} Z_k^P]|)}{(1-\gamma)^2}$ and $M = c_2' \frac{\log(6T|\mathcal{X}[\cup_{k=1}^{K_\omega} Z_k^P]|(1-\gamma)^{-1}\epsilon^{-1})}{(1-\gamma)^3}$ with large enough $c_1'$ and $c_2'$.*

**Lemma F.15** (Proof in Appendix F.18). *In the $\tau$-th epoch, with probability at least $1 - \frac{\delta}{2T}$, $B_2$ satisfies*

$$B_2 \le \frac{b_\tau'}{4}.$$

*if $N_\tau = c_1' \frac{4^\tau \log(8T|\mathcal{X}[\cup_{k=1}^{K_\omega} Z_k^P]|)}{(1-\gamma)^2}$ and $M = c_2' \frac{\log(6T|\mathcal{X}[\cup_{k=1}^{K_\omega} Z_k^P]|(1-\gamma)^{-1}\epsilon^{-1})}{(1-\gamma)^3}$ with large enough $c_1'$ and $c_2'$.*

By combining Lemmas F.14 and F.15, we obtain:

$$\|Q_M - Q_\omega^*\|_\infty \le B_1 + B_2 \le \frac{b_\tau'}{5} + \frac{6B_2}{5} = \frac{b_\tau'}{5} + \frac{6}{5} \cdot \frac{b_\tau'}{4} \le \frac{b_\tau'}{2}.$$

This completes the inductive step and hence the proof of Lemma E.2. $\square$

## F.17. Proof to Lemma F.14

*Proof.* We aim to bound $B_1 = \|Q_M - \widetilde{Q}\|_\infty$. Our approach mirrors the inductive case in the proof of Lemma E.1, with a key difference in the error decomposition.

Consider the difference between the iterates $Q_{t+1}$ and the reference $\widetilde{Q}$:

$$Q_{t+1} - \widetilde{Q} = (1 - \eta_t)Q_t + \eta_t \widehat{\mathcal{F}}_t(Q_t) - \widetilde{Q}.$$

Subtracting and adding $\eta_t \mathcal{F}_t(\widetilde{Q})$, and using $\mathcal{F}(\widetilde{Q}) = \widetilde{Q}$, we can rewrite this as

$$Q_{t+1} - \widetilde{Q} = (1 - \eta_t)(Q_t - \widetilde{Q}) + \eta_t \left( \widehat{\mathcal{F}}_t(Q_t) - \widehat{\mathcal{F}}_t(\widetilde{Q}) \right) + \eta_t \left( \widehat{\mathcal{F}}_t(\widetilde{Q}) - \mathcal{F}(\widetilde{Q}) \right).$$

The first term represents the contraction towards $\widetilde{Q}$. The second is $\gamma$-contractive because

$$\left\| \widehat{\mathcal{F}}_t(Q_t) - \widehat{\mathcal{F}}_t(\widetilde{Q}) \right\|_\infty \le \gamma \left\| Q_t - \widetilde{Q} \right\|_\infty,$$

which is easy to handle, and the third term is an *i.i.d.* zero-mean random variable across iterations $t$, with the magnitude can be bounded by:

$$\|\widehat{\mathcal{F}}_t(\widetilde{Q}) - \mathcal{F}(\widetilde{Q})\|_\infty = \|\widehat{T}_t(\widetilde{Q}) - \widehat{T}_t(\overline{Q}_\tau)\|_\infty + \|T_t(\widetilde{Q}) - T_t(\overline{Q}_\tau)\|_\infty \le 2\gamma \|\widetilde{Q} - \overline{Q}_\tau\|_\infty.$$

Hence, the iteration of $Q_{t+1} - \widetilde{Q}$ mirrors the iteration in Eq. (47). Following the same routine as the proof of Lemma F.9, we establish that, for a step sizes $\eta_t = \frac{1}{1 + (1-\gamma)(t+1)}$, with probability $1 - \frac{\delta}{2T}$,

$$\|Q_M - \widetilde{Q}\|_\infty \le c \left( \frac{1}{(1-\gamma)M} + \frac{1}{(1-\gamma)^{3/2}\sqrt{M}} \right) \log \left( \frac{6TM|\mathcal{X}[\cup_{k=1}^{K_\omega} Z_k^P]|}{\delta} \right) \|\widetilde{Q} - \overline{Q}_\tau\|_\infty,$$

where $c$ is a large enough constant.

By choosing $M = c_2 \dfrac{\log\left(\frac{6T|\mathcal{X}[\cup_{k=1}^{K_\omega} Z_k^P]|}{(1-\gamma)\delta}\right)}{(1-\gamma)^3}$ with sufficiently large $c_2$, we can ensure that the right-hand side is less than $\frac{1}{5}\|\widetilde{Q} - \overline{Q}_\tau\|_\infty$.

Observing that

$$\|\widetilde{Q} - \overline{Q}_\tau\|_\infty \leq \|\widetilde{Q} - Q^*\|_\infty + \|\overline{Q}_\tau - Q_\omega^*\|_\infty = B_2 + b_\tau',$$

we can conclude

$$B_1 = \|Q_M - \widetilde{Q}\|_\infty \leq \frac{1}{5}\|\widetilde{Q} - \overline{Q}_\tau\|_\infty \leq \frac{1}{5}(B_2 + b_\tau').$$

This establishes the desired bound on $B_1$, completing the proof of Lemma F.14. $\qquad\square$

### F.18. Proof to Lemma F.15

*Proof.* Our goal is to bound the term $B_2 = \|\widetilde{Q} - Q_\omega^*\|_\infty$, where $\widetilde{Q}$ is the fixed point of the operator $\mathcal{F}$ defined as:

$$\widetilde{Q} = \mathcal{F}(\widetilde{Q}) = \mathcal{H}(\widetilde{Q}) - \mathcal{H}(\overline{Q}_\tau) + \overline{\mathcal{H}}_{\tau+1}(\overline{Q}_\tau).$$

Recall that $Q_\omega^* = \mathcal{H}(Q_\omega^*)$. Thus, $\widetilde{Q}$ can be viewed as the fixed point of a Bellman operator with a perturbed reward function, where the perturbation magnitude is $-\mathcal{H}(\overline{Q}_\tau) + \overline{\mathcal{H}}_{\tau+1}(\overline{Q}_\tau)$. We seek to show that when the population size for estimating the reference Bellman operator is large enough, the term $-\mathcal{H}(\overline{Q}_\tau) + \overline{\mathcal{H}}_{\tau+1}(\overline{Q}_\tau)$ converges to 0 and $\mathcal{F}$ converges to the actual Bellman operator $\mathcal{H}$, allowing $\|\widetilde{Q} - Q_\omega^*\|_\infty \leq \frac{b_\tau'}{4}$. Essentially, the error $B_2$ comes from the reward function perturbation $\overline{\mathcal{H}}_{\tau+1}(\overline{Q}_\tau) - \mathcal{H}(\overline{Q}_\tau)$. In the first step, we bound the reward perturbation $\Delta r = -\mathcal{H}(\overline{Q}_\tau) + \overline{\mathcal{H}}_{\tau+1}(\overline{Q}_\tau)$ as follows:

$$
\begin{aligned}
|\Delta r| = |\overline{\mathcal{H}}_{\tau+1}(\overline{Q}_\tau) - \mathcal{H}(\overline{Q}_\tau)| &\leq |\overline{\mathcal{H}}_{\tau+1}(\overline{Q}_\tau) - \overline{\mathcal{H}}_{\tau+1}(Q_\omega^*) + \mathcal{H}(Q_\omega^*) - \mathcal{H}(Q_\omega^*) + \overline{\mathcal{H}}_{\tau+1}(Q_\omega^*) - \mathcal{H}(\overline{Q}_\tau)| \\
&\leq |\overline{\mathcal{H}}_{\tau+1}(\overline{Q}_\tau) - \overline{\mathcal{H}}_{\tau+1}(Q_\omega^*)| + |\mathcal{H}(\overline{Q}_\tau) - \mathcal{H}(Q_\omega^*)| + |\overline{\mathcal{H}}_{\tau+1}(Q_\omega^*) - \mathcal{H}(Q_\omega^*)| \\
&\leq 2\gamma\|\overline{Q}_\tau - Q^*\|_\infty \cdot \mathbf{1} + |\overline{\mathcal{H}}_{\tau+1}(Q_\omega^*) - \mathcal{H}(Q_\omega^*)|.
\end{aligned}
$$

The first terms can be directly bounded through Hoeffding's inequality. Similar to the proof of Lemma F.10, with probability at least $1 - \frac{\delta}{4T}$,

$$2\gamma\|\overline{Q}_\tau - Q_\omega^*\|_\infty \cdot \mathbf{1} \leq 4b_\tau'\sqrt{\frac{\log(8T|\mathcal{X}[\cup_{k\in[K_\omega]} Z_k^P]|/\delta)}{N_\tau}}.$$

The second term can be bounded by applying the result of Eq. (70)[2] in Lemma F.11, we have that with probability at least $1 - \frac{\delta}{4T}$,

$$|\overline{\mathcal{H}}_{\tau+1}(Q_\omega^*) - \mathcal{H}(Q_\omega^*)| \leq c\left\{\frac{\log(8T|\mathcal{X}[\cup_{k=1}^{K_\omega} Z_k^P]|/\delta) \cdot \mathbf{1}}{N_\tau(1-\gamma)} + \sqrt{\frac{\sigma_\mathcal{E}^2 \log(8T|\mathcal{X}[\cup_{k=1}^{K_\omega} Z_k^P]|/\delta)}{N_\tau}}\right\},$$

where $c$ is a large enough constant.

Combining these two conditions, we can derive that, with probability at least $1 - \frac{\delta}{2T}$,

$$|\Delta r| \leq c\left\{\frac{\log(8T|\mathcal{X}[\cup_{k=1}^{K_\omega} Z_k^P]|/\delta)}{N_\tau(1-\gamma)} \cdot \mathbf{1} + (b_\tau'\mathbf{1} + \sigma_\mathcal{E})\sqrt{\frac{\log(8T|\mathcal{X}[\cup_{k=1}^{K_\omega} Z_k^P]|/\delta)}{N_\tau}}\right\}, \tag{74}$$

where $c$ is a large enough constant.

---

[2]The only difference is to applying the bound on the vector $|\overline{\mathcal{H}}_{\tau+1}(Q_\omega^*) - \mathcal{H}(Q_\omega^*)|$, instead of the infinity norm.

Now we have already obtained the element-wise error bound of the reward function, which is the only difference for inducing $\widetilde{Q}$ and $Q^*$. Applying Lemma F.6 with Eq. (74), we have:

$$|\widetilde{Q} - Q_\omega^*| \leq \max\{\Delta_1, \Delta_2\},$$

where

$$\Delta_1 = c(I - \gamma P_\omega^{\pi_\omega^*})^{-1}\left\{\frac{\log(8T|\mathcal{X}[\cup_{k=1}^{K_\omega} Z_k^P]|/\delta)}{N_\tau(1-\gamma)} \cdot \mathbf{1} + (b_\tau'\mathbf{1} + \sigma_\mathcal{E})\sqrt{\frac{\log(8T|\mathcal{X}[\cup_{k=1}^{K_\omega} Z_k^P]|/\delta)}{N_\tau}}\right\}, \tag{75}$$

$$\Delta_2 = c(I - \gamma P_\omega^{\tilde{\pi}_\omega^*})^{-1}\left\{\frac{\log(8T|\mathcal{X}[\cup_{k=1}^{K_\omega} Z_k^P]|/\delta)}{N_\tau(1-\gamma)} \cdot \mathbf{1} + (b_\tau'\mathbf{1} + \sigma_\mathcal{E})\sqrt{\frac{\log(8T|\mathcal{X}[\cup_{k=1}^{K_\omega} Z_k^P]|/\delta)}{N_\tau}}\right\}. \tag{76}$$

**Bounding $\Delta_1$:** For bounding $\Delta_1$, the key is to bound $(I - \gamma P_\omega^{\pi_\omega^*})^{-1}\sigma_\mathcal{E}$. We have:

$$\|(I - \gamma P_\omega^{\pi_\omega^*})^{-1}\sigma_\mathcal{E}\|_\infty = \gamma^2\|(I - \gamma P_\omega^{\pi_\omega^*})^{-1}\text{Var}_{P_\omega^*}(V_\omega^*)\|_\infty \leq \frac{\sqrt{2}}{(1-\gamma)^{\frac{3}{2}}}, \tag{77}$$

where the second inequality comes from Eq. (45) in Appendix F.6. By using $\|(I - \gamma P_\omega^{\pi_\omega^*})^{-1}\|_\infty \leq \frac{1}{1-\gamma}$ and Eq. (77), we have:

$$\begin{aligned}
\Delta_1 &\leq c\left\{\frac{\log(8T|\mathcal{X}[\cup_{k=1}^{K_\omega} Z_k^P]|/\delta)}{N_\tau(1-\gamma)^2} \cdot \mathbf{1} + (b_\tau' + \frac{\sqrt{2}}{(1-\gamma)^{\frac{3}{2}}})\sqrt{\frac{\log(8T|\mathcal{X}[\cup_{k=1}^{K_\omega} Z_k^P]|/\delta)}{N_\tau}} \cdot \mathbf{1}\right\} \\
&\leq cb_\tau'\left\{\frac{2^\tau \log(8T|\mathcal{X}[\cup_{k=1}^{K_\omega} Z_k^P]|/\delta)}{N_\tau(1-\gamma)^{\frac{3}{2}}} + \frac{2^{\tau+2}}{1-\gamma}\sqrt{\frac{\log(8T|\mathcal{X}[\cup_{k=1}^{K_\omega} Z_k^P]|/\delta)}{N_\tau}}\right\} \cdot \mathbf{1},
\end{aligned} \tag{78}$$

where the second inequality is obtained by using $b_\tau' = \frac{2^{-\tau}}{\sqrt{1-\gamma}}$.

**Bounding $\Delta_1$:** For bounding $\Delta_2$, the difference comes from bounding $(I - \gamma P_\omega^{\tilde{\pi}_\omega^*})^{-1}\sigma_\mathcal{E}$. We define $\sigma_{\tilde{\mathcal{E}}}^2$ as the variance of induced by Q function $\widetilde{Q}$ and policy $\tilde{\pi}^*$. Then, we have:

$$\begin{aligned}
\|(I - \gamma P_\omega^{\tilde{\pi}_\omega^*})^{-1}\sigma_\mathcal{E}\|_\infty &= \|(I - \gamma P_\omega^{\tilde{\pi}_\omega^*})^{-1}(\sigma_{\tilde{\mathcal{E}}} - \sigma_{\tilde{\mathcal{E}}} + \sigma_\mathcal{E})\|_\infty \\
&= \|(I - \gamma P_\omega^{\tilde{\pi}_\omega^*})^{-1}\sigma_{\tilde{\mathcal{E}}}\|_\infty + \|(I - \gamma P_\omega^{\tilde{\pi}_\omega^*})^{-1}(\sigma_{\tilde{\mathcal{E}}} - \sigma_\mathcal{E})\|_\infty \\
&\leq \frac{3}{(1-\gamma)^{\frac{3}{2}}} + \frac{\|\widetilde{Q} - Q_\omega^*\|_\infty}{1-\gamma}.
\end{aligned} \tag{79}$$

Combining Eq. (79) with Eq. (76), we have:

$$\begin{aligned}
\Delta_2 \leq &cb_\tau'\left\{\frac{2^\tau \log(8T|\mathcal{X}[\cup_{k=1}^{K_\omega} Z_k^P]|/\delta)}{N_\tau(1-\gamma)^{\frac{3}{2}}} + \frac{2^{\tau+2}}{1-\gamma}\sqrt{\frac{\log(8T|\mathcal{X}[\cup_{k=1}^{K_\omega} Z_k^P]|/\delta)}{N_\tau}}\right\} \cdot \mathbf{1} \\
&+ c\|\widetilde{Q} - Q_\omega^*\|_\infty\sqrt{\frac{\log(8T|\mathcal{X}[\cup_{k=1}^{K_\omega} Z_k^P]|/\delta)}{(1-\gamma)^2 N_\tau}} \cdot \mathbf{1}.
\end{aligned} \tag{80}$$

Combining Eq. (80) and (78), we can conclude that:

$$\|\widetilde{Q} - Q_\omega^*\|_\infty \leq cb_\tau'\left\{\frac{2^\tau \log(8T|\mathcal{X}[\cup_{k=1}^{K_\omega} Z_k^P]|/\delta)}{N_\tau(1-\gamma)^{\frac{3}{2}}} + \frac{2^{\tau+2}}{1-\gamma}\sqrt{\frac{\log(8T|\mathcal{X}[\cup_{k=1}^{K_\omega} Z_k^P]|/\delta)}{N_\tau}}\right\}$$

$$+ c\|\widetilde{Q} - Q_\omega^*\|_\infty \sqrt{\frac{\log(8T|\mathcal{X}[\cup_{k=1}^{K_\omega} Z_k^P]|/\delta)}{(1-\gamma)^2 N_\tau}}. \tag{81}$$

Using a sample size $N_\tau = c_3 4^\tau \log\left(6T|\mathcal{X}[\cup_{k=1}^{K_\omega} Z_k^P]|\right)(1-\gamma)^{-2}$ with large enough $c_3$, we can guarantee:

$$c\sqrt{\frac{4^\tau \log(8T|\mathcal{X}[\cup_{k=1}^{K_\omega} Z_k^P]|/\delta)}{N_\tau}} \leq \frac{1}{2}, \tag{82}$$

$$\frac{c \cdot 2^\tau \log(8T|\mathcal{X}[\cup_{k=1}^{K_\omega} Z_k^P]|/\delta)}{(1-\gamma)^{\frac{3}{2}} N_\tau} \leq \frac{1}{8}, \tag{83}$$

$$c\sqrt{\frac{4^{\tau+1} \log(8T|\mathcal{X}[\cup_{k=1}^{K_\omega} Z_k^P]|/\delta)}{(1-\gamma)^2 N_\tau}} \leq \frac{1}{8}. \tag{84}$$

Substituting condition (82)-(84) into Eq. (81) leads to $\|\widetilde{Q} - Q_\omega^*\|_\infty \leq \frac{b_\tau'}{4}$. This concludes our proof. □

## G. Discussion on the Cost-Optimal Sampling Problem

### G.1. Relating the Cost-Optimal Sampling Problem to Graph Coloring

The Graph Coloring Problem (GCP) is defined as follows: Given a graph $G = (V, E)$, where $V$ is a set of vertices and $E$ is a set of edges, the task is to assign a color to each vertex such that no two adjacent vertices (i.e., vertices connected by an edge) share the same color. The objective is to find the optimal coloring scheme that minimizes the number of colors used, known as the *chromatic number* of the graph (Erdős & Hajnal, 1966).

In the Cost-Optimal Sampling Problem (COSP), components with scopes $Z_1^P, Z_2^P, \ldots, Z_{K_\omega}^P$ must be divided into different groups, where the scopes of components in the same subset do not overlap (i.e., $Z_i^P \cap Z_j^P = \emptyset$ for any $i$, $j$ in the same group). This problem can be modeled as a variant of the GCP. Specifically, each component $i$ represents a vertex in a graph, and two vertices $i$ and $j$ are connected by an edge if their scopes overlap, i.e., $Z_i^P \cap Z_j^P \neq \emptyset$. Assigning a color to a vertex corresponds to assigning the component to a group. Since components in the same subset must have disjoint scopes, no two connected vertices (representing components with overlapping scopes) can share the same color. The goal is to find a coloring scheme that minimizes the total costs across all groups, where the cost of each subset is determined by the scope with largest factor set space $|\mathcal{X}[Z_i^P]|$ in that group.

### G.2. Proof for NP-Completeness of the COSP

NP-completeness is used to characterize a subset of problems that are computationally hard to solve. Famous NP-complete problems include the Traveling Salesman Problem (TSP) (Lin, 1965), the Knapsack Problem (Kellerer et al., 2004), the Hamiltonian Path Problem (Gary & Johnson, 1979), and the Satisfiability (SAT) Problem (Schaefer, 1978).

A problem is classified as *NP-complete* if it satisfies two conditions:

- *In NP*: The problem belongs to the class NP, meaning that given a proposed solution, we can verify its feasibility in polynomial time.

- *NP-hard*: The problem is at least as hard as any other problem in NP. This means that any problem in NP can be transformed or reduced to an NP-hard problem in polynomial time, then we could solve the NP-hard problem to get the solution of any NP problem.

Graph coloring has been shown to be NP-complete (Karp, 2010), as it is computationally difficult to find the minimum number of colors for an arbitrary graph. We now show that the COSP is NP-complete by proving: (1) it belongs to NP, and (2) it is at least as hard as the NP-complete GCP.

### G.2.1. THE COSP BELONGS TO NP

In the COSP, given a solution (a partition of components into groups and their associated costs), we can verify its feasibility in polynomial time by:

- Verifying that the scopes within each subset are disjoint, which can be done by checking pairwise intersections within each group. This can be done in $\mathcal{O}(K_\omega^2(n + m))$ time.

- Calculating the total cost of a partition by identifying the largest scope size in each subset and summing these values, which can also be computed in polynomial time, specifically $\mathcal{O}(K_\omega)$.

### G.2.2. THE COSP IS NP-HARD

We prove NP-hardness by reducing the GCP to the COSP. We demonstrate that for any instance of the GCP, a corresponding instance of the COSP can be constructed, and based on the optimal solution of the COSP, we can derive the optimal solution to the GCP.

Given any instance of the GCP with a graph $G = (V, E)$, we construct a corresponding instance of the COSP as follows:

- *Step 1. Assign unique base dimensions for each vertex*: Let $|V|$ be the number of vertices, we initial $|V|$ components with the scope for each component $i \in [|V|]$ satisfies $Z_i^P = \{i\}$. This ensures that each component starts with a unique, non-overlapping dimension.

- *Step 2. Create shared dimensions for edges*: For each edge $(v_i, v_j) \in E$, we introduce a new and unique dimension $d_{i,j}$ that will belong to both $Z_i^P$ and $Z_j^P$. Specifically, we updates the scopes of components $i$ and $j$ by

$$Z_i^P = Z_i^P \cup \{d_{i,j}\}, \quad Z_j^P = Z_j^P \cup \{d_{i,j}\}.$$

  This ensures that for every edge in the graph, the corresponding components have overlapping scopes, mimicking the adjacency constraint in the GCP.

- *Step 3. Enforce disjointness for non-adjacent vertices*: If there is no edge between two vertices $v_i$ and $v_j$, the corresponding scopes should remain disjoint, i.e., $Z_i^P \cap Z_j^P = \emptyset$. This property is automatically maintained because shared dimensions are only introduced for pairs of vertices that are connected by an edge.

We provide an example to illustrate the construction process. Consider a small example graph $G = (V, E)$ with three vertices $V = \{v_1, v_2, v_3\}$ and edges $E = \{(v_1, v_2), (v_2, v_3)\}$. The construction process is:

- *Step 1. Assign unique base dimensions for each vertex*: $Z_1^P = \{1\}$, $Z_2^P = \{2\}$ and $Z_3^P = \{3\}$.

- *Step 2. Create shared dimensions for edges*: For edge $(v_1, v_2)$, we introduce $d_{1,2} = 4$, so $Z_1^P = \{1, 4\}, \quad Z_2^P = \{2, 4\}$. Also, for edge $(v_2, v_3)$, we introduce $d_{2,3} = 5$, so $Z_2^P = \{2, 4, 5\}, \quad Z_3^P = \{3, 5\}$.

- *Step 3. Resulting scopes*: the resulting scopes are:

$$Z_1^P = \{1, 4\}, \quad Z_2^P = \{2, 4, 5\}, \quad Z_3^P = \{3, 5\}.$$

The objective of the COSP is to partition these components into disjoint groups (sets of components with no overlapping scopes) while minimizing the total sampling cost. Since the sampling cost for each component is set to 1, minimizing the total sampling cost is equivalent to minimizing the number of groups (colors) used, as each subset contributes 1 to the total cost. Therefore, solving the COSP on this constructed instance is equivalent to solving the GCP.

### G.2.3. NP-COMPLETENESS

Since the GCP can be reduced to the COSP in polynomial time, and solving the COSP provides a solution to the GCP, the COSP is NP-hard. Furthermore, since the COSP belongs to NP, it is NP-complete.

### G.3. Computation Complexity

Intuitively, the SOSP involves grouping the factors of the MDP to minimize the total sampling cost. While the problem may appear combinatorial, it is computationally efficient in our setting for two reasons: (1) the number of factors $K$ is typically small (on the order of $\log(|\mathcal{S}||\mathcal{A}|)$), (2) there exist established exact and approximate algorithms to solve such problems efficiently.

Specifically, the problem can be formulated as an integer program of size $\mathcal{O}(K)$, solvable by modern solvers like Gurobi and CPLEX when $K$ is moderate (e.g., $K \leq 500$). Furthermore, as discussed in ealier subsections, it can be reduced to the classical weighted GCP, for which many scalable algorithms exist. In practice, even problems with thousands of nodes (e.g., $K = 5000$) can be solved to near-optimality within seconds (Shen et al., 2022; Dokeroglu & Sevinc, 2021).

## H. Numerical Experiments

We now present numerical experiments to evaluate our proposed model-based and model-free algorithms based on approximate factorization with comparisons to state-of-the-art RL algorithms. We focus on two types of tasks: (i) synthetic MDP tasks and (ii) an electricity storage control problem in power system operation.

### H.1. Synthetic MDP Tasks

The two synthetic MDP tasks that we consider are distinguished by their transition kernels: one with a perfectly factorizable transition kernel and another with an imperfectly factorizable one. For the former task, we consider a three-dimensional state space, where each state is represented as $s = (s[1], s[2], s[3])$, along with a one-dimensional action $a$. The transitions for $s[1]$ and $s[2]$ are independent of other components, while $s[3]$ is influenced by both its own state and the action $a$. Each substate and the action space consists of 5 discrete elements. Both the transition kernel and the reward function are randomly generated. We conduct 50 trials to compare our model-based and model-free algorithms (VRQL-AF) with approximate factorization against traditional model-based RL and a variance-reduced Q-learning method without factorization. Figures. 3(a) and 3(b) illustrate the $l_\infty$-norm of the $Q$-function error for both settings, respectively, as a function of the number of samples. As shown in the two figures, our approximate factorization methods (depicted in red) consistently exhibit lower $Q$-function error across varying sample sizes compared to the vanilla RL approaches (depicted in blue). This demonstrates that our approach significantly reduces sample complexity, showcasing superior sample efficiency. These findings underline the effectiveness of our factorization-based framework in enhancing learning efficiency for multi-dimensional MDPs.

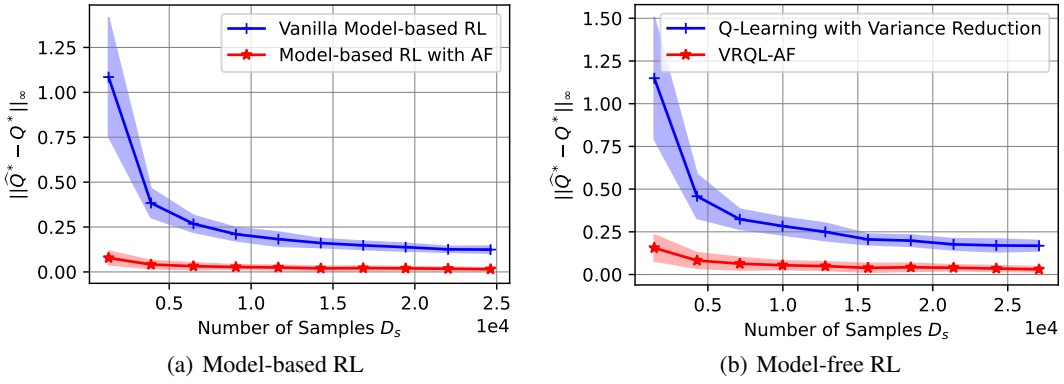

(a) Model-based RL             (b) Model-free RL

*Figure 3.* Performance on Perfectly Factorizable MDPs

The second task involves MDPs with imperfectly factorizable transition kernels. Here, the state consists of 4 substates, each taking one of 5 possible values. To investigate the trade-off between sample complexity and estimation error, we evaluate three factorization schemes: (1) full factorization with $K_\omega = 4$, (2) partial factorization with $K_\omega = 2$, and (3) no factorization with ($K_\omega = 1$), corresponding to vanilla RL. We conduct 200 trials to compare their performance with different sample amounts. As shown in Figure 4, our model-based and model-free RL algorithms exhibit distinct convergence behaviors under these different factorization levels. The results indicate two key breaking points $(D_1, \mathcal{E}_1)$ and $(D_2, \mathcal{E}_2)$ in performance trade-offs (illustrated in Figure 4(b)), which inform the optimal factorization choice depending on the sample

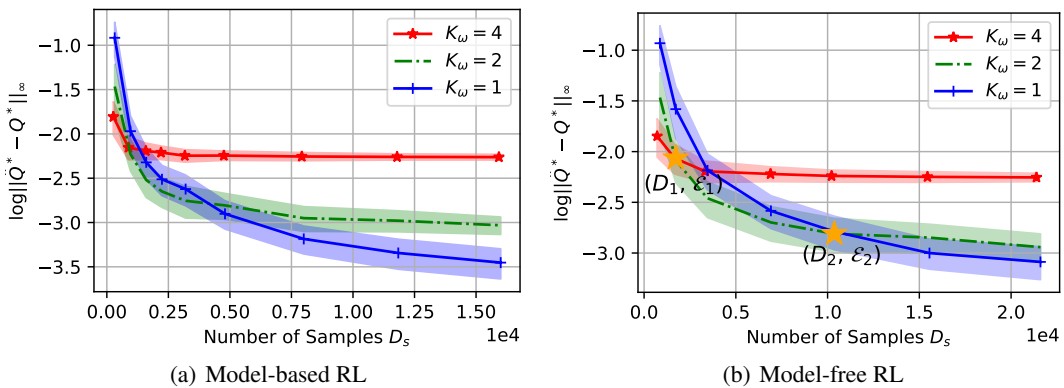

*Figure 4.* Performance on Imperfectly Factorizable MDPs

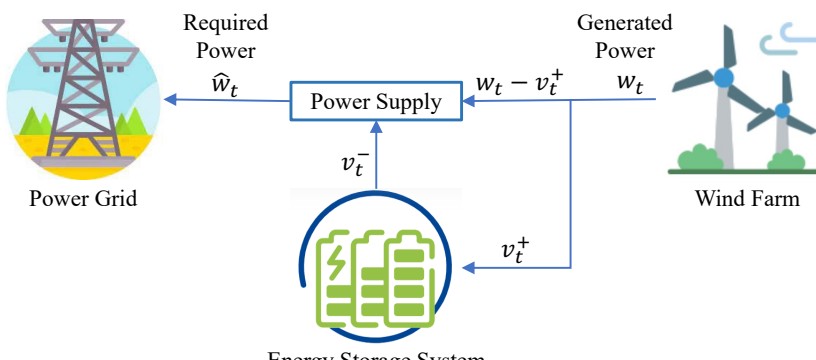

*Figure 5.* Wind Farm Storage Control. (Adapted from (Lu et al., 2024))

availability. Specifically, in the small sample range with $D_s \leq D_1$, the full factorization scheme ($K_\omega = 4$) converges quickly but has a higher asymptotic error, making it suitable when the samples are limited or the required accuracy is not high. In the intermediate sample range with $D_1 < D_s \leq D_2$, the partial factorization scheme ($K_\omega = 2$) provides a balanced trade-off, achieving moderate convergence speed and a lower asymptotic error. This scheme is advantageous when a compromise between sample efficiency and accuracy is needed. In the large sample range with $D_s > D_2$, the no factorization scheme ($K_\omega = 1$) results in the smallest asymptotic error but converges slowly. This approach is best suited when very high precision is required, and a large sample size is available. Therefore, it highlights the benefits of selecting an appropriate factorization level to match the sample size and desired accuracy. The flexibility in choosing among different factorization strategies enables the optimization of performance based on specific requirements, making our approach adaptable to various settings.

### H.2. Wind Farm Storage Control Problem

We next evaluate the performance of our proposed approach on the wind farm storage control problem introduced in Section B. As depicted in Figure 5, this problem involves managing energy storage systems to mitigate mismatches between variable wind power generation and demand, thereby minimizing penalty costs.

The control actions consist of charging and discharging decisions influenced by real-time wind generation and unit penalty prices. Constraints include storage capacity limits, prohibiting simultaneous charging and discharging, and lossy energy dynamics. For a detailed model description, please refer to Appendix I. The total state-action space size in this problem is $10^4$.

Figure 6 illustrates the performance of our approach, where we decompose the MDP into three components as described earlier. Specifically, Figure 6(a) demonstrates that our approximate factorization-based algorithm converges significantly faster than the vanilla model-based approach by leveraging the model's structural properties. Additionally, our method

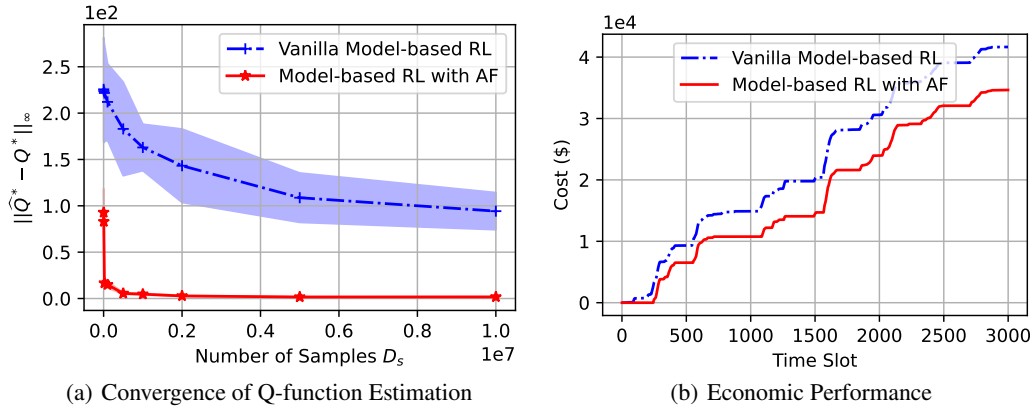

(a) Convergence of Q-function Estimation        (b) Economic Performance

*Figure 6.* Performance on Storage Control

exhibits considerably lower variance, indicating more stable performance for different sample conditions. Figure 6(b) presents the economic benefits of our algorithm using a sample size of 60,000 data points (typically taking the wind farm a year to collect), evaluating cumulative system penalty costs over 3,000 time slots. The results show that our approach reduces costs by 19.3% compared to the vanilla model-based method, highlighting its enhanced effectiveness in managing complex problems with large-scale state spaces.

# I. Detailed Model of Wind Farm-equipped Storage Control

## I.1. System Model

Consider a wind farm tasked with supplying power to the grid under an electricity supply contract. The predetermined supply quantity of wind power at any given time $t$ is denoted by $\widehat{w}_t$. This value is established based on wind power forecasts, which can be made from a minute to a day in advance, either by the wind farm itself or the Independent System Operator (ISO). As time progresses, the actual wind power generation $w_t$ is sequentially disclosed in real-time. The wind farm is equipped with a storage system, allowing it to either store wind energy, denoted as $u_t^+$, or release stored energy, denoted as $u_t^-$ into the grid at any time $t$. As a result, the total power delivered to the grid, represented by $g_t$, is determined by

$$g_t = w_t + v_t^- - v_t^+.$$

When the mismatch exists between commitment $\widehat{w}_t$ and the delivered power $g_t$, the wind farm will be charged with a penalty cost $c_t(\widehat{w}_t, g_t)$ as follows:

$$c_t(\widehat{w}_t, g_t) = p_t^+ \max(g_t - \widehat{w}_t, 0) + p_t^- \max(\widehat{w}_t - g_t, 0),$$

where $p_t^+$ and $p_t^-$ denote the unit penalty costs for wind generation surplus and shortage at time $t$, respectively.

The wind farm targets to minimize the accumulated mismatch penalty costs across all $T$ time slots by reasonably controling the storage system. Mathematically, the storage control problem can be formulated as follows:

$$\textbf{(P1)} \quad \min_{v_t^+, v_t^-, \forall t} \quad \sum_{t=1}^{T} c_t(\widehat{w}_t, g_t) \tag{85}$$

$$s.t. \quad g_t = w_t + v_t^- - v_t^+, \forall t, \tag{86}$$

$$SoC_1 = \frac{C}{2}, \tag{87}$$

$$SoC_{t+1} = SoC_t + \eta^+ v_t^+ - \eta^- v_t^-, \forall t, \tag{88}$$

$$\eta^+ v_t^+ \leq C - SoC_t, \forall t, \tag{89}$$

$$\eta^- v_t^- \leq SoC_t, \forall t, \tag{90}$$

$$v_t^+ \leq w_t, \forall t, \tag{91}$$

$$v_t^+, v_t^- \geq 0, \forall t, \tag{92}$$

$$v_t^+ v_t^- = 0, \forall t. \tag{93}$$

In the optimization, the decision variables at time $t$ include:

- $v_t^+$: generated wind power which is charged to the energy storage;

- $v_t^-$: discharged energy from the storage to the grid;

And the other functions, latent variables, and parameters include:

- $c_t(\cdot)$: total penalty cost at time $t$;

- $\widehat{w}_t$: committed wind power supply at time $t$;

- $w_t$: wind power generation at time $t$;

- $T$: total duration of storage control decisions;

- $p_t^+, p_t^-$: unit grid penalty prices for power generation shortage and surplus at time $t$, respectively;

- $g_t$: actual supplied energy at time $t$;

- $SoC_t$: state-of-charge (SOC) of storage at time $t$;

- $C$: energy storage capacity;

- $\eta^+, \eta^-$: charging and discharging efficiencies of storage;

Constraint (86) represents the delivered power; constraints (87) and (88) characterize the dynamics of storage; and constraints (89) and (90) denote the storage capacity limits. Constraint (91) and (92) show the upper and lower limits of storage control actions, and constraint (93) indicates that the storage cannot be charged and discharged simultaneously.

Due to the inherent stochasticity of penalty prices and renewable energy production, it is impractical to obtain the optimal future storage control decisions. Therefore, in practice, we often consider sequential storage control. Specifically, at each time $t$, we determine the current storage control actions $v_t^+, v_t^-, c_t$ based on the currently available information. Consequently, we establish the following online storage control problem at time $t$ as follows:

$$\textbf{(P2)} \quad \min_{v_t^+, v_t^-} \quad c_t(\widehat{w}_t, g_t) + \sum_{\tau=t+1}^{\infty} \gamma^{\tau-t} \mathbb{E}(c_\tau(\widehat{w}_\tau, g_\tau)) \tag{94}$$

$$s.t. \quad g_t = w_t + v_t^- - v_t^+, \tag{95}$$

$$SoC_{t+1} = SoC_t + \eta^+ v_t^+ - \eta^- v_t^-, \tag{96}$$

$$\eta^+ v_t^+ \leq C - SoC_t, \tag{97}$$

$$\eta^- v_t^- \leq SoC_t, \tag{98}$$

$$v_t^+ \leq w_t, \tag{99}$$

$$v_t^+, v_t^- \geq 0, \tag{100}$$

$$v_t^+ v_t^- = 0. \tag{101}$$

### I.2. Markov Decision Process Modeling

We highlight that problem **(P2)** can be transformed into MDP in the following manner:

**Markov Decision Process** $(\mathcal{S}, \mathcal{A}, \mathcal{P}, \mathcal{R})$:

- States $\mathcal{S}$: Any state $s \in \mathcal{S}$ is composed of the penalty prices $p^+, p^-$, the committed and real wind power generation $\widehat{w}$ and $w$, and state-of-charge $SoC$. Formally, $s = (p^+, p^-, \widehat{w}, w, SoC)$;

- Actions $\mathcal{A}$: Any action $a \in \mathcal{A}$ is composed of the charge amount $v^+$ and discharge amount $v^-$. Formally, $a = (v^+, v^-)$;

- Transition probability $\mathcal{P}$: $\mathcal{P}$ is the transiting probability matrix $\mathcal{P}_a = \{\mathbf{Pr}(s_{t+1} = s'|s_t = s, a_t = a), \forall s, s' \in \mathcal{S}, a \in \mathcal{A}\}$, which includes the probability of transiting from state $s$ to $s'$ with action $a$ for all $s$, $s'$ and $a$;

- Reward $\mathcal{R}$: $\mathcal{R}$ is the immediate reward (penalty in our case) after transiting from state $s$ to state $s'$ due to action $a$, i.e., $\mathcal{R} = \{r(s, a), \forall s, a\}$. Specifically, in the one-shot decision problem, the penalty $r(s, a)$ equals the negative of the penalty, i.e.,

$$r(s, a) = p^+ \max(w - \widehat{w}, 0) + p^- \max(\widehat{w} - w, 0).$$

We can observe that, for the storage control problem, the state space $\mathcal{S}$ and the action space $\mathcal{A}$ are known. The reward $\mathcal{R}$ is also known once the state $s$ and action $a$ are decided. The only unknown comes from the transition probability $\mathcal{P}$. However, some important observations for $\mathcal{P}$ can simplify the problem. Specifically, we can divide the state variables $s = (p^+, p^-, \widehat{w}, w, SoC)$ into one deterministic state and several random states. The deterministic state is $SoC$, which can be determined following Eq. (96) without any uncertainty. And the random states include $p^+, p^-, \widehat{w}, w$, which are fully random[3].

In our numerical study, we assume $p = p^+ = p^-$, and observe that only $\Delta = \widehat{w} - w$ exists in the reward function. Therefore, the state can be rewritten as $\mathcal{S} = (p, \Delta w, SoC)$. For the factorization scheme $\omega$, we use the following factorization to the transition kernel:

$$\widehat{P}(p_{t+1}, \Delta_{t+1}, SoC_{t+1}|p_t, \Delta_t, SoC_t, a_t) = \widehat{P}(p_{t+1}|p_t)\widehat{P}(\Delta_{t+1}|\Delta_t)\widehat{P}(SoC_{t+1}|SoC_t, a_t).$$

### I.3. Parameter Settings

In the numerical study, we utilized the California aggregate wind power generation dataset from CAISO (California ISO, 2021) containing predicted and real wind power generation data with a 5-minute resolution spanning from January 2020 to December 2020. The penalty price equals the average electricity price of CASIO (California ISO, 2021) with the matching resolution and periods. We set $C = 500$ kWh, $\gamma = 0.9$. The discretization levels of $p$, $\Delta w$ and $SoC$ are set to be 8, 8, 50, respectively. The action set includes 3 discretized choices: charging (discharging) to satisfy $100\%$, $50\%$, and $0\%$ of energy mismatches, respectively.

---

[3]Note that, the committed wind power $\widehat{w}$ is essentially not random. However, only the generation mismatch $w - \widehat{w}$ exists in our problem, and such mismatch can be regarded as a random variable.

