# OpenReview forum: "Overcoming the Curse of Dimensionality in Reinforcement Learning Through Approximate Factorization"
_ICML.cc/2025/Conference — ICML 2025 poster_

### Official Review · Reviewer_xjwC · 2025-02-24

**Overall Recommendation:** 4

**Summary:**

After rebuttal: Thanks for addressing my questions! I'll keep my rating.

===

This paper considers FMDP and provides a powerful framework to tackle the curse of dimensionality in reinforcement learning (RL) by breaking down high-dimensional MDPs into smaller, independently evolving components, i.e., using factored MDP.

Tis paper introduces approximate factorization, which extends FMDPs to handle imperfectly factored models. Additionally, a graph-coloring-based optimal synchronous sampling strategy is proposed to enhance sample efficiency.

A model-based and the first model-free variance-reduced Q-L algorithm for FMDP are proposed and their theoretical guarantees are shown. They both outcompete the STOA.

Numerical results are provided in the appendix.

**Claims And Evidence:**

Yes they are

**Essential References Not Discussed:**

no

**Experimental Designs Or Analyses:**

Numerical results are included in the appendix. Synthetic MDP tasks and an electricity storage control problem are considered and the STOA algorithms are compared.

**Methods And Evaluation Criteria:**

Yes they do.

**Other Comments Or Suggestions:**

It might be helpful to include a proof sketch to summarize the main techniques and novelty of the proof in the main text.

Can the algorithm be extended to the case where factorization is unknown?

Does the algorithm require factorization bias term Delta as an input? If so, is it possible to generalize to the unknown bias case?

Can you comment on the computation complexity of the algorithms? Appendix H might be related, but it is still unclear to me.

**Other Strengths And Weaknesses:**

The results are solid.

Both model-free and model-based methods are proposed.

Results improve the state of the art bounds.

Detailed comparison with the exact factorization case and the comparison of looseness of the bounds are discussed in the paper.

**Questions For Authors:**

The variance reduction method appears to rely on the reference function. Have you considered variance-dependent bounds, similar to those in Variance-Dependent Regret Bounds for Linear Bandits and Reinforcement Learning: Adaptivity and Computational Efficiency?

**Relation To Broader Scientific Literature:**

Many real-world applications feature structured transition probabilities and reward functions that can reduces the sample complexity of high-dimensional MDPs.

For example, robotic control problems can be effectively modeled using the factored MDP (FMDP) framework.

**Theoretical Claims:**

The theoretical bounds for model-based algorithms look good to me.

The theoretical bounds for model-free algorithms look good to me.

---

> ### Author Rebuttal · Authors · 2025-03-31
>
> We thank the reviewer for the insightful feedback and greatly appreciate the recognition of our work's novelty and contributions! Below, we provide point-by-point responses to the questions.
>
> **Q1: It might be helpful to include a proof sketch to summarize the main techniques and novelty of the proof in the main text.**
>
>
> Thank you for your constructive suggestion! We will include a "Proof Sketch" section in the revised version to highlight the key ideas and techniques behind our results.
>
> * **Sample Complexity of Model-based Method**.
> The main challenge lies in handling statistical correlations introduced by factorized sampling when estimating the finite-sample errors. The analysis relies on cross-component correlation control and variance analysis with customized Bernstein concentration bounds (e.g., Lemma B.2 and B.3-B.5).
>
> * **Sample Complexity of Model-free VRQL-AF**.
> The additional challenges for this algorithm come from the additional correlation during the two-timescale variance-reduced updates. Our improvement builds on the tailored factored empirical Bellman operator for approximately factorized MDPs, combined with a variance-reduction approach to minimize the variance of the stochastic iterative algorithm. The key idea is to show that this operator retains the non-asymptotic behavior of the standard one while achieving greater sample efficiency.
> We develop a refined statistical analysis to tightly control estimation errors and iteration variance across multiple factored components (e.g., Lemmas G.2, G.3, G.6, G.8, G.9).
>
> **Q2: Can the algorithm be extended to the case where factorization is unknown?**
>
> Thank you for the question. Yes, our algorithm works with any given factorization, even if it is not perfectly aligned with the true MDP. For each factorization, our analysis provides the corresponding sample complexity and bias guarantees. While the bias is typically unknown, the algorithm remains valid regardless.
>
> In practice—especially in FMDPs or approximately factored MDPs—domain knowledge often reveals strong and weak dependencies (e.g., Appendix B), which enables the choice of factorizations that balance sample efficiency and bias. We will clarify this in the revised version.
>
> **Q3: Does the algorithm require factorization bias term Delta as an input? If so, is it possible to generalize to the unknown bias case?**
>
> Thank you for the question. The algorithm does not require $\Delta$ as an input. While our analysis quantifies how performance depends on the factorization bias, the algorithm itself runs without access to knowing $\Delta$, and is fully applicable in the unknown bias case.
>
>
>
> **Q4: Can you comment on the computation complexity of the algorithms? Appendix H might be related, but it is still unclear to me.**
>
> This is a great point. Our algorithm consists of two components: computing the cost-optimal sampling policy and executing the RL procedure. The RL component follows standard implementation, while the cost-optimal sampling problem is discussed in more detail in Appendix H.
>
> Solving the optimal sampling policy, though combinatorial in nature, is computationally efficient for two reasons: (1) the number of factors $K$ is typically small--on the order of $\log(|\mathcal{S}||\mathcal{A}|)$, and (2) it can be solved efficiently using modern integer programming solvers (e.g., Gurobi, CPLEX) or established approximate algorithms.
>
>
> In particular, the problem reduces to a standard weighted graph coloring formulation of size $\mathcal{O}(K)$, with a lot of established solutions[1–2] that can solve near-optimal solutions within seconds even $K$ is very large (e.g., $K=5000$).
>
> >[1] Shen, Y., Sun, Y., Li, X., Eberhard, A., & Ernst, A. (2022, June). Enhancing column generation by a machine-learning-based pricing heuristic for graph coloring. AAAI.
>
> >[2] Dokeroglu, T., & Sevinc, E. (2021). Memetic Teaching–Learning-Based Optimization algorithms for large graph coloring problems. EAAI.
>
>
>
>
>
> **Q5: The variance reduction method appears to rely on the reference function. Have you considered variance-dependent bounds, similar to those in [1]?**
>
> This is a great point. The variance-dependent bounds aim to express regret or sample complexity in terms of the instance-dependent variance of the value function[1–2]. Adapting these ideas to our setting may involve incorporating variance-related quantities into the algorithm's step-wise parameters, and avoiding the standard worst-case scaling of $V$ and $Var(V)$ as $1/(1-\gamma)$ and $1/(1-\gamma)^2$. We agree this is a promising direction for future work.
>
>
> >[1] Zhao, Heyang, et al. "Variance-dependent regret bounds for linear bandits and reinforcement learning: Adaptivity and computational efficiency." COLT (2023).
>
> >[2] Yin, Ming, and Yu-Xiang Wang. "Towards instance-optimal offline reinforcement learning with pessimism." NeurIPS (2021).

---

### Official Review · Reviewer_rEVy · 2025-03-08

**Overall Recommendation:** 3

**Summary:**

This paper studied one key challenge of RL: curse of dimensionality related to high-dimensional state-action spaces. To address the challenges, the authors introduce approximate factorization, which extends the Factored Markov Decision Process (FMDP) framework to the imperfect misspecification case. Under a generative model, this paper studies both model-based and model-free RL algorithm and provide sample complexity bound.

**Claims And Evidence:**

In introduction (line 52), this paper claims that " limited extensions to non-linear methods, such as neural networks, which
typically rely on specific assumptions (Fan et al., 2020; Xu & Gu, 2020)." However, their are also a line of researches studied non-linear low-rank mdps proposed by [1], the extensions are not so "limited", I think the authors should do more literature reviews.

[1] FLAMBE: Structural Complexity and Representation Learning of Low Rank MDPs.

**Essential References Not Discussed:**

Many other papers tried to deal with curse of dimensionality by function approximation including both linear and non-linear approximation (like I state in Claims And Evidence*), I think the authors should also provide a discussion on these papers.

**Experimental Designs Or Analyses:**

This paper does not provide experiments.

**Methods And Evaluation Criteria:**

The evaluation criteria sample complexity is standard for RL problem.

**Other Comments Or Suggestions:**

line 136, right column, $P_k(s'[Z_k^S]|x)\in \mathbb{R}^{|\mathcal{X}|\times|\mathcal{S}[Z_k^S]|}$, in the defintion of the set of feasible marginal transition probabilities (line 159 left column), $P_k$ is defined to be in $\mathbb{R}^{|\mathcal{X}[Z^P_k]|\times|\mathcal{S}[Z_k^S]|}$.

**Other Strengths And Weaknesses:**

I think the biggest limitation for this paper is the assumption of generative models, which does reflect the real data collection process of RL under mdps.

Moreover, in line 200, this paper claims "In this case, instead of sampling from the global state-action space, we only need to sample from the substate-subaction spaces, which reduces the sample complexity". Such a sampling is even more unnatural.

Finally, this paper does not provide experiment to validate their theoretical results.

**Questions For Authors:**

line 136, right column, why do you need to define the set of feasible marginal transition probabilities as this set? And can you explain in high level what kind of properties does this set have?

**Relation To Broader Scientific Literature:**

This paper is related to function approximation in RL to deal with curse of dimensionality. However, the assumption of generative model makes it quite limited.

**Theoretical Claims:**

I didn't check the proof in detail.

---

> ### Author Rebuttal · Authors · 2025-03-31
>
> We thank the reviewer for their thoughtful questions. Below, we provide point-by-point responses to the reviewer's comments.
>
> **Q1: About the claim of " limited extensions to non-linear methods, such as neural networks, which typically rely on specific assumptions (Fan et al., 2020; Xu & Gu, 2020)." There exists a line of research studied non-linear low-rank mdps proposed by [1], the extensions are not so "limited".**
>
> We agree with the reviewer that there are some non-linear methods with different assumptions. In light of this, we will remove the term “limited” and include additional relevant references in the revised version.
>
> **Q2: This paper does not provide experiments to validate their theoretical results.**
>
> Due to space limitations, we included our experimental results in Appendices I and J (pages 46–50), which include a synthetic MDP and a real-world electricity storage task. These results demonstrate improved sample efficiency and validate our theoretical findings. We will highlight them more clearly in the main text in the next version, as suggested.
>
> **Q3: I think the biggest limitation for this paper is the assumption of generative models, which doesn't reflect the real data collection process of RL under mdps.**
>
> Thank you for raising this point.
>
> The generative model is a widely used and fundamental setting in the RL literature. It is particularly practical for applications with high-fidelity simulators (e.g., robotics). This setting is often the first step—prior to tackling more complex settings such as trajectory-based sampling—in developing a theoretical understanding of an algorithm’s behavior [1–3]. Even in this setting, the non-asymptotic analysis of (approximately) factored MDPs remains heavily understudied. Since we are the first to extend factorizability, establish model-free algorithms, and improve sample complexity guarantees for factored MDPs, we focus on the generative model setting in this work.
>
> That being said, we believe our algorithm and theoretical results can potentially be extended to the Markovian sampling setting. Specifically, as long as the behavior policy induces a uniformly ergodic Markov chain, the algorithm should work since each substate-subaction pair is sampled infinitely often along the trajectory. Recent techniques (e.g., [4]), based on mixing times and conditioning, also provide tools to convert generative-model bounds into trajectory-based bounds. Rigorously studying this extension is a promising direction for future work.
>
> >[1] A sparse sampling algorithm for near-optimal planning in large Markov decision processes. Machine Learning, 2002.
>
> >[2] On the Sample Complexity of Reinforcement Learning with a Generative Model. ICML 2012.
>
> >[3] Sample-efficient reinforcement learning for linearly-parameterized mdps with a generative model. NeurIPS 2021.
>
> >[4] Sample complexity of asynchronous Q-learning: Sharper analysis and variance reduction. NeurIPS 2020.
>
> **Q4: This paper claims "sample from the substate-subaction spaces". Such a sampling is even more unnatural.**
>
> We appreciate the reviewer’s concern. We do not assume direct access to substate-subaction transitions. Instead, we sample full transitions and then extract the components for specific dimensions, which is standard in factored MDPs.
>
> For example, in an electricity storage control problem with state $s=(s_1,s_2)$ where $s_1$ is the storage level and $s_2$ is the electricity price. The price $s_2$ is typically externally generated, observable, and unaffected by the system's actions. Therefore, we can estimate $P(s'_2|s_2)$ from full transition samples $(s_1,s_2,a,s'_1,s'_2)$. We will clarify this in the revised version.
>
> **Q5: $P_k$ used in line 136 and line 159 are with different shapes.**
>
> Thank you for pointing this out. We acknowledge the slight abuse of notation. We will revise the usage of $P_k$ in line 136 to make it consistent throughout the paper.
>
> **Q6: Why do you need to define the set of feasible marginal transition probabilities? And what are its properties at a high level?**
>
> Thank you for the question. At a high level, we introduce this feasible set to capture how factorization error may arise when full transitions are aggregated into factor-level marginals when the MDP is not perfectly factorizable. It thus bounds the possible deviations from perfect factorization for error definition, which is crucial for the theoretical analysis and doesn't influence algorithm design.
>
> For instance, for an MDP with state $s=(s_1,s_2)$ and action $a$, we aim to estimate the transition $P(s'_1 |s_1,a)$ of a factor. What we actually observe is $P(s'_1 |s_1,s_2,a)$ (because we need to sample from a full state-action pair). This is consistent across all $s_2$ in the perfectly factored case, but in the approximate case, different $s_2$ may introduce biases. The feasible set captures this variability and is used to define the factorization error. We will add more discussions in the revised version.

---

> > ### Comment · Reviewer_rEVy · 2025-04-02
> >
> > Thank you for the authors’ response, which partially addresses my concern. I have increased my score to 3, although I remain concerned about the generative model and hope the authors can consider extending their work to real RL sampling scenarios, perhaps as future work.

---

> > > ### Author Response · Authors · 2025-04-02
> > >
> > > We sincerely thank the reviewer for the thoughtful follow-up and for raising the score. We truly appreciate your engagement with our work and your constructive suggestion regarding extending our method to real RL sampling scenarios (Markov sampling).
> > >
> > > **Motivation**: We fully understand the concern about the use of a generative model. This design choice was made to help clearly isolate and present the core algorithmic and theoretical contributions of our work in a more "clean" setting, which is a common choice in RL theory literature [1-3]. While Markovian sampling better reflects real-world scenarios, it introduces sample dependencies and dynamics that, although quite addressable with standard techniques [4], would introduce complexity to both the analysis and its presentation in the main body—making it harder to clearly communicate our main structural contributions within the limited space of a conference paper. Incorporating these elements could have shifted the focus away from the core insights we aimed to highlight.
> > >
> > > **Justification**: That said, we fully agree with the reviewer that extending the analysis to standard Markovian sampling is a valuable direction. In fact, this extension is conceptually straightforward. To illustrate: in a generative model setting, one requires $|\mathcal{S}||\mathcal{A}|$ samples to uniformly cover all state-action pairs. In contrast, Markovian sampling under a behavior policy requires a trajectory long enough to get the samples covering all $(s, a)$ pairs. Intuitively, this only results in an additional sample size governed by coverage and mixing properties of the Markov chain, but it does not quite affect the algorithm’s design or the overall sample complexity scaling—only the constants.
> > >
> > > **Extension**: To address the reviewer's concern, we plan to include an additional lemma in the next version to formally characterize this extension and further clarify the generality of our approach.
> > >
> > > With this clarification, we now highlight the three core contributions of our work:
> > >
> > > 1. **An approximate factorization framework** that relaxes the strong assumption of perfect factorization in existing FMDPs, enabling more realistic and flexible modeling of complex systems.
> > >
> > > 2. **An optimal synchronous sampling strategy based on graph coloring**, which, for the first time, enables a single sample to be used in estimating multiple factors, significantly improving sample efficiency by reducing redundant queries across correlated components.
> > >
> > > 3. **Two corresponding algorithms with improved and near-minimax sample complexity guarantees**:
> > > (i) a model-based method under approximate factorization, and
> > > (ii) the first model-free, variance-reduced Q-learning algorithm for factored MDPs, enabled by a customized factored empirical Bellman operator and a refined statistical analysis framework.
> > >
> > > We are grateful for the reviewer’s thoughtful feedback and hope our response has clarified both the reasoning behind our design choices and how to extend our approach to more general sampling settings. We will include a corresponding formal lemma in the next version and are excited to further develop this line of research.
> > >
> > > >[1] Wang, B., Yan, Y., & Fan, J. (2021). Sample-efficient reinforcement learning for linearly-parameterized mdps with a generative model. Advances in Neural Information Processing Systems, 34, 23009-23022.
> > >
> > > >[2] Clavier, P., Shi, L., Le Pennec, E., Mazumdar, E., Wierman, A., & Geist, M. (2024). Near-Optimal Distributionally Robust Reinforcement Learning with General $ L_p $ Norms. Advances in Neural Information Processing Systems, 37, 1750-1810.
> > >
> > > >[3] Li, G., Wei, Y., Chi, Y., Gu, Y., & Chen, Y. (2020). Breaking the sample size barrier in model-based reinforcement learning with a generative model. Advances in neural information processing systems, 33, 12861-12872.
> > >
> > > >[4] Li, G., Wei, Y., Chi, Y., Gu, Y., & Chen, Y. (2020). Sample complexity of asynchronous Q-learning: Sharper analysis and variance reduction. Advances in neural information processing systems, 33, 7031-7043.

---

### Official Review · Reviewer_Curw · 2025-03-08

**Overall Recommendation:** 4

**Summary:**

The authors study Factored Markov Decision Processes (FMDPs). They extend this framework in significant ways and amend several of its shortcomings. For example, they develop a model-based RL algorithm achieving the first near-minimax sample complexity for FMDPs.

**Claims And Evidence:**

Yes.

**Essential References Not Discussed:**

No.

**Experimental Designs Or Analyses:**

Yes, to some extent.

**Methods And Evaluation Criteria:**

Yes.

**Other Comments Or Suggestions:**

No.

**Other Strengths And Weaknesses:**

None.

**Questions For Authors:**

Page 2:
Please add some discussion on the discount factor $\gamma$.

Page 3:
Should $s$ appear in the RHS of Equation (1)?

Page 4:
Can you please elaborate on the definition captured in Equation (6)?

Page 5:
Can you please elaborate on $\kappa_p \ll K_\omega$?

Page 6:
What is the main take-away from Equation (9)?

Page 7:
Algorithm 3:
How can we efficiently implement step 7?
Not sure if Equation (10) is efficient?

Page 8:
The parameters in Theorem 6.1 are a bit confusing :)

**Relation To Broader Scientific Literature:**

The authors study Factored Markov Decision Processes (FMDPs). They extend this framework in significant ways and amend several of its shortcomings.

**Theoretical Claims:**

Yes, to some extent.

---

> ### Author Rebuttal · Authors · 2025-03-31
>
> We would like to thank the reviewer for acknowledging the contributions of our work and providing insightful feedback! Below, we provide detailed responses to the reviewer's questions.
>
> **Q1: Please add some discussion on the discount factor $\gamma$.**
>
> In the discounted setting (which is standard in MDPs and reinforcement learning), the discount factor $\gamma \in [0, 1)$ determines how much weight the agent places on immediate versus future rewards. Smaller values prioritize short-term gains, while larger values emphasize long-term planning. We will include a more detailed discussion in the next version.
>
> **Q2: Should $s$ appear in the RHS of Equation (1)?**
>
> The current state $s$ is present on the right-hand side through the variable $x=(s,a)$, which denotes the state-action pair.
>
> **Q3: Can you please elaborate on the definition captured in Equation (6)?**
>
> Thank you for your constructive comment. Eq.(6) defines the **joint sampling set** used to simultaneously sample transitions for two factors $k_1$ and $k_2$, whose scopes are $Z_{k_1}$ and $Z_{k_2}$, respectively.
> Here, $x[Z_{k_1}^P]$ and $x[Z_{k_2}^P]$ enumerate the relevant input dimensions (by sampling from these dimensions, we can estimate the respective transitions of factors) for the two factors, $x^{\text{default}}[- (Z_{k_1}^P\cup Z_{k_2}^P)]$ assigns fixed default values for all the other irrelevant dimensions. The modulo operation ensures that the sampling cycles through all possible values within each relevant scope, and $D_{\max} = \max(|\mathcal{X}[Z_{k_1}^P]|,|\mathcal{X}[Z_{k_2}^P]|)$ ensures full coverage of both factors' input spaces. This construction enables compact and efficient joint sampling, allowing shared samples to be reused across different factors.
>
> **Q4: Can you please elaborate on $\kappa_p\ll K_\omega$?**
>
> As discussed in Sections 4.1, 5.2, and Appendix H, the relationship $\kappa_p \ll K_\omega$ arises from the graph coloring perspective. We construct an undirected graph with $K_\omega$ nodes—one per factor—and edges between dependent factors. The **chromatic number** $\kappa_p$ is the minimum number of colors needed so that no adjacent nodes share a color, enabling efficient sampling in our setting.
>
>
> Intuitively, $\kappa_p$ can be much smaller than $K_\omega$, especially when the graph is sparse. For example, in a star graph with one central node connected to $K_\omega-1$ outer nodes (which are not connected to each other), only two colors are needed: one for the center and another shared by all outer nodes. Thus, $\kappa_p=2\ll K_\omega$. In practice, many FMDPs exhibit sparse interaction structures among factors, leading to graphs with low chromatic numbers. Therefore, in such settings, it is common that $\kappa_p\ll K_\omega$ and even $\kappa_p= \mathcal{O}(1)$.
>
> **Q5: What is the main take-away from Equation (9)?**
>
> The key takeaway is that our algorithm significantly reduces the sample complexity depending on the size of the state-action space. Specifically, our dependence is $\sum_{k \in [\kappa_p]} |\mathcal{X}[Z^P_k]|$.
> 1. Compared with classical MDPs with dependence $|\mathcal{S}||\mathcal{A}|$, ours is much smaller because $|\mathcal{X}[Z^P_k]|$ is the state-action space size of a single factor, which is exponentially smaller than $|\mathcal{S}||\mathcal{A}|$.
> 2. Compared with SOTA FMDP results, which is $\sum_{k \in [K_\omega]} |\mathcal{X}[Z^P_k]|$, ours is also much smaller because we have shown $\kappa_p\ll K_\omega$ before.
> 3. We show that when $\kappa_p= \mathcal{O}(1)$, our result is minimax optimal.
>
>
> **Q6: Algorithm 3: How can we efficiently implement step 7? Not sure if Equation (10) is efficient?**
>
> Thank you for the comment. Equation (10) provides a formal and rigorous definition of the empirical factored Bellman operator used in Step 7 of Algorithm 4. While the notation may seem complex, the implementation is efficient in practice.
>
> Due to the synchronous sampling design, a single sample can be reused to estimate transitions for multiple factors. Equation (10) simply extracts the relevant dimensions from each sample to update the corresponding factors. This makes the implementation straightforward and efficient.
>
> We will include more discussions in the next version.
>
>
>
> **Q7: The parameters in Theorem 6.1 are a bit confusing :)**
>
> Thank you for the comment. The parameters in Theorem 6.1 correspond to key configuration choices in the VRQL-AF algorithm, including the number of epochs, epoch length, learning rate, and others. We agree that their roles could be explained more clearly and will revise the theorem statement and surrounding text to improve clarity in the next version.

---

### Official Review · Reviewer_TcCo · 2025-03-10

**Overall Recommendation:** 3

**Summary:**

This paper studied factorized Markov Decision Processes (MDPs) and proposed two algorithms based on a newly developed synchronous sampling methods. This method efficiently sample and estimate different component of the transitions and therefore help the algorithm to achieve less sample complexity in both model-based and model-free approach. This paper also characterize the effect of imperfect modeling, when the MDP is an approximated factorized MDP.

**Claims And Evidence:**

Yes. The claims are generally well-supported by the analysis and results.

**Essential References Not Discussed:**

No

**Experimental Designs Or Analyses:**

The experiments in appendix are generally sound and valid. One potential issue is the lack of comparison to a policy-based method such as policy gradient.

**Methods And Evaluation Criteria:**

Yes.

**Other Comments Or Suggestions:**

N/A

**Other Strengths And Weaknesses:**

Strength: the synchronize sampling method seems interesting and novel.

Weaknesses:  The cost-optimal sampling problem might be hard to solve.

**Questions For Authors:**

This paper studied tabular RL, where the state and action space are finite. Is it possible to consider infinite state space? Because it is most likely the case in real-world applications.

**Relation To Broader Scientific Literature:**

N/A

**Theoretical Claims:**

I did not fully check the proof, but it seems correct to me based on the main paper.

---

> ### Author Rebuttal · Authors · 2025-03-31
>
> We would like to thank the reviewer for acknowledging our novelty and contributions and providing insightful feedback. Below, we provide detailed responses to address the reviewer's comments.
>
> **Q1: The experiments in Appendix are generally sound and valid. One potential issue is the lack of comparison to a policy-based method such as policy gradient.**
>
> Thank the reviewer for acknowledging the soundlessness of our experiments. Since our work mainly focuses on leveraging approximate factorizable structures to design customized value-based algorithms, we primarily compare against the same types of value-based baselines. That being said, the proposed algorithmic framework can also be used to solve the policy evaluation subproblem (which is based on solving Bellman equations) within an actor-critic framework. This can be combined with policy gradient methods (i.e., the actor) to develop efficient policy-based algorithms that leverage the approximate factorization structure. We leave providing theoretical guarantees and conducting numerical simulations in this direction as future work.
>
> **Q2: The cost-optimal sampling problem might be hard to solve.**
>
> Thank the reviewer for raising this point. Intuitively, the cost-optimal sampling problem involves grouping the $K$ factors of the MDP to minimize the total sampling cost. While the problem may appear combinatorial, it is computationally efficient in our setting for two reasons: (1) the number of factors $K$ is typically small (on the order of $\log(|\mathcal{S}||\mathcal{A}|)$, where $|\mathcal{S}||\mathcal{A}|$ is the state-action space size), (2) there exist established exact and approximate algorithms to solve such problems efficiently.
>
>
> Specifically, the problem can be formulated as an integer program of size $\mathcal{O}(K)$, solvable by modern solvers like Gurobi and CPLEX when $K$ is moderate (e.g., $K\leq 500$). Furthermore, as discussed in Appendix H, it can be reduced to the classical weighted graph coloring problem, for which many scalable algorithms exist. In practice, even problems with thousands of nodes (e.g., $K=5000$) can be solved to near-optimality within seconds [2–4].
>
> To address the reviewer's concern, we will include a more detailed discussion in the revised version.
>
> >[1] Chen, X., Hu, J., Li, L., & Wang, L. Efficient Reinforcement Learning in Factored MDPs with Application to Constrained RL. In International Conference on Learning Representations. (2021)
>
> >[2]  Shen, Y., Sun, Y., Li, X., Eberhard, A., & Ernst, A. (2022, June). Enhancing column generation by a machine-learning-based pricing heuristic for graph coloring. In Proceedings of the AAAI conference on artificial intelligence (Vol. 36, No. 9, pp. 9926-9934).
>
> >[3] Saeed, A., Husnain, A., Zahoor, A., & Gondal, R. M. (2024). A comparative study of cat swarm algorithm for graph coloring problem: Convergence analysis and performance evaluation. International Journal of Innovative Research in Computer Science and Technology (IJIRCST), 12(4), 1-9.
>
> >[4] Dokeroglu, T., & Sevinc, E. (2021). Memetic Teaching–Learning-Based Optimization algorithms for large graph coloring problems. Engineering Applications of Artificial Intelligence, 102, 104282.
>
> **Q3: This paper studied tabular RL, where the state and action space are finite. Is it possible to consider infinite state space? Because it is most likely the case in real-world applications.**
>
>
> We thank the reviewer for their question. While our work focuses on the tabular setting to enable a clear theoretical understanding, we believe it can potentially be extended to MDPs with infinite (or continuous) state spaces. Specifically, the core structural insight—factored transition kernels—can naturally extend to continuous domains. For example, a discrete factorization $P(s'_1, s'_2 \mid s_1, s_2) = P(s'_1 \mid s_1) P(s'_2 \mid s_2)$ can be analogously expressed in terms of transition densities as $f(s'_1, s'_2 \mid s_1, s_2) = f(s'_1 \mid s_1) f(s'_2 \mid s_2)$ in the continuous case. However, working with MDPs with infinite (or continuous) state spaces presents other challenges, such as the potential unboundedness of the reward function (e.g., in the LQR setting in control). In addition, certain parameterizations (i.e., function approximation) might be required to ensure that Q-functions are updated infinitely often for each state-action pair. We view this as a promising direction for future work.

---

### Decision · Program_Chairs · 2025-05-01

**Decision:**

Accept (poster)

**Comment:**

This paper presents a theoretical contribution to reinforcement learning in Factored Markov Decision Processes (FMDPs), introducing novel techniques aimed at improving sample efficiency and applicability. Reviewers recognized several key strengths: the development of an innovative synchronous sampling method, the formalization and handling of approximate factorizations (addressing model misspecification), and the derivation of strong theoretical results. The paper provides sample complexity bounds for both model-based and model-free algorithms (including variance reduction) that demonstrably improve upon state-of-the-art guarantees, with near-minimax rates achieved in the model-based setting. However, some weaknesses and concerns were also raised. The reliance on a generative model was noted as a potential limitation regarding practical RL scenarios by one reviewer. Another pointed out the absence of experimental validation within the main paper (though numerical results are in the appendix) and suggested a potentially incomplete discussion of related function approximation literature. Questions regarding the practical implementation (e.g., solving the sampling problem, computational complexity) and potential extensions (e.g., continuous spaces, unknown factorization) were also brought up. Despite these concerns, the overall sentiment is positive, leaning towards acceptance (2 Accepts, 2 Weak Accepts), based on the significance and solidity of the theoretical advancements for FMDPs.